# CASCADING REINFORCEMENT LEARNING

**Yihan Du**
Electrical and Computer Engineering
University of Illinois Urbana-Champaign
Urbana, IL 61801, USA
yihandu@illinois.edu

**R. Srikant**[*]
Electrical and Computer Engineering
University of Illinois Urbana-Champaign
Urbana, IL 61801, USA
rsrikant@illinois.edu

**Wei Chen**[*]
Microsoft Research
Beijing 100080, China
weic@microsoft.com

## ABSTRACT

Cascading bandits have gained popularity in recent years due to their applicability to recommendation systems and online advertising. In the cascading bandit model, at each timestep, an agent recommends an ordered subset of items (called an item list) from a pool of items, each associated with an unknown attraction probability. Then, the user examines the list, and clicks the first attractive item (if any), and after that, the agent receives a reward. The goal of the agent is to maximize the expected cumulative reward. However, the prior literature on cascading bandits ignores the influences of user states (e.g., historical behaviors) on recommendations and the change of states as the session proceeds. Motivated by this fact, we propose a generalized cascading RL framework, which considers the impact of user states and state transition into decisions. In cascading RL, we need to select items not only with large attraction probabilities but also leading to good successor states. This imposes a huge computational challenge due to the combinatorial action space. To tackle this challenge, we delve into the properties of value functions, and design an oracle `BestPerm` to efficiently find the optimal item list. Equipped with `BestPerm`, we develop two algorithms `CascadingVI` and `CascadingBPI`, which are both computation-efficient and sample-efficient, and provide near-optimal regret and sample complexity guarantees. Furthermore, we present experiments to show the improved computational and sample efficiencies of our algorithms compared to straightforward adaptations of existing RL algorithms in practice.

## 1 INTRODUCTION

In recent years, a model called cascading bandits (Kveton et al., 2015; Combes et al., 2015; Li et al., 2016; Vial et al., 2022) has received extensive attention in the online learning community, and found various applications such as recommendation systems (Mary et al., 2015) and online advertising (Tang et al., 2013). In this model, an agent is given a ground set of items, each with an unknown attraction probability. At each timestep, the agent recommends an ordered list of items to a user, and the user examines the items one by one, where the probability that each item attracts the user is equal to its attraction probability. Then, the user clicks the first attractive item (if any), and skips the following items. If an item is clicked in the list, the agent receives a reward; if no item is clicked, the agent receives no reward. The objective of the agent is to maximize the expected cumulative reward.

While the cascading bandit model has been extensively studied, it neglects the influences of user states (e.g., users' past behaviors) on recommendations, and the fact that states can transition as users take actions. For example, in personalized video recommendation, the recommendation system usually suggests a list of videos according to the characteristics and viewing records of users. If the user clicks a video, the environment of the system can transition to a next state that stores the

---

[*]Corresponding authors.

user's latest behavior and interest. Next, the recommendation system will suggest videos of a similar type as what the user watched before to improve the click-through rate. While contextual cascading bandits (Li et al., 2016; Zong et al., 2016) can also be used to formulate users' historical behaviors as part of contexts, cascading RL is more suitable for long-term reward maximization, since it considers potential rewards from future states. For instance, to maximize the long-term reward, the recommendation system may recommend videos of TV series, since users may keep watching subsequent videos of the same TV series once they get attracted to one of them.

To model such state-dependent behavior, we propose a novel framework called cascading reinforcement learning (RL), which generalizes the conventional cascading bandit model to depict the influence of user states on recommendations and the transition of states in realistic applications. In this framework, there is a pool of $N$ items and a space of states. Each state-item pair has an unknown attraction probability, an underlying transition distribution and a deterministic reward. In each episode (e.g., a session in recommendation systems), at each step, the agent first observes the current state (e.g., the user's past behavior in the session), and recommends a list of at most $m \leq N$ items. Then, the user goes over the list one by one, and clicks the first interesting item. After that, the agent receives a reward, and transitions to a next state according to the current state and clicked item. If no item in the list interests the user, the agent receives zero reward, and transitions to a next state according to the current state and a virtual item $a_\perp$. Here we say that the user clicks $a_\perp$ if no item is clicked. We define a policy as a mapping from the space of states and the current step to the space of item lists, and the optimal policy as the policy that maximizes the expected cumulative reward.

Our work distinguishes itself from prior cascading bandit studies (Kveton et al., 2015; Vial et al., 2022) on its unique computational challenge. In prior cascading bandit studies, there is no state, and they only need to select $m$ items with the highest attraction probabilities, which does not involve computational difficulty. In contrast, in cascading RL, states also matter, and we need to balance between the maximization of the attraction probabilities of chosen items and the optimization of the expected reward obtained from future states. This poses a great computational difficulty in the planning of the optimal policy under a combinatorial action space. Moreover, the combinatorial action space also brings a challenge on sample efficiency, i.e., how to avoid a dependency on the exponential number of actions in sample complexity.

To handle these challenges, we conduct a fine-grained analysis on the properties of value functions, and design an efficient oracle `BestPerm` to find the optimal item list, based on a novel dynamic programming for combinatorial optimization. Furthermore, we design computation-efficient and sample-efficient algorithms `CascadingVI` and `CascadingBPI`, which employ oracle `BestPerm` to only maintain the estimates for items and avoid enumerating over item lists. Finally, we also conduct experiments to demonstrate the superiority of our algorithms over naive adaptations of classic RL algorithms in computation and sampling.

The contributions of this work are summarized as follows.

- We propose the cascading RL framework, which generalizes the traditional cascading bandit model to formulate the influence of user states (e.g., historical behaviors) on recommendations and the change of states through time. This framework can be applied to various real-world scenarios, e.g., personalized recommendation systems and online advertising.

- To tackle the computational challenge of cascading RL, we leverage the properties of value functions to develop a novel oracle `BestPerm`, which uses a carefully-designed dynamic programming to efficiently find the optimal item list under combinatorial action spaces.

- For the regret minimization objective, with oracle `BestPerm`, we design an efficient algorithm `CascadingVI`, and establish a $\tilde{O}(H\sqrt{HSNK})$ regret, which matches a known lower bound for the general episodic RL setting up to $\tilde{O}(\sqrt{H})$. Here $S$ is the number of states, $H$ is the length of each episode, and $K$ is the number of episodes. Note that the regret depends only on the number of items $N$, instead of the exponential number of item lists.

- For the best policy identification objective, we devise a computation and sample efficient algorithm `CascadingBPI`, and provide $\tilde{O}(\frac{H^3SN}{\varepsilon^2})$ sample complexity. `CascadingBPI` is optimal up to a factor of $\tilde{O}(H)$ when $\varepsilon < \frac{H}{S^2}$, where $\varepsilon$ is an accuracy parameter.

## 2    RELATED WORK

**Cascading bandits.**  Kveton et al. (2015) and Combes et al. (2015) concurrently introduce the cascading bandit model, and design algorithms based on upper confidence bounds. Cheung et al. (2019) and Zhong et al. (2021) propose Thompson Sampling-type (Thompson, 1933) algorithms. Vial et al. (2022) develop algorithms equipped with variance-aware confidence intervals, and achieve near-optimal regret bounds. In these cascading bandit studies, there is no state (context), and the order of selected items does not matter. Thus, they only need to select $m$ items with the maximum attraction probabilities. Li et al. (2016); Zong et al. (2016) and Li & Zhang (2018) study linear contextual cascading bandits. In (Zong et al., 2016; Li & Zhang, 2018), the attraction probabilities depend on contexts, and the order of selected items still does not matter. Hence, they need to select $m$ items with the maximum attraction probabilities in the current context. Li et al. (2016) consider position discount factors, where the order of selected items is important.

Different from the above studies, our cascading RL formulation further considers state transition, and the attraction probabilities and rewards depend on states. Thus, we need to put the items that induce higher expected future rewards in the current state in the front. In addition, we require to both maximize the attraction probabilities of selected items, and optimize the potential future reward.

**Provably efficient RL.** In recent years, there have been a number of studies on provably efficient RL, e.g., (Jaksch et al., 2010; Dann & Brunskill, 2015; Azar et al., 2017; Jin et al., 2018; Zanette & Brunskill, 2019). However, none of the above studies tackle the challenge of combinatorial action space and the computation and sample efficiency issues it brings. In contrast, we have to directly face this challenge when we generalize cascading bandits to the RL framework to better formulate the state transition in real-world recommendation and advertising applications, and we provide satisfactory solutions with near optimal computation and sample efficiency.

## 3    PROBLEM FORMULATION

We consider an episodic cascading Markov decision process (MDP) $\mathcal{M}(\mathcal{S}, A^{\text{ground}}, \mathcal{A}, m, q, p, r, H)$. Here $\mathcal{S}$ is the space of states. $A^{\text{ground}} := \{a_1, \ldots, a_N, a_\perp\}$ is the ground set of items, where $a_1, \ldots, a_N$ are regular items and $a_\perp$ is a *virtual item*. Item $a_\perp$ is put at the end of each item list by default, which represents that no item in the list is clicked. An action (i.e., item list) $A$ is a permutation which consists of at least one and at most $m \le N$ regular items and the item $a_\perp$ at the end. $\mathcal{A}$ is the collection of all actions. We use "action" and "item list" interchangeably throughout the paper. For any $A \in \mathcal{A}$ and $i \in [|A|]$, let $A(i)$ denote the $i$-th item in $A$, and we have $A(|A|) = a_\perp$.

For any $(s, a) \in \mathcal{S} \times A^{\text{ground}}$, $q(s, a) \in [0, 1]$ denotes the attraction probability, which gives the probability that item $a$ is clicked in state $s$; $r(s, a) \in [0, 1]$ is the deterministic reward of clicking item $a$ in state $s$; $p(\cdot|s, a) \in \triangle_{\mathcal{S}}$ is the transition distribution, so that for any $s' \in \mathcal{S}$, $p(s'|s, a)$ gives the probability of transitioning to state $s'$ if item $a$ is clicked in state $s$. For any $s \in \mathcal{S}$, we define $q(s, a_\perp) := 1$ and $r(s, a_\perp) := 0$ (because if no regular item is clicked, the agent must click $a_\perp$ and receives zero reward). Transition probability $p(\cdot|s, a_\perp)$ allows to formulate interesting transition scenarios that could happen when none of the recommended items is clicked. One of such scenarios is that the session ends with no more reward, which is modeled by letting $p(s_0|s, a_\perp) = 1$ where $s_0$ is a special absorbing state that always induces zero reward.

$H$ is the length of each episode. In addition, we define a deterministic policy $\pi = \{\pi_h : \mathcal{S} \to \mathcal{A}\}_{h \in [H]}$ as a collection of $H$ mappings from the state space to the action space, so that $\pi_h(s)$ gives what item list to choose in state $s$ at step $h$.

The cascading RL game is as follows. In each episode $k$, an agent first chooses a policy $\pi^k$, and starts from a fixed initial state $s_1^k := s_1$. At each step $h \in [H]$, the agent first observes the current state $s_h^k$ (e.g., stored historical behaviors of the user), and then selects an item list $A_h^k = \pi_h^k(s_h^k)$ according to her policy. After that, the environment (user) browses the selected item list one by one following the order given in $A_h^k$. When the user browses the $i$-th item $A_h^k(i)$ ($i \in [|A_h^k|]$), there is a probability of $q(s_h^k, A_h^k(i))$ that the user is attracted by the item and clicks it. This attraction and clicking event is independent among all items. Once an item $A_h^k(i)$ is clicked, the agent observes which item is clicked and receives reward $r(s_h^k, A_h^k(i))$, skipping the subsequent items $\{A_h^k(j)\}_{i<j\le|A_h^k|}$, and the system transitions to a next state $s_{h+1}^k \sim p(\cdot|s_h^k, A_h^k(i))$. On the other hand, if no regular item in $A_h^k$

is clicked, i.e., we say that $a_\perp$ is clicked, then the agent receives zero reward and transitions to a next state $s_{h+1}^k \sim p(\cdot|s_h^k, a_\perp)$. After step $H$, this episode ends and the agent enters the next episode.

For any $k > 0$ and $h \in [H]$, we use $I_{k,h}$ to denote the index of the clicked item in $A_h^k$, and $I_{k,h} = |A_h^k|$ if no regular item is clicked. In our cascading RL model, the agent only observes the attraction of items $\{A_h^k(j)\}_{j \le I_{k,h}}$ (i.e., items $\{A_h^k(j)\}_{j < I_{k,h}}$ are not clicked and item $A_h^k(i)$ is clicked), and the state $s_{h+1}^k$ the system transitions into based on the unknown $p(\cdot|s_h^k, A_h^k(I_{k,h}))$. Whether the user would click any item in $\{A_h^k(j)\}_{j > I_{k,h}}$ is *unobserved*.

For any policy $\pi$, $h \in [H]$ and $s \in \mathcal{S}$, we define value function $V_h^\pi : \mathcal{S} \to [0, H]$ as the expected cumulative reward that can be obtained under policy $\pi$, starting from state $s$ at step $h$, till the end of the episode. Formally, $V_h^\pi(s) := \mathbb{E}_{q,p,\pi}[\sum_{h'=h}^H r(s_{h'}, A_{h'}(I_{h'}))|s_h = s]$. Similarly, for any policy $\pi$, $h \in [H]$ and $(s, A) \in \mathcal{S} \times \mathcal{A}$, we define Q-value function $Q_h^\pi : \mathcal{S} \times \mathcal{A} \to [0, H]$ as the expected cumulative reward received under policy $\pi$, starting from $(s, A)$ at step $h$, till the end of the episode, i.e., $Q_h^\pi(s, A) := \mathbb{E}_{q,p,\pi}[\sum_{h'=h}^H r(s_{h'}, A_{h'}(I_{h'}))|s_h = s, A_h = A]$. Since $\mathcal{S}$, $\mathcal{A}$ and $H$ are finite, there exists a deterministic optimal policy $\pi^*$ which always gives the maximum value $V_h^{\pi^*}(s)$ for all $s \in \mathcal{S}$ and $h \in [H]$ (Sutton & Barto, 2018). The Bellman equation and Bellman optimality equation for cascading RL can be stated as follows.

$$\begin{cases} Q_h^\pi(s, A) = \sum_{i=1}^{|A|} \prod_{j=1}^{i-1} \Big(1 - q(s, A(j))\Big) q(s, A(i)) \Big(r(s, A(i)) + p(\cdot|s, A(i))^\top V_{h+1}^\pi\Big), \\ V_h^\pi(s) = Q_h^\pi(s, \pi_h(s)), \\ V_{H+1}^\pi(s) = 0, \quad \forall s \in \mathcal{S}, \forall A \in \mathcal{A}, \forall h \in [H]. \end{cases}$$

$$\begin{cases} Q_h^*(s, A) = \sum_{i=1}^{|A|} \prod_{j=1}^{i-1} \Big(1 - q(s, A(j))\Big) q(s, A(i)) \Big(r(s, A(i)) + p(\cdot|s, A(i))^\top V_{h+1}^*\Big), \\ V_h^*(s) = \max_{A \in \mathcal{A}} Q_h^*(s, A), \\ V_{H+1}^*(s) = 0, \quad \forall s \in \mathcal{S}, \forall A \in \mathcal{A}, \forall h \in [H]. \end{cases} \quad (1)$$

Here $\prod_{j=1}^{i-1}(1 - q(s, A(j)))q(s, A(i))$ is the probability that item $A(i)$ is clicked, which captures the cascading feature. $r(s, A(i)) + p(\cdot|s, A(i))^\top V_{h+1}^*$ is the expected cumulative reward received from step $h$ onward if item $A(i)$ is clicked at step $h$. $Q_h^*(s, A)$ is the summation of the expected cumulative reward over each item in $A$. In this work, we focus on the tabular setting where $S$ is not too large. This is practical in category-based recommendation applications. For example, in video recommendation, the videos are categorized into multiple types, and the recommendation system can suggest videos according to the types of the latest one or two videos that the user just watched.

We investigate two popular objectives in RL, i.e., regret minimization and best policy identification. In the regret minimization setting, the agent plays $K$ episodes with the goal of minimizing the regret $\mathcal{R}(K) = \sum_{k=1}^K V_1^*(s_1) - V_1^{\pi^k}(s_1)$. In the best policy identification setting, given a confidence parameter $\delta \in (0, 1)$ and an accuracy parameter $\varepsilon$, the agent aims to identify an $\varepsilon$-optimal policy $\hat{\pi}$ which satisfies $V_1^{\hat{\pi}}(s_1) \ge V_1^*(s_1) - \varepsilon$ with probability at least $1 - \delta$, using as few episodes as possible. Here the performance is measured by the number of episodes used, i.e., sample complexity.

## 4 AN EFFICIENT ORACLE FOR CASCADING RL

In the framework of model-based RL, it is typically assumed that one has estimates (possibly including exploration bonuses) of the model, and then a planning problem is solved to compute the value functions. In our problem, this would correspond to solving the maximization in Eq. (1). Different from planning in classic RL (Jaksch et al., 2010; Azar et al., 2017; Sutton & Barto, 2018), in cascading RL, a naive implementation of this maximization will incur exponential computation complexity due to the fact that we have to consider all $A \in \mathcal{A}$.

To tackle this computational difficulty, we note that each backward recursion step in Eq. (1) involves the solution to a combinatorial optimization of the following form: For any ordered subset of $A^{\text{ground}}$

denoted by $A$, $u : A^{\text{ground}} \rightarrow \mathbb{R}$ and $w : A^{\text{ground}} \rightarrow \mathbb{R}$,

$$\max_{A \in \mathcal{A}} f(A, u, w) := \sum_{i=1}^{|A|} \prod_{j=1}^{i-1} \Big( 1 - u(A(j)) \Big) u(A(i)) w(A(i)). \tag{2}$$

Here $f(A, u, w)$ corresponds to $Q_h^*(s, A)$, $u(A(i))$ represents $q(s, A(i))$, and $w(A(i))$ stands for $r(s, A(i)) + p(\cdot|s, A(i))^\top V_{h+1}^*$. We have $u(a_\perp) = 1$ to match with $q(s, a_\perp) = 1$.

## 4.1 CRUCIAL PROPERTIES OF PROBLEM (2)

Before introducing an efficient oracle to solve problem Eq. (2), we first exhibit several nice properties of this optimization problem, which serve as the foundation of our oracle design.

For any subset of items $X \subseteq A^{\text{ground}}$, let $\text{Perm}(X)$ denote the collection of permutations of the items in $X$, and $\text{DesW}(X) \in \text{Perm}(X)$ denote the permutation where items are sorted in descending order of $w$. For convenience of analysis, here we treat $a_\perp$ as an ordinary item as $a_1, \dots, a_N$, and $a_\perp$ can appear in any position in the permutations in $\text{Perm}(X)$.

**Lemma 1.** *The weighted cascading reward function $f(A, u, w)$ satisfies the following properties:*

*(i) For any $u$, $w$ and $X \subseteq A^{\text{ground}}$, we have*

$$f(\text{DesW}(X), u, w) = \max_{A \in \text{Perm}(X)} f(A, u, w).$$

*(ii) For any $u$, $w$ and disjoint $X, X' \subseteq A^{\text{ground}} \setminus \{a_\perp\}$ such that $w(a) > w(a_\perp)$ for any $a \in X, X'$, we have*

$$f((\text{DesW}(X), a_\perp), u, w) \leq f((\text{DesW}(X \cup X'), a_\perp), u, w).$$

*Furthermore, for any $u$, $w$ and disjoint $X, X' \subseteq A^{\text{ground}} \setminus \{a_\perp\}$ such that $w(a) > w(a_\perp)$ for any $a \in X$, and $w(a) < w(a_\perp)$ for any $a \in X'$, we have*

$$f((\text{DesW}(X, X'), a_\perp), u, w) \leq f((\text{DesW}(X), a_\perp), u, w).$$

Property (i) can be proved by leveraging the continued product structure in $f(A, u, w)$ and a similar analysis as the interchange argument (Bertsekas & Castanon, 1999; Ross, 2014). Property (ii) follows from property (i) and the fact that $u(a_\perp) = 1$. The detailed proofs are presented in Appendix B.

**Remark.** Property (i) exhibits that when fixing a subset of $A^{\text{ground}}$, the best order of this subset is to rank items in descending order of $w$. Then, the best permutation selection problem in Eq. (2) can be reduced to a *best subset selection* problem.

Then, a natural question is what items and how many items should the best subset (permutation) include? Property (ii) gives an answer —— we should *include* the items with weights above $w(a_\perp)$, and *discard* the items with weights below $w(a_\perp)$. The intuition behind is as follows: If a permutation does not include the items in $X'$ such that $w(a) > w(a_\perp)$ for any $a \in X'$, this is equivalent to putting $X'$ behind $a_\perp$, since $u(a_\perp) = 1$. Then, according to property (i), we can arrange the items in $X'$ in front of $a_\perp$ (i.e., include them) to obtain a better permutation. Similarly, if a permutation includes the items $X'$ such that $w(a) < w(a_\perp)$ for any $a \in X'$, from property (i), we can also obtain a better permutation by putting $X'$ behind $a_\perp$, i.e., discarding them.

## 4.2 ORACLE BestPerm

Making use of the properties of $f(A, u, w)$, we develop an efficient oracle `BestPerm` to solve problem Eq. (2), based on a carefully-designed dynamic programming to find the optimal item subset.

Algorithm 1 presents the pseudo-code of oracle `BestPerm`. Given attraction probabilities $u$ and weights $w$, we first sort the items in $A^{\text{ground}}$ in descending order of $w$, and denote the sorted sequence by $a_1, \dots, a_J, a_\perp, a_{J+1}, \dots, a_N$. Here $J$ denotes the number of items with weights above $w(a_\perp)$. If $J = 0$, i.e., $a_\perp$ has the highest weight, since the solution must contain at least one item, we choose the best single item as the solution (lines 3-4). If $1 \leq J \leq m$, $(a_1, \dots, a_J)$ satisfies the cardinality constraint and is the best permutation (lines 6-7). If $J > m$, to meet the cardinality constraint, we

---

**Algorithm 1:** BestPerm: find $\text{argmax}_{A \in \mathcal{A}} f(A, u, w)$ and $\max_{A \in \mathcal{A}} f(A, u, w)$

---

**Input:** $A^{\text{ground}}$, $u : A^{\text{ground}} \to [0, 1]$, $w : A^{\text{ground}} \to \mathbb{R}$.

1  Sort the items in $A^{\text{ground}}$ in descending order of $w$, and denote the sorted sequence by
   $a_1, \ldots, a_J, a_\perp, a_{J+1}, \ldots, a_N$. Here $J$ denotes the number of items with weights above $w(a_\perp)$;

2  **if** $J = 0$ **then**

3  $\quad$ $a' \leftarrow \text{argmax}_{a \in \{a_1, \ldots, a_N\}} \{u(a)w(a) + (1 - u(a))w(a_\perp)\}$;   // Select the best single item

4  $\quad$ $S^{\text{best}} \leftarrow (a')$. $F^{\text{best}} \leftarrow u(a')w(a') + (1 - u(a'))w(a_\perp)$;

5  **if** $1 \leq J \leq m$ **then**

6  $\quad$ $S^{\text{best}} \leftarrow (a_1, \ldots, a_J)$;                                     // Simply output $(a_1, \ldots, a_J)$

7  $\quad$ $F^{\text{best}} \leftarrow \sum_{i=1}^{J} \prod_{j=1}^{i-1} (1 - u(a_j))u(a_i)w(a_i) + \prod_{j=1}^{J}(1 - u(a_j))w(a_\perp)$;

8  **if** $J > m$ **then**

9  $\quad$ $S[J][1] \leftarrow (a_J)$;                                         // Select $m$ best items from $(a_1, \ldots, a_J)$

10 $\quad$ $F[J][1] \leftarrow u(a_J)w(a_J) + (1 - u(a_J))w(a_\perp)$;

11 $\quad$ For any $i \in [J]$, $S[i][0] \leftarrow \emptyset$ and $F[i][0] \leftarrow w(a_\perp)$;

12 $\quad$ For any $i \in [J]$ and $J - i + 1 < k \leq m$, $F[i][k] \leftarrow -\infty$;

13 $\quad$ **for** $i = J - 1, J - 2, \ldots, 1$ **do**

14 $\quad\quad$ **for** $k = 1, 2, \ldots, \min\{m, J - i + 1\}$ **do**

15 $\quad\quad\quad$ **if** $F[i+1][k] \geq w(a_i)u(a_i) + (1 - u(a_i))F[i+1][k-1]$ **then**

16 $\quad\quad\quad\quad$ $S[i][k] \leftarrow S[i+1][k]$. $F[i][k] \leftarrow F[i+1][k]$;

17 $\quad\quad\quad$ **else**

18 $\quad\quad\quad\quad$ $S[i][k] \leftarrow (a_i, S[i+1][k-1])$;

19 $\quad\quad\quad\quad$ $F[i][k] \leftarrow u(a_i)w(a_i) + (1 - u(a_i))F[i+1][k-1]$;

20 $\quad$ $S^{\text{best}} \leftarrow S[1][m]$. $F^{\text{best}} \leftarrow F[1][m]$;

21 **return** $F^{\text{best}}, S^{\text{best}}$;

---

need to *select $m$ best items from* $(a_1, \ldots, a_J)$ which maximize the objective value, i.e.,

$$\max_{(a_1', \ldots, a_m') \subseteq (a_1, \ldots, a_J)} f((a_1', \ldots, a_m'), u, w). \tag{3}$$

Eq. (3) is a challenging combinatorial optimization problem, and costs exponential computation complexity if one performs naive exhaustive search. To solve Eq. (3), we resort to dynamic programming. For any $i \in [J]$ and $k \in [\min\{m, J - i + 1\}]$, let $S[i][k]$ and $F[i][k]$ denote the optimal solution and optimal value of the problem $\max_{(a_1', \ldots, a_k') \subseteq (a_i, \ldots, a_J)} f((a_1', \ldots, a_k'), u, w)$. Utilizing the structure of $f(A, u, w)$, we have

$$\begin{cases} F[J][1] = u(a_J)w(a_J) + (1 - u(a_J))w(a_\perp), & \\ F[i][0] = w(a_\perp), & 1 \leq i \leq J, \\ F[i][k] = -\infty, & 1 \leq i \leq J, \ J - i + 1 < k \leq m, \\ F[i][k] = \max\{F[i+1][k], & \\ \quad u(a_i)w(a_i) + (1 - u(a_i))F[i+1][k-1]\}, & 1 \leq i \leq J - 1, \ 1 \leq k \leq \min\{m, J - i + 1\}. \end{cases}$$

The idea of this dynamic programming is as follows. Consider that we want to select $k$ best items from $(a_i, \ldots, a_J)$. If we put $a_i$ into the solution, we need to further select $k - 1$ best items from $(a_{i+1}, \ldots, a_J)$. In this case, we have $F[i][k] = u(a_i)w(a_i) + (1 - u(a_i))F[i+1][k-1]$; Otherwise, if we do not put $a_i$ into the solution, we are just selecting $k$ best items from $(a_{i+1}, \ldots, a_J)$, and then $F[i][k] = F[i+1][k]$. After computing this dynamic programming, we output $S[1][m]$ and $F[1][m]$ as the optimal solution and optimal value of Eq. (2) (lines 9-20). Below we formally state the correctness of BestPerm.

**Lemma 2** (Correctness of Oracle BestPerm). *Given any $u : A^{\text{ground}} \to [0, 1]$ and $w : A^{\text{ground}} \to \mathbb{R}$, the permutation $S^{\text{best}}$ returned by algorithm* BestPerm *satisfies*

$$f(S^{\text{best}}, u, w) = \max_{A \in \mathcal{A}} f(A, u, w).$$

BestPerm achieves $O(Nm + N \log(N))$ computation complexity. This is dramatically better than the computation complexity of the naive exhaustive search, which is $O(|\mathcal{A}|) = O(N^m)$.

---

**Algorithm 2:** CascadingVI

---

**Input:** $\delta$, $\delta' := \frac{\delta}{14}$, $L := \log(\frac{KHSN}{\delta'})$. For any $k > 0$ and $s \in \mathcal{S}$, $\bar{q}^k(s, a_\perp) = \underline{q}^k(s, a_\perp) := 1$
and $\bar{V}_{H+1}^k(s) = \underline{V}_{H+1}^k(s) := 0$. Initialize $n^{1,q}(s, a) = n^{1,p}(s, a) := 0$ for any
$(s, a) \in \mathcal{S} \times \mathcal{A}$.

1 **for** $k = 1, 2, \ldots, K$ **do**
2    **for** $h = H, H-1, \ldots, 1$ **do**
3      **for** $s \in \mathcal{S}$ **do**
4        **for** $a \in A^{\text{ground}} \setminus \{a_\perp\}$ **do**
5          $b^{k,q}(s, a) \leftarrow \min\{2\sqrt{\frac{\hat{q}^k(s,a)(1-\hat{q}^k(s,a))L}{n^{k,q}(s,a)}} + \frac{5L}{n^{k,q}(s,a)}, 1\}$;
6          $\bar{q}^k(s, a) \leftarrow \hat{q}^k(s, a) + b^{k,q}(s, a)$. $\underline{q}^k(s, a) \leftarrow \hat{q}^k(s, a) - b^{k,q}(s, a)$;
7        **for** $a \in A^{\text{ground}}$ **do**
8          $b^{k,pV}(s, a) \leftarrow \min\{2\sqrt{\frac{\text{Var}_{s' \sim \hat{p}^k}(\bar{V}_{h+1}^k(s'))L}{n^{k,p}(s,a)}} +$
$2\sqrt{\frac{\mathbb{E}_{s' \sim \hat{p}^k}[(\bar{V}_{h+1}^k(s') - \underline{V}_{h+1}^k(s'))^2]L}{n^{k,p}(s,a)}} + \frac{5HL}{n^{k,p}(s,a)}, H\}$;
9          $\bar{w}^k(s, a) \leftarrow r(s, a) + \hat{p}^k(\cdot|s, a)^\top \bar{V}_{h+1}^k + b^{k,pV}(s, a)$;
10      $\bar{V}_h^k(s), \pi_h^k(s) \leftarrow \texttt{BestPerm}(A^{\text{ground}}, \bar{q}^k(s, \cdot), \bar{w}^k(s, \cdot))$;
11      $\bar{V}_h^k(s) \leftarrow \min\{\bar{V}_h^k(s), H\}$. $A' \leftarrow \pi_h^k(s)$;
12      $\underline{V}_h^k(s) \leftarrow \max\{\sum_{i=1}^{|A'|} \prod_{j=1}^{i-1}(1 - \bar{q}^k(s, A'(j)))\underline{q}^k(s, A'(i))(r(s, A'(i)) +$
$\hat{p}^k(\cdot|s, A'(i))^\top \underline{V}_{h+1}^k - b^{k,pV}(s, A'(i))), 0\}$;
13    **for** $h = 1, 2, \ldots, H$ **do**
14      Observe the current state $s_h^k$;      // Take policy $\pi^k$ and observe the trajectory
15      Take action $A_h^k = \pi_h^k(s_h^k)$. $i \leftarrow 1$;
16      **while** $i \leq m$ **do**
17        Observe if $A_h^k(i)$ is clicked or not. Update $n^{k,q}(s_h^k, A_h^k(i))$ and $\hat{q}^k(s_h^k, A_h^k(i))$;
18        **if** $A_h^k(i)$ is clicked **then**
19          Receive reward $r(s_h^k, A_h^k(i))$, and transition to a next state
$s_{h+1}^k \sim p(\cdot|s_h^k, A_h^k(i))$;
20          $I_{k,h} \leftarrow i$. Update $n^{k,p}(s_h^k, A_h^k(i))$ and $\hat{p}^k(\cdot|s_h^k, A_h^k(i))$;
21          **break while**;      // Skip subsequent items
22        **else**
23          $i \leftarrow i + 1$;
24      **if** $i = m + 1$ **then**
25        Transition to a next state $s_{h+1}^k \sim p(\cdot|s_h^k, a_\perp)$;      // No item was clicked
26        Update $n^{k,p}(s_h^k, a_\perp)$ and $\hat{p}^k(\cdot|s_h^k, a_\perp)$;

---

## 5 REGRET MINIMIZATION FOR CASCADING RL

In this section, we study cascading RL with the regret minimization objective. Building upon oracle `BestPerm`, we propose an efficient algorithm `CascadingVI`, which is both computation and sample efficient and has a regret bound that nearly matches the lower bound.

### 5.1 ALGORITHM CascadingVI

Algorithm 2 describes `CascadingVI`. In each episode $k$, we construct the exploration bonus for the attraction probability $b^{k,q}$ and the exploration bonus for the expected future reward $b^{k,pV}$. Then, we calculate the optimistic attraction probability $\bar{q}^k(s, a)$ and weight $\bar{w}^k(s, a)$, by adding exploration bonuses for $q(s, a)$ and $p(\cdot|s, a)^\top V$ *individually* (lines 6 and 9). Here the weight $\bar{w}^k(s, a)$ represents an optimistic estimate of the expected cumulative reward that can be obtained if item $a$ is clicked in state $s$. Then, utilizing the monotonicity property of $f(A, u, w)$ with respect to the attraction

probability $u$ and weight $w$, we invoke oracle `BestPerm` with $\bar{q}^k(s, \cdot)$ and $\bar{w}^k(s, \cdot)$ to efficiently compute the optimistic value function $\bar{V}_h^k(s)$ and its associated greedy policy $\pi_h^k(s)$ (line 10).

After obtaining policy $\pi^k$, we play episode $k$ with $\pi^k$ (lines 14-26). At each step $h \in [H]$, we first observe the current state $s_h^k$, and select the item list $A_h^k = \pi_h^k(s)$. Then, the environment (user) examines the items in $A_h^k$ and clicks the first attractive item. Once an item $A_h^k(i)$ is clicked, the agent receives a reward and transitions to $s_{h+1}^k \sim p(\cdot|s_h^k, A_h^k(i))$, skipping the subsequent items. If no item in $A_h^k$ is clicked, the agent receives zero reward and transitions to $s_{h+1}^k \sim p(\cdot|s_h^k, a_\perp)$. In line 17, we increment the number of attraction observations $n^{k,q}(s_h^k, A_h^k(i))$ by 1, and update the empirical attraction probability $\hat{q}^k(s_h^k, A_h^k(i))$. In lines 20 and 26, we increment the number of transition observations $n^{k,p}(s_h^k, A_h^k(i))$ (or $n^{k,p}(s_h^k, a_\perp)$) by 1, and update the empirical transition distribution $\hat{p}^k(\cdot|s_h^k, A_h^k(i))$ (or $\hat{p}^k(\cdot|s_h^k, a_\perp)$) by incrementing the number of transitions to $s_{h+1}^k$ by 1 and keeping the number of transitions to other states unchanged.

If one naively applies classical RL algorithms (Azar et al., 2012; Zanette & Brunskill, 2019) by treating each $A \in \mathcal{A}$ as an ordinary action, one will suffer a dependency on $|\mathcal{A}|$ in the results. By contrast, `CascadingVI` only maintains the estimates of attraction and transition probabilities for each $(s, a)$, and employs oracle `BestPerm` to directly compute $\bar{V}_h^k(s)$, without enumerating over $A \in \mathcal{A}$. Therefore, `CascadingVI` avoids the dependency on $|A|$ in computation and statistical complexities.

## 5.2 Theoretical Guarantee of Algorithm `CascadingVI`

In regret analysis, we need to tackle several challenges: (i) how to guarantee the optimism of the output value by oracle `BestPerm` with the optimistic estimated inputs, and (ii) how to achieve the optimal regret bound when the problem degenerates to cascading bandits (Kveton et al., 2015; Vial et al., 2022). To handle these challenges, we prove the monotonicity property of $f(A, u, w)$ with respect to the attraction probability $u$ and weight $w$, leveraging the fact that the items in the optimal permutation are ranked in descending order of $w$. This monotonicity property ensures the optimism of the value function $\bar{V}_h^k(s)$. In addition, we employ the variance-awareness of the exploration bonus $b^{k,q}(s, a)$, scaling as $\hat{q}^k(s, a)(1 - \hat{q}^k(s, a))$, to save a factor of $\sqrt{m}$ in the regret bound. This enables our result to match the optimal result for cascading bandits (Vial et al., 2022) in the degenerated case.

**Theorem 1** (Regret Upper Bound). *With probability at least $1 - \delta$, the regret of algorithm* `CascadingVI` *is bounded by*

$$\tilde{O}\left(H\sqrt{HSNK}\right).$$

From Theorem 1, we see that the regret of `CascadingVI` depends only on the number of items $N$, instead of the number of item lists $|\mathcal{A}| = O(N^m)$. This demonstrates the efficiency of our estimation scheme and exploration bonus design. When our problem degenerates to cascading bandits, i.e., $S = H = 1$, our regret bound matches the optimal result for cascading bandits (Vial et al., 2022).

**Lower bound and optimality.** Recall from (Jaksch et al., 2010; Osband & Van Roy, 2016) that the lower bound for classic RL is $\Omega(H\sqrt{SNK})$. Since cascading RL reduces to classic RL when $q(s, a) = 1$ for any $(s, a) \in \mathcal{S} \times A^{\text{ground}}$ (i.e., the user always clicks the first item), the lower bound $\Omega(H\sqrt{SNK})$ also holds for cascading RL.

Our regret bound matches the lower bound up to a factor of $\sqrt{H}$ when ignoring logarithmic factors. Regarding this gap, one possibility is that it comes from our upper bound analysis. Specifically, we add exploration bonuses for $q$ and $p^\top V$ individually, which leads to a term $(q + b^{k,q})(\hat{p}^\top \bar{V} + b^{k,pV} - p^\top V)$ in regret decomposition. Here the term $b^{k,q}(\hat{p}^\top \bar{V} + b^{k,pV} - p^\top V) = \tilde{O}(Hb^{k,q})$ leads to a $\tilde{O}(H\sqrt{HK})$ regret, which causes the extra $\sqrt{H}$ gap. A straightforward idea to avoid this is to regard $q$ and $p$ as an integrated transition distribution of $(s, A)$, and construct an overall exploration bonus for $(s, A)$. However, this strategy forces us to maintain the estimates for all $A \in \mathcal{A}$, and will incur a dependency on $|\mathcal{A}|$ in computation and statistical complexities. Another possibility behind the gap $\sqrt{H}$ is that it is needed in a tight lower bound for any polynomial-time algorithm for cascading RL. Therefore, how to close the gap $\sqrt{H}$ while maintaining the computational efficiency remains open for future work.

Figure 1: Experiments for cascading RL on real-world data.

# 6 BEST POLICY IDENTIFICATION FOR CASCADING RL

Now we turn to cascading RL with best policy identification. We propose an efficient algorithm `CascadingBPI` to find an $\varepsilon$-optimal policy. Here we mainly introduce the idea of `CascadingBPI`, and defer the detailed pseudo-code and description to Appendix D.1 due to space limit.

In `CascadingBPI`, we optimistically estimate $q(s, a)$ and $p(\cdot|s, a)^\top V$, and add exploration bonuses for them individually. Then, we call the oracle `BestPerm` to compute the optimistic value function and hypothesized optimal policy. Furthermore, we construct an estimation error to bound the deviation between the optimistic value function and true value function. If this estimation error shrinks within $\varepsilon$, we simply output the hypothesized optimal policy; Otherwise, we play an episode with this policy. In the following, we provide the sample complexity of `CascadingBPI`.

**Theorem 2.** *With probability at least* $1 - \delta$, *algorithm* `CascadingBPI` *returns an $\varepsilon$-optimal policy, and the number of episodes used is bounded by* $\tilde{O}(\frac{H^3 SN}{\varepsilon^2} \log(\frac{1}{\delta}) + \frac{H^2 \sqrt{H} SN}{\varepsilon \sqrt{\varepsilon}}(\log(\frac{1}{\delta}) + S))$, *where* $\tilde{O}(\cdot)$ *hides logarithmic factors with respect to* $N, H, S$ *and* $\varepsilon$.

Theorem 2 reveals that the sample complexity of `CascadingBPI` is polynomial in problem parameters $N, H, S$ and $\varepsilon$, and independent of $|\mathcal{A}|$. Regarding the optimality, since cascading RL reduces to classic RL when $q(s, a) = 1$ for all $(s, a) \in \mathcal{S} \times A^{\text{ground}}$, existing lower bound for classic best policy identification $\Omega(\frac{H^2 SN}{\varepsilon^2} \log(\frac{1}{\delta}))$ (Dann & Brunskill, 2015) also applies here. This corroborates that `CascadingBPI` is near-optimal up to a factor of $H$ when $\varepsilon < \frac{H}{S^2}$.

# 7 EXPERIMENTS

In this section, we present experimental results on a real-world dataset MovieLens (Harper & Konstan, 2015), which contains millions of ratings for movies by users. We set $\delta = 0.005$, $K = 100000$, $H = 3$, $m = 3$, $S = 20$, $N \in \{10, 15, 20, 25\}$ and $|\mathcal{A}| \in \{820, 2955, 7240, 14425\}$. We defer the detailed setup and more results to Appendix A.

We compare our algorithm `CascadingVI` with three baselines, i.e., `CascadingVI-Oracle`, `CascadingVI-Bonus` and `AdaptVI`. Specifically, `CascadingVI-Oracle` replaces the efficient oracle `BestPerm` by a naive exhaustive search. `CascadingVI-Bonus` replaces the variance-aware exploration bonus $b^{k,q}$ by a variance-unaware bonus. `AdaptVI` adapts the classic RL algorithm (Zanette & Brunskill, 2019) to the combinatorial action space, which maintains the estimates for all $(s, A)$. As shown in Figure 1, our algorithm `CascadingVI` achieves the lowest regret and a fast running time. `CascadingVI-Oracle` has a comparative regret performance to `CascadingVI`, but suffers a much higher running time, which demonstrates the power of our oracle `BestPerm` in computation. `CascadingVI-Bonus` attains a similar running time as `CascadingVI`, but has a worse regret. This corroborates the effectiveness of our variance-aware exploration bonus in enhancing sample efficiency. `AdaptVI` suffers a very high regret and running time, since it learns the information of all permutations independently and its statistical and computational complexities depend on $|\mathcal{A}|$.

# 8 CONCLUSION

In this work, we formulate a cascading RL framework, which generalizes the cascading bandit model to characterize the impacts of user states and state transition in applications such as recommendation systems. We design a novel oracle `BestPerm` to efficiently identify the optimal item list under combinatorial action spaces. Building upon this oracle, we develop efficient algorithms `CascadingVI` and `CascadingBPI` with near-optimal regret and sample complexity guarantees.

## ACKNOWLEDGEMENT

The work of Yihan Du and R. Srikant is supported in part by AFOSR Grant FA9550-24-1-0002, ONR Grant N00014-19-1-2566, and NSF Grants CNS 23-12714, CNS 21-06801, CCF 19-34986, and CCF 22-07547.

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

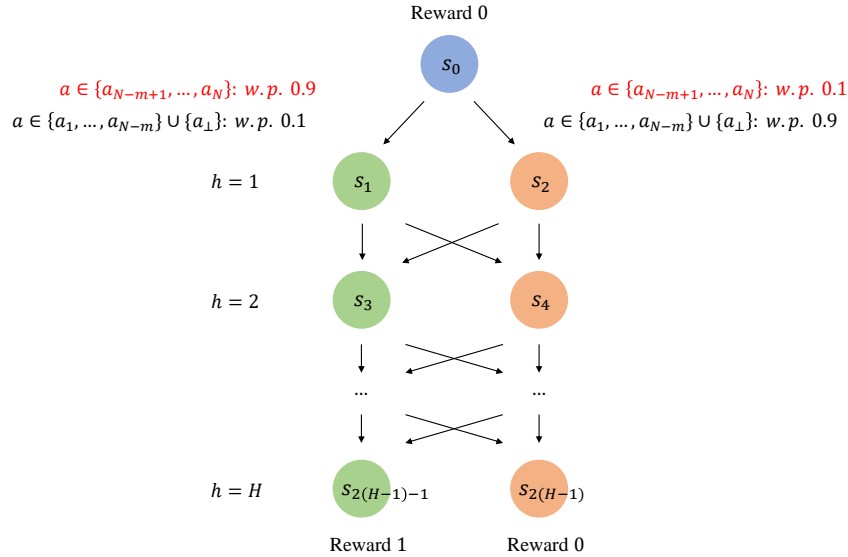

Figure 2: The constructed cascading MDP in synthetic data.

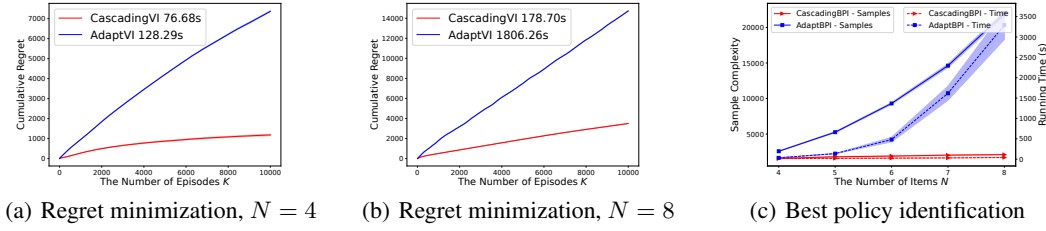

(a) Regret minimization, $N = 4$    (b) Regret minimization, $N = 8$    (c) Best policy identification

Figure 3: Experiments for cascading RL on synthetic data.

# APPENDIX

## A   MORE EXPERIMENTS

In this section, we describe the setup for the experiments on real-world data (Figure 1), and present more experimental results on synthetic data (Figure 3).

### A.1   EXPERIMENTAL SETUP WITH REAL-WORLD DATA

In our experiments in Figure 1, we consider the real-world dataset MovieLens (Harper & Konstan, 2015), which is also used in prior cascading bandit works (Zong et al., 2016; Vial et al., 2022). This dataset contains 25 million ratings on a 5-star scale for 62000 movies by 162000 users. We regard each user as a state, and each movie as an item. For each user-movie pair, we scale the rating to $[0, 1]$ and regard it as the attraction probability. The reward of each user-movie pair is set to 1. For each user-movie pair which has a rating no lower than $4.5$ stars, we set the transition probability to this state (user) itself as $0.9$, and that to other states (users) as $\frac{0.9}{S-1}$. For each user-movie pair which has a rating lower than $4.5$ stars, we set the transition probability to all states (users) as $\frac{1}{S}$. We use a subset of data from MovieLens, and set $\delta = 0.005$, $K = 100000$, $H = 3$, $m = 3$, $S = 20$, $N \in \{10, 15, 20, 25\}$ and $|\mathcal{A}| = \sum_{\tilde{m}=1}^{m} \binom{N}{\tilde{m}} \tilde{m}! \in \{820, 2955, 7240, 14425\}$.

A.2 Experiments on Synthetic Data

For synthetic data, we consider a cascading MDP with $H$ layers, $S = 2H - 1$ states and $N$ items as shown in Figure 2: There is only an initial state $s_0$ in the first layer. For any $2 \leq h \leq H$, there are a good state $s_{2(h-1)-1}$ and a bad state $s_{2(h-1)}$ in layer $h$. The reward function depends only on states. All good states induce reward 1, and all bad states and the initial state give reward 0. The attraction probability for all state-item pairs is $\frac{1}{2}$. Denote $A^{\text{ground}} = \{a_1, \ldots, a_N\}$. For any $h \in [H]$, under a good item $a \in \{a_{N-m+1}, \ldots, a_N\}$, the transition probabilities from each state in layer $h$ to the good state and the bad state in layer $h + 1$ are 0.9 and 0.1, respectively. On the contrary, under a bad item $a \in \{a_1, \ldots, a_{N-m}\} \cup \{a_\perp\}$, the transition probabilities from each state in layer $h$ to the good state and the bad state in layer $h + 1$ are 0.1 and 0.9, respectively. Therefore, in this MDP, an optimal policy is to select good items $(a_{N-m+1}, \ldots, a_N)$ in all states.

We set $\delta = 0.005$, $H = 5$, $S = 9$ and $m = 3$. Each algorithm is performed for 20 independent runs. In the regret minimization setting, we let $N \in \{4, 8\}$ and $K = 10000$, and show the average cumulative regrets and average running times (in the legend) across runs. In the best policy identification setting, we set $\epsilon = 0.5$ and $N \in \{4, 5, 6, 7, 8\}$, and plot the average sample complexities and average running times across runs with $95\%$ confidence intervals.

Under the regret minimization objective, we compare our algorithm `CascadingVI` with `AdaptVI`. From Figures 3(a) and 3(b), one can see that `CascadingVI` achieves significantly lower regret and running time than `AdaptVI`, and this advantage becomes more clear as $N$ increases. This result demonstrates the efficiency of our computation oracle and estimation scheme.

Regarding the best policy identification objective, we compare our algorithm `CascadingBPI` with `AdaptBPI`, an adaptation of a classic best policy identification algorithm in (Ménard et al., 2021) to combinatorial actions. In Figure 3(c), as $N$ increases, the sample complexity and running time of `AdaptBPI` increase exponentially fast. By contrast, `CascadingBPI` has much lower sample complexity and running time, and enjoys a mild growth rate as $N$ increases. This matches our theoretical result that the sample and computation complexities of `CascadingBPI` are polynomial in $N$.

## B  Proofs for Oracle `BestPerm`

In this section, we present the proofs for oracle `BestPerm`.

First, we introduce two important lemmas which are used in the proof of Lemma 1.

**Lemma 3** (Interchange by Descending Weights). *For any* $u : A^{\text{ground}} \mapsto [0, 1]$, $w : A^{\text{ground}} \mapsto \mathbb{R}$ *and* $A = (a_1, \ldots, a_\ell, a_{\ell+1}, \ldots, a_n)$ *such that* $1 \leq \ell < n$ *and* $w(a_\ell) < w(a_{\ell+1})$, *denoting* $A' := (a_1, \ldots, a_{\ell+1}, a_\ell, \ldots, a_n)$, *we have*

$$f(A, u, w) \leq f(A', u, w).$$

*Proof of Lemma 3.* This proof uses a similar idea as the interchange argument (Bertsekas & Castanon, 1999; Ross, 2014). We have

$$f(A, u, w) - f(A', u, w)$$

$$= \prod_{j=1}^{\ell-1}(1 - u(a_j)) \cdot (u(a_\ell)w(a_\ell) + (1 - u(a_\ell))u(a_{\ell+1})w(a_{\ell+1}))$$

$$\quad - \prod_{j=1}^{\ell-1}(1 - u(a_j)) \cdot (u(a_{\ell+1})w(a_{\ell+1}) + (1 - u(a_{\ell+1}))u(a_\ell)w(a_\ell))$$

$$= \prod_{j=1}^{\ell-1}(1 - u(a_j)) \cdot (u(a_{\ell+1})u(a_\ell)w(a_\ell) - u(a_\ell)u(a_{\ell+1})w(a_{\ell+1}))$$

$$= \prod_{j=1}^{\ell-1}(1 - u(a_j)) \cdot u(a_\ell)u(a_{\ell+1}) \cdot (w(a_\ell) - w(a_{\ell+1}))$$

$$\leq 0.$$

□

**Lemma 4** (Items behind $a_\perp$ Do Not Matter). *For any ordered subsets of $\{a_1, \ldots, a_N\}$, $A$ and $A'$, such that $A \cap A' = \emptyset$, we have*

$$f((A, a_\perp, A'), u, w) = f((A, a_\perp), u, w).$$

*Proof.* Since $u(a_\perp) = 1$, we have

$$
\begin{aligned}
&f((A, a_\perp, A'), u, w) \\
&= \sum_{i=1}^{|A|} \prod_{j=1}^{i-1} \Big(1 - u(A(j))\Big) u(A(i)) w(A(i)) + \prod_{j=1}^{|A|} \Big(1 - u(A(j))\Big) u(a_\perp) w(a_\perp) \\
&\quad + \sum_{i=1}^{|A'|} \Big( \prod_{\ell=1}^{|A|} \Big(1 - u(A(\ell))\Big) \Big(1 - u(a_\perp)\Big) \prod_{j=1}^{i-1} \Big(1 - u(A'(j))\Big) \Big) u(A'(i)) w(A'(i)) \\
&= \sum_{i=1}^{|A|} \prod_{j=1}^{i-1} \Big(1 - u(A(j))\Big) u(A(i)) w(A(i)) + \prod_{j=1}^{|A|} \Big(1 - u(A(j))\Big) u(a_\perp) w(a_\perp) \\
&= f((A, a_\perp), u, w).
\end{aligned}
$$

□

Now we prove Lemma 1.

For any $X \subseteq A^{\text{ground}}$, let $\text{Perm}(X)$ denote the collection of permutations of the items in $X$, and $\text{DesW}(X) \in \text{Perm}(X)$ denote the permutation where items are sorted in descending order of $w$.

*Proof of Lemma 1.* First, we prove property (i) by contradiction.

Suppose that the best permutation $A^* = \text{argmax}_{A \in \text{Perm}(X)} f(A, u, w)$ does not rank items in descending order of $w$. In other words, there exist some $a_\ell, a_{\ell+1}$ such that we can write $A^* = (a_1, \ldots, a_\ell, a_{\ell+1}, \ldots, a_{|X|})$ and $w(a_\ell) < w(a_{\ell+1})$.

Then, using Lemma 3, we have that $A' = (a_1, \ldots, a_{\ell+1}, a_\ell, \ldots, a_{|X|})$ satisfies $f(A', u, w) \geq f(A^*, u, w)$, which contradicts the supposition. Given any permutation in $\text{Perm}(X)$, we can repeatedly perform Lemma 3 to obtain a better permutation as bubble sort, until all items are ranked in descending order of $w$. Therefore, we obtain property (i).

Next, we prove property (ii).

For any $u, w$ and disjoint $X, X' \subseteq A^{\text{ground}} \setminus \{a_\perp\}$ such that $w(a) > w(a_\perp)$ for any $a \in X, X'$, we have

$$f((\text{DesW}(X), a_\perp), u, w) \overset{\text{(a)}}{=} f((\text{DesW}(X), a_\perp, X'), u, w)$$
$$\overset{\text{(b)}}{\leq} f((\text{DesW}(X \cup X'), a_\perp), u, w),$$

Here in the right-hand side of equality (a), $X'$ can be in any order, and inequality (b) uses property (i).

Furthermore, for any $u, w$ and disjoint $X, X' \subseteq A^{\text{ground}} \setminus \{a_\perp\}$ such that $w(a) > w(a_\perp)$ for any $a \in X$, and $w(a) < w(a_\perp)$ for any $a \in X'$, we have

$$f((\text{DesW}(X, X'), a_\perp), u, w) \overset{\text{(c)}}{\leq} f((\text{DesW}(X), a_\perp, \text{DesW}(X')), u, w)$$
$$= f((\text{DesW}(X), a_\perp), u, w),$$

where inequality (c) is due to property (i).

□

Next, we prove Lemma 2.

*Proof of Lemma 2.* From Lemma 1 (i), we have that when fixing a subset of $A^{\text{ground}}$, the best order of this subset is to rank items in descending order of $w$. Thus, the problem of finding the best permutation in Eq. (2) reduces to finding the best subset, and then we can just sort items in this subset by descending $w$ to obtain the solution.

We sort the items in $A^{\text{ground}}$ in descending order of $w$, and denote the sorted sequence by $a_1, \ldots, a_J, a_\perp, a_{J+1}, \ldots, a_N$. Here $J$ denotes the number of items with weights above $a_\perp$.

According to Lemma 1 (ii), we have that the best permutation only consists of items in $a_1, \ldots, a_J$. In other words, we should discard $a_{J+1}, \ldots, a_N$.

**Case (i).** If $1 \leq J \leq m$, $(a_1, \ldots, a_J)$ satisfies the cardinality constraint and is the best permutation.

**Case (ii).** Otherwise, if $J = 0$, we have to select a single best item to satisfy that there is at least one regular item in the solution.

Why do not we select more items? We can prove that including more items gives a worse permutation by contradiction. Without loss of generality, consider a permutation with an additional item, i.e., $(a, a', a_\perp)$, where $a, a' \in A^{\text{ground}} \setminus \{a_\perp\}$ and $w(a), w(a') < w(a_\perp)$. Using Lemma 3, we have

$$f((a, a', a_\perp), u, w) \leq f((a, a_\perp, a'), u, w)$$
$$= f((a, a_\perp), u, w).$$

In this case, the best permutation is the best single item $\arg\max_{a \in \{a_1, \ldots, a_N\}} f((a, a_\perp), u, w)$.

**Case (iii).** If $J > m$, the problem reduces to selecting $m$ best items from $a_1, \ldots, a_J$. For any $i \in [J]$ and $k \in [\min\{m, J - i + 1\}]$, let $F[i][k]$ denote the optimal value of the problem $\max_{(a'_1, \ldots, a'_k) \subseteq (a_i, \ldots, a_J)} f((a'_1, \ldots, a'_k), u, w)$. From the structure of $f(A, u, w)$, we have the following dynamic programming:

$$\begin{cases} F[J][1] = u(a_J)w(a_J) + (1 - u(a_J))w(a_\perp), \\ F[i][0] = w(a_\perp), & 1 \leq i \leq J, \\ F[i][k] = -\infty, & 1 \leq i \leq J, \ J - i + 1 < k \leq m, \\ F[i][k] = \max\{F[i+1][k], \\ \quad u(a_i)w(a_i) + (1 - u(a_i))F[i+1][k-1]\}, & 1 \leq i \leq J - 1, \ 1 \leq k \leq \min\{m, J - i + 1\}. \end{cases}$$

Then, $F[1][m]$ gives the objective value of the best permutation.

Combining the above analysis, we have that `BestPerm` returns the best permutation, i.e., $f(S^{\text{best}}, u, w) = \max_{A \in \mathcal{A}} f(A, u, w)$. □

## C PROOFS FOR CASCADING RL WITH REGRET MINIMIZATION

In this section, we provide the proofs for cascading RL with the regret minimization objective.

### C.1 VALUE DIFFERENCE LEMMA FOR CASCADING MDP

We first give the value difference lemma for cascading MDP, which is useful for regret decomposition.

**Lemma 5** (Cascading Value Difference Lemma). *For any two cascading MDPs* $\mathcal{M}'(\mathcal{S}, A^{\text{ground}}, \mathcal{A}, m, H, q', p', r')$ *and* $\mathcal{M}''(\mathcal{S}, A^{\text{ground}}, \mathcal{A}, m, H, q'', p'', r'')$, *the difference in values under the same policy* $\pi$ *satisfies*

$$V_h'^\pi(s) - V_h''^\pi(s) = \sum_{t=h}^{H} \mathbb{E}_{q'', p'', \pi} \left[ \sum_{i=1}^{|A_t|} \prod_{j=1}^{i-1} (1 - q'(s_t, A_t(j)))q'(s_t, A_t(i))r'(s_t, A_t(i)) \right.$$

$$\left. - \sum_{i=1}^{|A_t|} \prod_{j=1}^{i-1} (1 - q''(s_t, A_t(j)))q''(s_t, A_t(i))r''(s_t, A_t(i)) \right.$$

$$+ \left( \sum_{i=1}^{|A_t|} \prod_{j=1}^{i-1}(1 - q'(s_t, A_t(j)))q'(s_t, A_t(i))p'(\cdot|s_t, A_t(i)) \right.$$

$$\left. - \sum_{i=1}^{|A_t|} \prod_{j=1}^{i-1}(1 - q''(s_t, A_t(j)))q''(s_t, A_t(i))p''(\cdot|s_t, A_t(i)) \right)^\top V'^\pi_{t+1} \Bigg| s_h = s, A_h = \pi_h(s_h) \Bigg].$$

*Proof of Lemma 5.* Let $A := \pi_h(s)$. We have

$$V'^\pi_h(s) - V''^\pi_h(s)$$

$$= \sum_{i=1}^{|A|} \prod_{j=1}^{i-1}(1 - q'(s, A(j)))q'(s, A(i)) \left( r'(s, A(i)) + p'(\cdot|s, A(i))^\top V'^\pi_{h+1} \right)$$

$$- \sum_{i=1}^{|A|} \prod_{j=1}^{i-1}(1 - q''(s, A(j)))q''(s, A(i)) \left( r''(s, A(i)) + p''(\cdot|s, A(i))^\top V''^\pi_{h+1} \right)$$

$$= \sum_{i=1}^{|A|} \prod_{j=1}^{i-1}(1 - q'(s, A(j)))q'(s, A(i))r'(s, A(i)) - \sum_{i=1}^{|A|} \prod_{j=1}^{i-1}(1 - q''(s, A(j)))q''(s, A(i))r''(s, A(i))$$

$$+ \left( \sum_{i=1}^{|A|} \prod_{j=1}^{i-1}(1 - q'(s, A(j)))q'(s, A(i))p'(\cdot|s, A(i)) \right.$$

$$\left. - \sum_{i=1}^{|A|} \prod_{j=1}^{i-1}(1 - q''(s, A(j)))q''(s, A(i))p''(\cdot|s, A(i)) \right)^\top V'^\pi_{h+1}$$

$$+ \sum_{i=1}^{|A|} \prod_{j=1}^{i-1}(1 - q''(s, A(j)))q''(s, A(i))p''(\cdot|s, A(i))^\top \left( V'^\pi_{h+1} - V''^\pi_{h+1} \right)$$

$$= \sum_{i=1}^{|A|} \prod_{j=1}^{i-1}(1 - q'(s, A(j)))q'(s, A(i))r'(s, A(i)) - \sum_{i=1}^{|A|} \prod_{j=1}^{i-1}(1 - q''(s, A(j)))q''(s, A(i))r''(s, A(i))$$

$$+ \left( \sum_{i=1}^{|A|} \prod_{j=1}^{i-1}(1 - q'(s, A(j)))q'(s, A(i))p'(\cdot|s, A(i)) \right.$$

$$\left. - \sum_{i=1}^{|A|} \prod_{j=1}^{i-1}(1 - q''(s, A(j)))q''(s, A(i))p''(\cdot|s, A(i)) \right)^\top V'^\pi_{h+1}$$

$$+ \mathbb{E}_{q'',p'',\pi} \left[ V'^\pi_{h+1}(s_{h+1}) - V''^\pi_{h+1}(s_{h+1})|s_h = s, \pi \right]$$

$$= \mathbb{E}_{q'',p'',\pi} \left[ \sum_{t=h}^{H} \left( \sum_{i=1}^{|A_t|} \prod_{j=1}^{i-1}(1 - q'(s_t, A_t(j)))q'(s_t, A_t(i))r'(s_t, A_t(i)) \right. \right.$$

$$- \sum_{i=1}^{|A_t|1} \prod_{j=1}^{i-1}(1 - q''(s_t, A_t(j)))q''(s_t, A_t(i))r''(s_t, A_t(i))$$

$$+ \left( \sum_{i=1}^{|A_t|} \prod_{j=1}^{i-1}(1 - q'(s_t, A_t(j)))q'(s_t, A_t(i))p'(\cdot|s_t, A_t(i)) \right.$$

$$\left. \left. \left. - \sum_{i=1}^{|A_t|} \prod_{j=1}^{i-1}(1 - q''(s_t, A_t(j)))q''(s_t, A_t(i))p''(\cdot|s_t, A_t(i)) \right)^\top V'^\pi_{t+1} \right) \Bigg| s_h = s, A_h = \pi_h(s_h) \right].$$

$\square$

## C.2 Regret Upper Bound for algorithm `CascadingVI`

Below we prove the regret upper bound for algorithm `CascadingVI`.

### C.2.1 Concentration

For any $k > 0$, $s \in \mathcal{S}$ and $a \in A^{\text{ground}} \setminus \{a_\perp\}$, let $n^{k,q}(s,a)$ denote the number of times that the attraction of $(s,a)$ is observed up to episode $k$. In addition, for any $k > 0$, $s \in \mathcal{S}$ and $a \in A^{\text{ground}}$, let $n^{k,p}(s,a)$ denote the number of times that the transition of $(s,a)$ is observed up to episode $k$.

Let event

$$
\mathcal{E} := \left\{ \left| \hat{q}^k(s,a) - q(s,a) \right| \le 2\sqrt{\frac{\hat{q}^k(s,a)(1 - \hat{q}^k(s,a)) \log\left(\frac{KHSA}{\delta'}\right)}{n^{k,q}(s,a)}} + \frac{5 \log\left(\frac{KHSA}{\delta'}\right)}{n^{k,q}(s,a)}, \right.
$$
$$
\left| \sqrt{\hat{q}^k(s,a)(1 - \hat{q}^k(s,a))} - \sqrt{q(s,a)(1 - q(s,a))} \right| \le 2\sqrt{\frac{\log\left(\frac{KHSA}{\delta'}\right)}{n^{k,q}(s,a)}},
$$
$$
\left. \forall k \in [K], \forall (s,a) \in \mathcal{S} \times (A^{\text{ground}} \setminus \{a_\perp\}) \right\}.
$$

**Lemma 6** (Concentration of Attractive Probability). *It holds that*
$$
\Pr[\mathcal{E}] \ge 1 - 4\delta'.
$$

*Proof of Lemma 6.* Using Bernstern's inequality and a union bound over $n^{k,q}(s,a) \in [KH]$, $k \in [K]$ and $(s,a) \in \mathcal{S} \times A^{\text{ground}}$, we have that with probability $1 - 2\delta'$,

$$
\left| \hat{q}^k(s,a) - q(s,a) \right| \le 2\sqrt{\frac{q(s,a)(1 - q(s,a)) \log\left(\frac{KHSA}{\delta'}\right)}{n^{k,q}(s,a)}} + \frac{\log\left(\frac{KHSA}{\delta'}\right)}{n^{k,q}(s,a)}. \tag{4}
$$

Moreover, applying Lemma 1 in (Zanette & Brunskill, 2019), we have that with probability $1 - 2\delta'$,

$$
\left| \sqrt{\hat{q}^k(s,a)(1 - \hat{q}^k(s,a))} - \sqrt{q(s,a)(1 - q(s,a))} \right| \le 2\sqrt{\frac{\log\left(\frac{KHSA}{\delta'}\right)}{n^{k,q}(s,a)}}. \tag{5}
$$

Combining Eqs. (4) and (5), we obtain this lemma. $\qquad \square$

Let event

$$
\mathcal{F} := \left\{ \left| \left( \hat{p}^k(\cdot|s,a) - p(\cdot|s,a) \right)^\top V_{h+1}^* \right| \le 2\sqrt{\frac{\text{Var}_{s' \sim p}\left( V_{h+1}^*(s') \right) \log\left(\frac{KHSA}{\delta'}\right)}{n^{k,p}(s,a)}} + \frac{H \log\left(\frac{KHSA}{\delta'}\right)}{n^{k,p}(s,a)}, \right.
$$
$$
\tag{6}
$$
$$
\left| \hat{p}^k(s'|s,a) - p(s'|s,a) \right| \le \sqrt{\frac{p(s'|s,a)(1 - p(s'|s,a)) \log\left(\frac{KHSA}{\delta'}\right)}{n^{k,p}(s,a)}} + \frac{\log\left(\frac{KHSA}{\delta'}\right)}{n^{k,p}(s,a)},
$$
$$
\tag{7}
$$
$$
\left\| \hat{p}^k(\cdot|s,a) - p(\cdot|s,a) \right\|_1 \le \sqrt{\frac{2S \log\left(\frac{KHSA}{\delta'}\right)}{n^{k,p}(s,a)}}, \tag{8}
$$
$$
\left. \forall k \in [K], \forall h \in [H], \forall (s,a) \in \mathcal{S} \times A^{\text{ground}} \right\}.
$$

**Lemma 7** (Concentration of Transition Probability). *It holds that*
$$
\Pr[\mathcal{F}] \ge 1 - 6\delta'.
$$

*Proof of Lemma 7.* According to Bernstein's inequality and a union bound over $n^{k,p}(s,a) \in [KH]$, $k \in [K]$, $h \in [H]$ and $(s,a) \in \mathcal{S} \times A^{\text{ground}}$, we obtain Eqs. (6) and (7). In addition, Eq. (8) follows from (Weissman et al., 2003) and Eq. (55) in (Zanette & Brunskill, 2019) . $\qquad \square$

Let event

$$\mathcal{G} := \left\{ \left| \sqrt{\mathrm{Var}_{s' \sim \hat{p}^k} \left( V_{h+1}^*(s') \right)} - \sqrt{\mathrm{Var}_{s' \sim p} \left( V_{h+1}^*(s') \right)} \right| \leq 2H \sqrt{\frac{\log \left( \frac{KHSA}{\delta'} \right)}{n^{k,p}(s,a)}}, \right.$$

$$\left. \forall k \in [K], \forall h \in [H], \forall (s,a) \in \mathcal{S} \times A^{\mathrm{ground}} \right\}. \tag{9}$$

**Lemma 8** (Concentration of Variance). *It holds that*

$$\Pr[\mathcal{G}] \geq 1 - 2\delta'.$$

*Furthermore, assume event $\mathcal{F} \cap \mathcal{G}$ holds. Then, for any $k \in [K]$, $h \in [H]$ and $(s,a) \in \mathcal{S} \times A^{\mathrm{ground}}$, if $\bar{V}_{h+1}^k(s') \geq V_{h+1}^*(s') \geq \underline{V}_{h+1}^k(s')$ for any $s' \in \mathcal{S}$, we have*

$$\left| \left( \hat{p}^k(\cdot|s,a) - p(\cdot|s,a) \right)^\top V_{h+1}^* \right| \leq 2 \sqrt{\frac{\mathrm{Var}_{s' \sim \hat{p}^k} \left( \bar{V}_{h+1}^k(s') \right) \log \left( \frac{KHSA}{\delta'} \right)}{n^{k,p}(s,a)}}$$

$$+ 2 \sqrt{\frac{\mathbb{E}_{s' \sim \hat{p}^k} \left[ \left( \bar{V}_{h+1}^k(s') - \underline{V}_{h+1}^k(s') \right)^2 \right] \log \left( \frac{KHSA}{\delta'} \right)}{n^{k,p}(s,a)}} + \frac{5H \log \left( \frac{KHSA}{\delta'} \right)}{n^{k,p}(s,a)}.$$

*Proof of Lemma 8.* According to Eq. (53) in (Zanette & Brunskill, 2019), we have

$$\Pr[\mathcal{G}] \geq 1 - 2\delta'.$$

Moreover, assume event $\mathcal{F} \cap \mathcal{G}$ holds. Then, for any $k \in [K]$, $h \in [H]$ and $(s,a) \in \mathcal{S} \times A^{\mathrm{ground}}$, if $\bar{V}_{h+1}^k(s') \geq V_{h+1}^*(s') \geq \underline{V}_{h+1}^k(s')$ for any $s' \in \mathcal{S}$, we have

$$\left| \sqrt{\mathrm{Var}_{s' \sim \hat{p}^k} \left( \bar{V}_{h+1}^k(s') \right)} - \sqrt{\mathrm{Var}_{s' \sim p} \left( V_{h+1}^*(s') \right)} \right|$$

$$\leq \left| \sqrt{\mathrm{Var}_{s' \sim \hat{p}^k} \left( \bar{V}_{h+1}^k(s') \right)} - \sqrt{\mathrm{Var}_{s' \sim \hat{p}^k} \left( V_{h+1}^*(s') \right)} \right|$$

$$+ \left| \sqrt{\mathrm{Var}_{s' \sim \hat{p}^k} \left( V_{h+1}^*(s') \right)} - \sqrt{\mathrm{Var}_{s' \sim p} \left( V_{h+1}^*(s') \right)} \right|$$

$$\overset{(a)}{\leq} \sqrt{\mathbb{E}_{s' \sim \hat{p}^k} \left[ \left( \bar{V}_{h+1}^k(s') - \underline{V}_{h+1}^k(s') \right)^2 \right]} + 2H \sqrt{\frac{\log \left( \frac{KHSA}{\delta'} \right)}{n^{k,p}(s,a)}}, \tag{10}$$

where inequality (a) comes from Eqs. (48)-(52) in (Zanette & Brunskill, 2019).

Plugging Eq. (10) into Eq. (6), we have

$$\left| \left( \hat{p}^k(\cdot|s,a) - p(\cdot|s,a) \right)^\top V_{h+1}^* \right| \leq 2 \sqrt{\frac{\mathrm{Var}_{s' \sim \hat{p}^k} \left( \bar{V}_{h+1}^k(s') \right) \log \left( \frac{KHSA}{\delta'} \right)}{n^{k,p}(s,a)}}$$

$$+ 2 \sqrt{\frac{\mathbb{E}_{s' \sim \hat{p}^k} \left[ \left( \bar{V}_{h+1}^k(s') - \underline{V}_{h+1}^k(s') \right)^2 \right] \log \left( \frac{KHSA}{\delta'} \right)}{n^{k,p}(s,a)}} + \frac{5H \log \left( \frac{KHSA}{\delta'} \right)}{n^{k,p}(s,a)}.$$

$$\square$$

For any $k > 0$, $h \in [H]$, $i \in [m]$ and $(s,a) \in \mathcal{S} \times (A^{\mathrm{ground}} \setminus \{a_\perp\})$, let $v_{k,h,i}^{observe,q}(s,a)$ denote the probability that the attraction of $(s,a)$ is observed in the $i$-th position at step $h$ of episode $k$. Let $v_{k,h}^{observe,q}(s,a) := \sum_{i=1}^m v_{k,h,i}^{observe,q}(s,a)$ and $v_k^{observe,q}(s,a) := \sum_{h=1}^H \sum_{i=1}^m v_{k,h,i}^{observe,q}(s,a)$.

For any $k > 0$, $h \in [H]$, $i \in [m+1]$ and $(s,a) \in \mathcal{S} \times A^{\mathrm{ground}}$, let $v_{k,h,i}^{observe,p}(s,a)$ denote the probability that the transition of $(s,a)$ is observed (i.e., $(s,a)$ is clicked) in the $i$-th posi-

tion at step $h$ of episode $k$. Let $v_{k,h}^{observe,p}(s,a) := \sum_{i=1}^{m+1} v_{k,h,i}^{observe,p}(s,a)$ and $v_k^{observe,p}(s,a) := \sum_{h=1}^{H} \sum_{i=1}^{m+1} v_{k,h,i}^{observe,p}(s,a)$.

**Lemma 9.** *For any $k > 0$ and $h \in [H]$, we have*

$$\sum_{(s,a)\in\mathcal{S}\times A^{\text{ground}}\setminus\{a_\perp\}} v_{k,h}^{observe,q}(s,a)q(s,a) \le 1.$$

*Proof.* For any $k > 0$, $h \in [H]$, $s \in \mathcal{S}$ and $A \in \mathcal{A}$, let $w_{k,h}(s,A)$ denote the probability that $(s,A)$ is visited at step $h$ in episode $k$.

It holds that

$$\sum_{(s,a)\in\mathcal{S}\times A^{\text{ground}}\setminus\{a_\perp\}} v_{k,h}^{observe,q}(s,a)q(s,a)$$

$$= \sum_{(s,a)\in\mathcal{S}\times A^{\text{ground}}\setminus\{a_\perp\}} \sum_{i=1}^{m} v_{k,h,i}^{observe,q}(s,a)q(s,a)$$

$$= \sum_{(s,a)\in\mathcal{S}\times A^{\text{ground}}\setminus\{a_\perp\}} \sum_{i=1}^{m} \sum_{\substack{A\in\mathcal{A}\\A(i)=a}} w_{k,h}(s,A) \prod_{j=1}^{i-1}(1-q(s,A(j)))q(s,a)$$

$$= \sum_{(s,a)\in\mathcal{S}\times A^{\text{ground}}\setminus\{a_\perp\}} \sum_{i=1}^{m} \Pr[(s,a) \text{ is clicked in the } i\text{-th position at step } h \text{ of episode } k]$$

$$= \sum_{(s,a)\in\mathcal{S}\times A^{\text{ground}}\setminus\{a_\perp\}} \Pr[(s,a) \text{ is clicked at step } h \text{ of episode } k]$$

$$\le 1.$$

$\square$

Let event

$$\mathcal{K} := \left\{ n^{k,q}(s,a) \ge \frac{1}{2}\sum_{k'<k} v_{k'}^{observe,q}(s,a) - H\log\left(\frac{HSA}{\delta'}\right), \right.$$

$$\forall k \in [K], \forall (s,a) \in \mathcal{S}\times(A^{\text{ground}}\setminus\{a_\perp\}),$$

$$\left. n^{k,p}(s,a) \ge \frac{1}{2}\sum_{k'<k} v_{k'}^{observe,p}(s,a) - H\log\left(\frac{HSA}{\delta'}\right), \forall k \in [K], \forall (s,a) \in \mathcal{S}\times A^{\text{ground}} \right\}$$

**Lemma 10.** *It holds that*

$$\Pr[\mathcal{K}] \ge 1 - 2\delta'.$$

*Proof of Lemma 10.* This lemma can be obtained by Lemma F.4 in (Dann et al., 2017). $\square$

### C.2.2 VISITATION

For any $k > 0$, we define the following two sets:

$$B_k^q := \left\{ (s,a) \in \mathcal{S}\times(A^{\text{ground}}\setminus\{a_\perp\}) : \frac{1}{4}\sum_{k'<k} v_{k'}^{observe,q}(s,a) \ge H\log\left(\frac{HSN}{\delta'}\right) + H \right\},$$

$$B_k^p := \left\{ (s,a) \in \mathcal{S}\times A^{\text{ground}} : \frac{1}{4}\sum_{k'<k} v_{k'}^{observe,p}(s,a) \ge H\log\left(\frac{HSN}{\delta'}\right) + H \right\}.$$

$B_k^q$ and $B_k^p$ stand for the sets of state-item pairs whose attraction and transition are sufficiently observed in expectation up to episode $k$, respectively.

**Lemma 11** (Sufficient Visitation). *Assume that event $\mathcal{K}$ holds. Then, if $(s,a) \in B_k^q$, we have*

$$n^{k,q}(s,a) \geq \frac{1}{4} \sum_{k' \leq k} v_{k'}^{observe,q}(s,a).$$

*If $(s,a) \in B_k^p$, we have*

$$n^{k,p}(s,a) \geq \frac{1}{4} \sum_{k' \leq k} v_{k'}^{observe,p}(s,a).$$

*Proof of Lemma 11.* If $(s,a) \in B_k^q$, we have

$$
\begin{aligned}
n^{k,q}(s,a) &\geq \frac{1}{2} \sum_{k'<k} v_{k'}^{observe,q}(s,a) - H \log\left(\frac{HSA}{\delta'}\right) \\
&= \frac{1}{4} \sum_{k'<k} v_{k'}^{observe,q}(s,a) + \frac{1}{4} \sum_{k'<k} v_{k'}^{observe,q}(s,a) - H \log\left(\frac{HSA}{\delta'}\right) \\
&\overset{(a)}{\geq} \frac{1}{4} \sum_{k'<k} v_{k'}^{observe,q}(s,a) + H \\
&\geq \frac{1}{4} \sum_{k'<k} v_{k'}^{observe,q}(s,a) + v_k^{observe,q}(s,a) \\
&= \frac{1}{4} \sum_{k' \leq k} v_{k'}^{observe,q}(s,a),
\end{aligned}
$$

where inequality (a) is due to the definition of $B_k^q$.

By a similar analysis, we can also obtain the second inequality in this lemma. $\quad\square$

**Lemma 12** (Minimal Regret). *It holds that*

$$\sum_{k=1}^{K} \sum_{h=1}^{H} \sum_{(s,a) \notin B_k^q} v_{k,h}^{observe,q}(s,a) \leq 8HSA \log\left(\frac{HSN}{\delta'}\right),$$

$$\sum_{k=1}^{K} \sum_{h=1}^{H} \sum_{(s,a) \notin B_k^p} v_{k,h}^{observe,p}(s,a) \leq 8HSA \log\left(\frac{HSN}{\delta'}\right).$$

*Proof of Lemma 12.* We have

$$
\begin{aligned}
\sum_{k=1}^{K} \sum_{h=1}^{H} \sum_{(s,a) \notin B_k^q} v_{k,h}^{observe,q}(s,a) &= \sum_{(s,a) \in \mathcal{S} \times A^{\mathrm{ground}} \setminus \{a_\perp\}} \sum_{k=1}^{K} \sum_{h=1}^{H} v_{k,h}^{observe,q}(s,a) \mathbb{I}\left\{(s,a) \notin B_k^q\right\} \\
&= \sum_{(s,a) \in \mathcal{S} \times A^{\mathrm{ground}} \setminus \{a_\perp\}} \sum_{k=1}^{K} v_k^{observe,q}(s,a) \mathbb{I}\left\{(s,a) \notin B_k^q\right\} \\
&\overset{(a)}{\leq} \sum_{(s,a) \in \mathcal{S} \times A^{\mathrm{ground}} \setminus \{a_\perp\}} \left(4H \log\left(\frac{HSN}{\delta'}\right) + 4H\right) \\
&= 4HSA \log\left(\frac{HSN}{\delta'}\right) + 4HSA \\
&\leq 8HSA \log\left(\frac{HSN}{\delta'}\right),
\end{aligned}
$$

where inequality (a) uses the definition of $B_k^q$.

The second inequality in this lemma can be obtained by applying a similar analysis. $\quad\square$

**Lemma 13** (Visitation Ratio). *It holds that*

$$\sum_{k=1}^{K}\sum_{h=1}^{H}\sum_{(s,a)\in B_k^q}\frac{v_{k,h}^{observe,q}(s,a)}{n^{k,q}(s,a)}\leq 8SA\log\left(KH\right),$$

$$\sum_{k=1}^{K}\sum_{h=1}^{H}\sum_{(s,a)\in B_k^p}\frac{v_{k,h}^{observe,p}(s,a)}{n^{k,p}(s,a)}\leq 8SA\log\left(KH\right).$$

*Proof of Lemma 13.* We have

$$\sum_{k=1}^{K}\sum_{h=1}^{H}\sum_{(s,a)\in B_k^q}\frac{v_{k,h}^{observe,q}(s,a)}{n^{k,q}(s,a)}=\sum_{k=1}^{K}\sum_{(s,a)\in B_k^q}\frac{v_{k}^{observe,q}(s,a)}{n^{k,q}(s,a)}$$

$$=\sum_{k=1}^{K}\sum_{(s,a)\in\mathcal{S}\times A^{\text{ground}}\setminus\{a_\perp\}}\frac{v_{k}^{observe,q}(s,a)}{n^{k,q}(s,a)}\mathbb{I}\left\{(s,a)\in B_k^q\right\}$$

$$\leq 4\sum_{(s,a)\in\mathcal{S}\times A^{\text{ground}}\setminus\{a_\perp\}}\sum_{k=1}^{K}\frac{v_{k}^{observe,q}(s,a)}{\sum_{k'\leq k}v_{k'}^{observe,q}(s,a)}\mathbb{I}\left\{(s,a)\in B_k^q\right\}$$

$$\overset{(a)}{\leq}8SA\log\left(KH\right),$$

where inequality (a) comes from Lemma 13 in (Zanette & Brunskill, 2019).

Using a similar analysis, we can also obtain the second inequality in this lemma. $\square$

### C.2.3 OPTIMISM AND PESSIMISM

Let $L:=\log(\frac{KHSA}{\delta'})$. For any $k>0$, $h\in[H]$ and $s\in\mathcal{S}$, we define

$$\begin{cases} b^{k,q}(s,a):=\min\left\{2\sqrt{\dfrac{\hat{q}^k(s,a)(1-\hat{q}^k(s,a))L}{n^{k,q}(s,a)}}+\dfrac{5L}{n^{k,q}(s,a)},\ 1\right\},\ \forall a\in A^{\text{ground}}\setminus\{a_\perp\}, \\[3ex] b^{k,q}(s,a_\perp):=0, \\[3ex] b^{k,pV}(s,a):=\min\left\{2\sqrt{\dfrac{\text{Var}_{s'\sim\hat{p}^k}\left(\bar{V}_{h+1}^k(s')\right)L}{n^{k,p}(s,a)}}+2\sqrt{\dfrac{\mathbb{E}_{s'\sim\hat{p}^k}\left[\left(\bar{V}_{h+1}^k(s')-\underline{V}_{h+1}^k(s')\right)^2\right]L}{n^{k,p}(s,a)}}\right. \\[3ex] \left.\qquad\qquad +\dfrac{5HL}{n^{k,p}(s,a)},\ H\right\},\ \forall a\in A^{\text{ground}}. \end{cases}$$

For any $k>0$, $h\in[H]$, $s\in\mathcal{S}$ and $A\in\mathcal{A}$, we define

$$\begin{cases} \bar{Q}_h^k(s,A)=\min\left\{\sum_{i=1}^{|A|}\prod_{j=1}^{i-1}(1-\bar{q}^k(s,A(j)))\bar{q}^k(s,A(i))\cdot\right. \\[3ex] \qquad\qquad\left. \left(r(s,A(i))+\hat{p}^k(\cdot|s,A(i))^\top\bar{V}_{h+1}^k+b^{k,pV}(s,A(i))\right),\ H\right\} \\[3ex] \pi_h^k(s)=\underset{A\in\mathcal{A}}{\operatorname{argmax}}\ \bar{Q}_h^k(s,A), \\[1.5ex] \bar{V}_h^k(s)=\bar{Q}_h^k(s,\pi_h^k(s)), \\[1.5ex] \bar{V}_{H+1}^k(s)=0, \end{cases}$$

$$\begin{cases} \underline{Q}_h^k(s, A) = \max\Bigg\{ \sum_{i=1}^{|A|} \prod_{j=1}^{i-1}(1 - \bar{q}^k(s, A(j)))\underline{q}^k(s, A(i)) \cdot \\ \qquad\qquad \Big( r(s, A(i)) + \hat{p}^k(\cdot|s, A(i))^\top \underline{V}_{h+1}^k - b^{k,pV}(s, A(i)) \Big), \ 0 \Bigg\}, \\ \underline{V}_h^k(s) = \underline{Q}_h^k(s, \pi_h^k(s)), \\ \underline{V}_{H+1}^k(s) = 0. \end{cases}$$

We first prove the monotonicity of $f$, which will be used in the proof of optimism.

**Lemma 14** (Monotonicity of $f$). *For any* $A = (a_1, \ldots, a_n, a_\perp) \in \mathcal{A}$, $w : A^{\text{ground}} \mapsto \mathbb{R}$ *and* $\bar{u}, u : A^{\text{ground}} \mapsto [0, 1]$ *such that* $1 \le n \le m$, $w(a_1) \ge \cdots \ge w(a_n) \ge w(a_\perp)$ *and* $\bar{u}(a) \ge u(a)$ *for any* $a \in A$, *we have*

$$f(A, \bar{u}, w) \ge f(A, u, w).$$

*Proof of Lemma 14.* In the following, we make the convention $a_{n+1} = a_\perp$ and prove $f((a_1, \ldots, a_k), \bar{u}, w) \ge f((a_1, \ldots, a_k), u, w)$ for $k = 1, 2, \ldots, n+1$ by induction.

Then, it suffices to prove that for $k = 1, 2, \ldots, n+1$,

$$\sum_{i=1}^{k} \prod_{j=1}^{i-1}(1 - \bar{u}(a_j))\bar{u}(a_i)w(a_i) - \sum_{i=1}^{k} \prod_{j=1}^{i-1}(1 - u(a_j))u(a_i)w(a_i)$$
$$\ge (1 - \bar{u}(a_1)) \cdots (1 - \bar{u}(a_{k-1}))(\bar{u}(a_k) - u(a_k))w(a_k)$$
$$+ (1 - \bar{u}(a_1)) \cdots (1 - \bar{u}(a_{k-2}))(\bar{u}(a_{k-1}) - u(a_{k-1}))(1 - u(a_k))w(a_k) + \ldots$$
$$+ (\bar{u}(a_1) - u(a_1))(1 - u(a_2)) \cdots (1 - u(a_k))w(a_k), \qquad (11)$$

since the right-hand side of the above equation is nonnegative.

First, for $k = 1$, we have

$$\bar{u}(a_1)w(a_1) - u(a_1)w(a_1) = (\bar{u}(a_1) - u(a_1))\, w(a_1),$$

and thus Eq. (11) holds.

Then, for any $1 \le k \le n$, supposing that Eq. (11) holds for $k$, we prove that it also holds for $k+1$.

$$\sum_{i=1}^{k+1} \prod_{j=1}^{i-1}(1 - \bar{u}(a_j))\bar{u}(a_i)w(a_i) - \sum_{i=1}^{k+1} \prod_{j=1}^{i-1}(1 - u(a_j))u(a_i)w(a_i)$$
$$\ge \sum_{i=1}^{k} \prod_{j=1}^{i-1}(1 - \bar{u}(a_j))\bar{u}(a_i)w(a_i) - \sum_{i=1}^{k} \prod_{j=1}^{i-1}(1 - u(a_j))u(a_i)w(a_i)$$
$$+ \underbrace{(1-\bar{u}(a_1)) \cdots (1-\bar{u}(a_k))\bar{u}(a_{k+1})w(a_{k+1}) - (1-u(a_1)) \cdots (1-u(a_k))u(a_{k+1})w(a_{k+1})}_{\text{Term } D}.$$
$$(12)$$

Here, we have

$$\text{Term } D = (1 - \bar{u}(a_1)) \cdots (1 - \bar{u}(a_k))\bar{u}(a_{k+1})w(a_{k+1})$$
$$- (1 - \bar{u}(a_1)) \cdots (1 - \bar{u}(a_k))u(a_{k+1})w(a_{k+1})$$
$$+ (1 - \bar{u}(a_1)) \cdots (1 - \bar{u}(a_k))u(a_{k+1})w(a_{k+1})$$
$$- (1 - u(a_1)) \cdots (1 - u(a_k))u(a_{k+1})w(a_{k+1})$$
$$= (1 - \bar{u}(a_1)) \cdots (1 - \bar{u}(a_k))\,(\bar{u}(a_{k+1}) - u(a_{k+1}))\, w(a_{k+1})$$
$$+ \Big[(1 - \bar{u}(a_1)) \cdots (1 - \bar{u}(a_k)) - (1 - u(a_1)) \cdots (1 - u(a_k))\Big] u(a_{k+1})w(a_{k+1})$$
$$= (1 - \bar{u}(a_1)) \cdots (1 - \bar{u}(a_k))\,(\bar{u}(a_{k+1}) - u(a_{k+1}))\, w(a_{k+1})$$
$$+ \Big[(1 - \bar{u}(a_1)) \cdots (1 - \bar{u}(a_k))$$

$$
\begin{aligned}
&\quad - (1-\bar{u}(a_1))\cdots(1-\bar{u}(a_{k-1})(1-u(a_k)) \\
&\quad + (1-\bar{u}(a_1))\cdots(1-\bar{u}(a_{k-1})(1-u(a_k)) \\
&\quad \Big. - (1-u(a_1))\cdots(1-u(a_k))\Big]u(a_{k+1})w(a_{k+1}) \\
&= (1-\bar{u}(a_1))\cdots(1-\bar{u}(a_k))\left(\bar{u}(a_{k+1})-u(a_{k+1})\right)w(a_{k+1}) \\
&\quad + (1-\bar{u}(a_1))\cdots(1-\bar{u}(a_{k-1})(u(a_k)-\bar{u}(a_k))u(a_{k+1})w(a_{k+1}) \\
&\quad + \Big[(1-\bar{u}(a_1))\cdots(1-\bar{u}(a_{k-1})-(1-u(a_1))\cdots(1-u(a_{k-1}))\Big]\cdot \\
&\quad (1-u(a_k))u(a_{k+1})w(a_{k+1}) \\
&= (1-\bar{u}(a_1))\cdots(1-\bar{u}(a_k))\left(\bar{u}(a_{k+1})-u(a_{k+1})\right)w(a_{k+1}) \\
&\quad + (1-\bar{u}(a_1))\cdots(1-\bar{u}(a_{k-1})(u(a_k)-\bar{u}(a_k))u(a_{k+1})w(a_{k+1}) \\
&\quad + (1-\bar{u}(a_1))\cdots(1-\bar{u}(a_{k-2})(u(a_{k-1})-\bar{u}(a_{k-1}))(1-u(a_k))u(a_{k+1})w(a_{k+1}) \\
&\quad + \ldots \\
&\quad + (u(a_1)-\bar{u}(a_1))(1-u(a_2))\cdots(1-u(a_k))u(a_{k+1})w(a_{k+1}).
\end{aligned} \tag{13}
$$

Plugging Eq. (13) and the induction hypothesis into Eq. (12), we have

$$
\begin{aligned}
&\sum_{i=1}^{k+1}\prod_{j=1}^{i-1}(1-\bar{u}(a_j))\bar{u}(a_i)w(a_i) - \sum_{i=1}^{k+1}\prod_{j=1}^{i-1}(1-u(a_j))u(a_i)w(a_i) \\
&\geq (1-\bar{u}(a_1))\cdots(1-\bar{u}(a_k))\left(\bar{u}(a_{k+1})-u(a_{k+1})\right)w(a_{k+1}) \\
&\quad + (1-\bar{u}(a_1))\cdots(1-\bar{u}(a_{k-1})(\bar{u}(a_k)-u(a_k))\Big[w(a_k)-u(a_{k+1})w(a_{k+1})\Big] \\
&\quad + (1-\bar{u}(a_1))\cdots(1-\bar{u}(a_{k-2})(\bar{u}(a_{k-1})-u(a_{k-1}))(1-u(a_k))\Big[w(a_k)-u(a_{k+1})w(a_{k+1})\Big] \\
&\quad + \ldots \\
&\quad + (\bar{u}(a_1)-u(a_1))(1-u(a_2))\cdots(1-u(a_k))\Big[w(a_k)-u(a_{k+1})w(a_{k+1})\Big] \\
&\geq (1-\bar{u}(a_1))\cdots(1-\bar{u}(a_k))\left(\bar{u}(a_{k+1})-u(a_{k+1})\right)w(a_{k+1}) \\
&\quad + (1-\bar{u}(a_1))\cdots(1-\bar{u}(a_{k-1})(\bar{u}(a_k)-u(a_k))(1-u(a_{k+1}))w(a_{k+1}) \\
&\quad + (1-\bar{u}(a_1))\cdots(1-\bar{u}(a_{k-2})(\bar{u}(a_{k-1})-u(a_{k-1}))(1-u(a_k))(1-u(a_{k+1}))w(a_{k+1}) \\
&\quad + \ldots \\
&\quad + (\bar{u}(a_1)-u(a_1))(1-u(a_2))\cdots(1-u(a_k))(1-u(a_{k+1}))w(a_{k+1}).
\end{aligned}
$$

Therefore, Eq. (11) holds for $k+1$, and we complete the proof. $\qquad\square$

Now we prove the optimism of $\bar{V}_h^k(s)$ and pessimism of $\underline{V}_h^k(s)$.

**Lemma 15** (Optimism and Pessimism). *Assume that event $\mathcal{E}\cap\mathcal{F}\cap\mathcal{G}$ holds. Then, for any $k\in[K]$, $h\in[H]$ and $s\in\mathcal{S}$,*

$$
\bar{V}_h^k(s) \geq V_h^*(s) \geq \underline{V}_h^k(s).
$$

*In addition, for any $k\in[K]$, $h\in[H]$ and $(s,a)\in\mathcal{S}\times A^{\mathrm{ground}}$, it holds that*

$$
\begin{aligned}
&\left|\left(\hat{p}^k(\cdot|s,a)-p(\cdot|s,a)\right)^\top V_{h+1}^*\right| \leq 2\sqrt{\frac{\mathrm{Var}_{s'\sim\hat{p}^k}\left(\bar{V}_{h+1}^k(s')\right)\log\left(\frac{KHSA}{\delta'}\right)}{n^{k,p}(s,a)}} \\
&\quad + 2\sqrt{\frac{\mathbb{E}_{s'\sim\hat{p}^k}\left[\left(\bar{V}_{h+1}^k(s')-\underline{V}_{h+1}^k(s')\right)^2\right]\log\left(\frac{KHSA}{\delta'}\right)}{n^{k,p}(s,a)}} + \frac{5H\log\left(\frac{KHSA}{\delta'}\right)}{n^{k,p}(s,a)}.
\end{aligned}
$$

*Proof of Lemma 15.* We prove this lemma by induction.

For any $k\in[K]$ and $s\in\mathcal{S}$, it holds that $\bar{V}_{H+1}^k(s)=V_{H+1}^*(s)=\underline{V}_{H+1}^k(s)=0$.

First, we prove optimism. For any $k \in [K]$ and $h \in [H]$, if $\bar{V}_h^k(s) = H$, then $\bar{V}_h^k(s) \geq V_h^*(s)$ trivially holds. Otherwise, supposing $\bar{V}_{h+1}^k(s') \geq V_{h+1}^*(s') \geq \underline{V}_{h+1}^k(s')$ for any $s' \in \mathcal{S}$, we have

$$
\begin{aligned}
\bar{V}_h^k(s) &= \bar{Q}_h^k(s, \pi_h^k(s)) \\
&\geq \bar{Q}_h^k(s, A^*) \\
&= \sum_{i=1}^{|A^*|} \prod_{j=1}^{i-1}(1 - \bar{q}^k(s, A^*(j)))\bar{q}^k(s, A^*(i))\Big(r(s, A^*(i)) + \hat{p}^k(\cdot|s, A^*(i))^\top \bar{V}_{h+1}^k \\
&\quad + b^{k,pV}(s, A^*(i))\Big) \\
&\overset{(a)}{\geq} \sum_{i=1}^{|A^*|} \prod_{j=1}^{i-1}(1 - \bar{q}^k(s, A^*(j)))\bar{q}^k(s, A^*(i))\Big(r(s, A^*(i)) + \hat{p}^k(\cdot|s, A^*(i))^\top V_{h+1}^* \\
&\quad + b^{k,pV}(s, A^*(i))\Big) \\
&\overset{(b)}{\geq} \sum_{i=1}^{|A^*|} \prod_{j=1}^{i-1}(1 - \bar{q}^k(s, A^*(j)))\bar{q}^k(s, A^*(i))\Big(r(s, A^*(i)) + p(\cdot|s, A^*(i))^\top V_{h+1}^*\Big) \\
&\overset{(c)}{\geq} \sum_{i=1}^{|A^*|} \prod_{j=1}^{i-1}(1 - q(s, A^*(j)))q(s, A^*(i))\Big(r(s, A^*(i)) + p(\cdot|s, A^*(i))^\top V_{h+1}^*\Big) \\
&= Q_h^*(s, A^*). \\
&= V_h^*(s).
\end{aligned}
$$

Here $A^* := \operatorname{argmax}_{A \in \mathcal{A}} \sum_{i=1}^{|A|} \prod_{j=1}^{i-1}(1 - q(s, A(j)))q(s, A(i))(r(s, A(i)) + p(\cdot|s, A(i))^\top V_{h+1}^*)$. Inequality (a) uses the induction hypothesis. Inequality (b) applies the second statement of Lemma 8 with the induction hypothesis. Inequality (c) follows from Lemma 14 and the fact that the optimal permutation $A^*$ satisfies that the items in $A^*$ are ranked in descending order of $w(a) := r(s, a) + p(\cdot|s, a)^\top V_{h+1}^*$.

Next, we prove pessimism. For any $k \in [K]$ and $h \in [H]$, if $\underline{V}_h^k(s) = 0$, then $\underline{V}_h^k(s) \leq V_h^*(s)$ trivially holds. Otherwise, supposing $\bar{V}_{h+1}^k(s') \geq V_{h+1}^*(s') \geq \underline{V}_{h+1}^k(s')$ for any $s' \in \mathcal{S}$, we have

$$
\begin{aligned}
\underline{Q}_h^k(s, A) &= \sum_{i=1}^{|A|} \prod_{j=1}^{i-1}(1 - \bar{q}^k(s, A(j)))\underline{q}^k(s, A(i)) \cdot \\
&\quad \Big(r(s, A(i)) + \hat{p}^k(\cdot|s, A(i))^\top \underline{V}_{h+1}^k - b^{k,pV}(s, A(i))\Big) \\
&\overset{(a)}{\leq} \sum_{i=1}^{|A|} \prod_{j=1}^{i-1}(1 - \bar{q}^k(s, A(j)))\underline{q}^k(s, A(i)) \cdot \\
&\quad \Big(r(s, A(i)) + \hat{p}^k(\cdot|s, A(i))^\top V_{h+1}^* - b^{k,pV}(s, A(i))\Big) \\
&\leq \sum_{i=1}^{|A|} \prod_{j=1}^{i-1}(1 - q(s, A(j)))q(s, A(i))\Big(r(s, A(i)) + p(\cdot|s, A(i))^\top V_{h+1}^*\Big) \\
&= Q_h^*(s, A),
\end{aligned}
$$

where inequality (a) uses the induction hypothesis.

Then, we have

$$
\underline{V}_h^k(s) = \underline{Q}_h^k(s, \pi_h^k(s)) \leq Q_h^*(s, \pi_h^k(s)) \leq Q_h^*(s, A^*) = V_h^*(s),
$$

which completes the proof of the first statement.

Combining the first statement and Lemma 8, we obtain the second statement. $\qquad\square$

### C.2.4 SECOND ORDER TERM

**Lemma 16** (Gap between Optimism and Pessimism). *For any $k > 0$, $h \in [H]$ and $s \in \mathcal{S}$,*

$$
\bar{V}_h^k(s) - \underline{V}_h^k(s) \leq \sum_{t=h}^{H} \mathbb{E}_{(s_t, A_t) \sim \pi^k} \left[ \sum_{i=1}^{|A_t|-1} \prod_{j=1}^{i-1} (1 - q^k(s_t, A_t(j))) \frac{66 H \sqrt{L}}{\sqrt{n^{k,q}(s_t, A_t(i))}} \right.
$$
$$
\left. + \sum_{i=1}^{|A_t|} \prod_{j=1}^{i-1} (1 - q^k(s_t, A_t(j))) q(s_t, A_t(i)) \frac{20 H L \sqrt{S}}{\sqrt{n^{k,p}(s_t, A_t(i))}} \Big| s_h = s, \pi^k \right].
$$

*Proof of Lemma 16.* Let $A' = \pi_h^k(s)$. Recall that

$$
\bar{V}_h^k(s) \leq \sum_{i=1}^{|A'|} \prod_{j=1}^{i-1} (1 - \bar{q}^k(s, A'(j))) \bar{q}^k(s, A'(i)) \cdot
$$
$$
\left( r(s, A'(i)) + \hat{p}^k(\cdot|s, A'(i))^\top \bar{V}_{h+1}^k + b^{k,pV}(s, A'(i)) \right),
$$

and

$$
\underline{V}_h^k(s) \geq \sum_{i=1}^{|A'|} \prod_{j=1}^{i-1} (1 - \bar{q}^k(s, A'(j))) \underline{q}^k(s, A'(i)) \cdot
$$
$$
\left( r(s, A'(i)) + \hat{p}^k(\cdot|s, A'(i))^\top \underline{V}_{h+1}^k - b^{k,pV}(s, A'(i)) \right).
$$

Then, we have

$$
\bar{V}_h^k(s) - \underline{V}_h^k(s) \leq \sum_{i=1}^{|A'|} \prod_{j=1}^{i-1} (1 - \bar{q}^k(s, A'(j))) \cdot \Big( 2 r(s, A'(i)) b^{k,q}(s, A'(i))
$$
$$
+ \left( \underline{q}^k(s, A'(i)) + 2 b^{k,q}(s, A'(i)) \right) \left( \hat{p}^k(\cdot|s, A'(i))^\top \bar{V}_{h+1}^k + b^{k,pV}(s, A'(i)) \right)
$$
$$
- \underline{q}^k(s, A'(i)) \left( \hat{p}^k(\cdot|s, A'(i))^\top \underline{V}_{h+1}^k - b^{k,pV}(s, A'(i)) \right) \Big)
$$
$$
\leq \sum_{i=1}^{|A'|} \prod_{j=1}^{i-1} (1 - q^k(s, A'(j))) \cdot \Big( q(s, A'(i)) \Big( \hat{p}^k(\cdot|s, A'(i))^\top \bar{V}_{h+1}^k
$$
$$
- \hat{p}^k(\cdot|s, A'(i))^\top \underline{V}_{h+1}^k + 2 b^{k,pV}(s, A'(i)) \Big) + 6 H b^{k,q}(s, A'(i)) \Big)
$$
$$
= 6H \sum_{i=1}^{|A'|-1} \prod_{j=1}^{i-1} (1 - q^k(s, A'(j))) \left( 2 \sqrt{\frac{\hat{q}^k(s,a)(1 - \hat{q}^k(s,a)) L}{n^{k,q}(s,a)}} + \frac{5L}{n^{k,q}(s,a)} \right)
$$
$$
+ 2 \sum_{i=1}^{|A'|} \prod_{j=1}^{i-1} (1 - q^k(s, A'(j))) q(s, A'(i)) \cdot \left( 2 \sqrt{\frac{\mathrm{Var}_{s' \sim \hat{p}^k} \left( \bar{V}_{h+1}^k(s') \right) L}{n^{k,p}(s, A'(i))}} \right.
$$
$$
\left. + 2 \sqrt{\frac{\mathbb{E}_{s' \sim \hat{p}^k} \left[ \left( \bar{V}_{h+1}^k(s') - \underline{V}_{h+1}^k(s') \right)^2 \right] L}{n^{k,p}(s, A'(i))}} + \frac{5HL}{n^{k,p}(s, A'(i))} \right)
$$
$$
+ \sum_{i=1}^{|A'|} \prod_{j=1}^{i-1} (1 - q^k(s, A'(j))) q(s, A'(i)) \cdot
$$
$$
\left( \hat{p}^k(\cdot|s, A'(i)) - p(\cdot|s, A'(i)) \right)^\top \left( \bar{V}_{h+1}^k - \underline{V}_{h+1}^k \right)
$$
$$
+ \sum_{i=1}^{|A'|} \prod_{j=1}^{i-1} (1 - q^k(s, A'(j))) q(s, A'(i)) p(\cdot|s, A'(i))^\top \left( \bar{V}_{h+1}^k - \underline{V}_{h+1}^k \right)
$$

$$
\stackrel{(a)}{\leq} 6H \sum_{i=1}^{|A'|-1} \prod_{j=1}^{i-1} (1 - q^k(s, A'(j))) \left( 2\sqrt{\frac{q(s,a)(1-q(s,a))L}{n^{k,q}(s,a)}} + \frac{9L}{n^{k,q}(s,a)} \right)
$$

$$
+ \sum_{i=1}^{|A'|} \prod_{j=1}^{i-1} (1 - q^k(s, A'(j))) q(s, A'(i)) \frac{18HL}{\sqrt{n^{k,p}(s, A'(i))}}
$$

$$
+ \sum_{i=1}^{|A'|} \prod_{j=1}^{i-1} (1 - q^k(s, A'(j))) q(s, A'(i)) \frac{2H\sqrt{SL}}{\sqrt{n^{k,p}(s, A'(i))}}
$$

$$
+ \sum_{i=1}^{|A'|} \prod_{j=1}^{i-1} (1 - q^k(s, A'(j))) q(s, A'(i)) p(\cdot|s, A'(i))^\top \left( \bar{V}_{h+1}^k - \underline{V}_{h+1}^k \right)
$$

$$
\leq \sum_{i=1}^{|A'|-1} \prod_{j=1}^{i-1} (1 - q^k(s, A'(j))) \frac{66H\sqrt{L}}{\sqrt{n^{k,q}(s, A'(i))}}
$$

$$
+ \sum_{i=1}^{|A'|} \prod_{j=1}^{i-1} (1 - q^k(s, A'(j))) q(s, A'(i)) \frac{20HL\sqrt{S}}{\sqrt{n^{k,p}(s, A'(i))}}
$$

$$
+ \sum_{i=1}^{|A'|} \prod_{j=1}^{i-1} (1 - q^k(s, A'(j))) q(s, A'(i)) p(\cdot|s, A'(i))^\top \left( \bar{V}_{h+1}^k - \underline{V}_{h+1}^k \right)
$$

$$
\leq \sum_{t=h}^{H} \mathbb{E}_{(s_t, A_t) \sim \pi^k} \left[ \sum_{i=1}^{|A_t|-1} \prod_{j=1}^{i-1} (1 - q^k(s_t, A_t(j))) \frac{66H\sqrt{L}}{\sqrt{n^{k,q}(s_t, A_t(i))}} \right.
$$

$$
\left. + \sum_{i=1}^{|A_t|} \prod_{j=1}^{i-1} (1 - q^k(s_t, A_t(j))) q(s_t, A_t(i)) \frac{20HL\sqrt{S}}{\sqrt{n^{k,p}(s_t, A_t(i))}} \Big| s_h = s, \pi^k \right],
$$

where inequality (a) is due to Eq. (8). $\qquad \square$

**Lemma 17** (Cumulative Gap between Optimism and Pessimism). *It holds that*

$$
\sum_{k=1}^{K} \sum_{h=1}^{H} \sum_{(s,a) \in \mathcal{S} \times A^{\text{ground}}} v_{k,h}^{observe,p}(s,a) \cdot p(\cdot|s,a)^\top \left( \bar{V}_{h+1}^k - \underline{V}_{h+1}^k \right)^2 \leq 152192(m+1)H^5 S^2 N L^3.
$$

*Proof of Lemma 17.* For any $k > 0$, $h \in [H]$ and $s \in \mathcal{S}$, let $w_{k,h}(s)$ denote the probability that state $s$ is visited at step $h$ in episode $k$.

We have

$$
\sum_{k=1}^{K} \sum_{h=1}^{H} \sum_{(s,a) \in \mathcal{S} \times A^{\text{ground}}} v_{k,h}^{observe,p}(s,a) \cdot p(\cdot|s,a)^\top \left( \bar{V}_{h+1}^k - \underline{V}_{h+1}^k \right)^2
$$

$$
= \sum_{k=1}^{K} \sum_{h=1}^{H} \sum_{(s,a) \in \mathcal{S} \times A^{\text{ground}}} v_{k,h}^{observe,p}(s,a) \sum_{s' \in \mathcal{S}} p(s'|s,a) \left( \bar{V}_{h+1}^k(s') - \underline{V}_{h+1}^k(s') \right)^2
$$

$$
= \sum_{k=1}^{K} \sum_{h=1}^{H} \sum_{(s,a) \in \mathcal{S} \times A^{\text{ground}}} \sum_{s' \in \mathcal{S}} v_{k,h}^{observe,p}(s',s,a) \left( \bar{V}_{h+1}^k(s') - \underline{V}_{h+1}^k(s') \right)^2
$$

$$
= \sum_{k=1}^{K} \sum_{h=1}^{H} \sum_{s' \in \mathcal{S}} w_{k,h+1}(s') \left( \bar{V}_{h+1}^k(s') - \underline{V}_{h+1}^k(s') \right)^2
$$

$$
= \sum_{k=1}^{K} \sum_{h=1}^{H} \mathbb{E}_{s_{h+1} \sim \pi^k} \left[ \left( \bar{V}_{h+1}^k(s_{h+1}) - \underline{V}_{h+1}^k(s_{h+1}) \right)^2 \right]
$$

$$\leq \sum_{k=1}^{K}\sum_{h=1}^{H}\mathbb{E}_{s_h\sim\pi^k}\left[\left(\bar{V}_h^k(s_h)-\underline{V}_h^k(s_h)\right)^2\right]$$

$$\overset{(a)}{\leq}\sum_{k=1}^{K}\sum_{h=1}^{H}\mathbb{E}_{s_h\sim\pi^k}\left[\left(\sum_{t=h}^{H}\mathbb{E}_{(s_t,A_t)\sim\pi^k}\left[\sum_{i=1}^{|A_t|-1}\prod_{j=1}^{i-1}(1-q^k(s_t,A_t(j)))\frac{66H\sqrt{L}}{\sqrt{n^{k,q}(s_t,A_t(i))}}\right.\right.\right.$$

$$\left.\left.\left.+\sum_{i=1}^{|A_t|}\prod_{j=1}^{i-1}(1-q^k(s,A_t(j)))q(s_t,A_t(i))\frac{20HL\sqrt{S}}{\sqrt{n^{k,p}(s_t,A_t(i))}}\Big|s_h=s,\pi^k\right]\right)^2\right]$$

$$\overset{(b)}{\leq}H\sum_{k=1}^{K}\sum_{h=1}^{H}\mathbb{E}_{s_h\sim\pi^k}\left[\sum_{t=h}^{H}\left(\mathbb{E}_{(s_t,A_t)\sim\pi^k}\left[\sum_{i=1}^{|A_t|-1}\prod_{j=1}^{i-1}(1-q(s_t,A_t(j)))\frac{66H\sqrt{L}}{\sqrt{n^{k,q}(s_t,A_t(i))}}\right.\right.\right.$$

$$\left.\left.\left.+\sum_{i=1}^{|A_t|}\prod_{j=1}^{i-1}(1-q(s,A_t(j)))q(s_t,A_t(i))\frac{20HL\sqrt{S}}{\sqrt{n^{k,p}(s_t,A_t(i))}}\Big|s_h=s,\pi^k\right]\right)^2\right]$$

$$\overset{(c)}{\leq}H\sum_{k=1}^{K}\sum_{h=1}^{H}\mathbb{E}_{s_h\sim\pi^k}\left[\sum_{t=h}^{H}\mathbb{E}_{(s_t,A_t)\sim\pi^k}\left[\left(\sum_{i=1}^{|A_t|-1}\prod_{j=1}^{i-1}(1-q(s_t,A_t(j)))\frac{66H\sqrt{L}}{\sqrt{n^{k,q}(s_t,A_t(i))}}\right.\right.\right.$$

$$\left.\left.\left.+\sum_{i=1}^{|A_t|}\prod_{j=1}^{i-1}(1-q(s,A_t(j)))q(s_t,A_t(i))\frac{20HL\sqrt{S}}{\sqrt{n^{k,p}(s_t,A_t(i))}}\right)^2\Big|s_h=s,\pi^k\right]\right]$$

$$\overset{(d)}{\leq}2(m+1)H\sum_{k=1}^{K}\sum_{h=1}^{H}\sum_{t=h}^{H}\mathbb{E}_{(s_t,A_t)\sim\pi^k}\left[\sum_{i=1}^{|A_t|-1}\prod_{j=1}^{i-1}(1-q(s_t,A_t(j)))^2\frac{4356H^2L}{n^{k,q}(s_t,A_t(i))}\right.$$

$$\left.+\sum_{i=1}^{|A_t|}\prod_{j=1}^{i-1}(1-q(s,A_t(j)))^2q(s_t,A_t(i))^2\frac{400H^2L^2S}{n^{k,p}(s_t,A_t(i))}\right]$$

$$=2(m+1)H^2\sum_{k=1}^{K}\sum_{h=1}^{H}\left(\sum_{(s,a)\in\mathcal{S}\times(A^{\mathrm{ground}}\setminus\{a_\perp\})}v_{k,h}^{observe,q}(s,a)\frac{4356H^2L}{n^{k,q}(s,a)}\right.$$

$$\left.+\sum_{(s,a)\in\mathcal{S}\times A^{\mathrm{ground}}}v_{k,h}^{observe,p}(s,a)\frac{400H^2L^2S}{n^{k,p}(s,a)}\right)$$

$$\overset{(e)}{\leq}2(m+1)H^2\cdot\left(8SN\log(KH)+8HSN\log\left(\frac{HSN}{\delta'}\right)\right)\cdot(4356H^2L+400H^2L^2S)$$

$$\leq152192(m+1)H^5S^2NL^3.$$

Here inequality (a) uses Lemma 16. Inequalities (b) and (d) are due to the Cauchy-Schwarz inequality. Inequality (c) comes from Jensen's inequality. Inequality (e) follows from Lemmas 12 and 13. $\square$

### C.2.5 PROOF OF THEOREM 1

*Proof of Theorem 1.* Recall that $\delta':=\frac{\delta}{14}$. Combining Lemmas 6-10, we have $\Pr[\mathcal{E}\cap\mathcal{F}\cap\mathcal{G}\cap\mathcal{K}]\geq 1-14\delta'=1-\delta$.

In the following, we assume that event $\mathcal{E}\cap\mathcal{F}\cap\mathcal{G}\cap\mathcal{K}$ holds, and then prove the regret upper bound.

$$\mathcal{R}(K)=\sum_{k=1}^{K}\left(V_1^*(s_1^k)-V_1^{\pi_k}(s_1^k)\right)$$

$$\leq\sum_{k=1}^{K}\left(\bar{V}_1^k(s_1^k)-V_1^{\pi_k}(s_1^k)\right)$$

$$\leq \sum_{k=1}^{K}\sum_{h=1}^{H}\mathbb{E}\Bigg[\sum_{i=1}^{|A_h|}\prod_{j=1}^{i-1}(1-\bar{q}^k(s_h,A_h(j)))\bar{q}^k(s_h,A_h(i))r(s_h,A_h(i))$$

$$-\sum_{i=1}^{|A_h|}\prod_{j=1}^{i-1}(1-q(s_h,A_h(j)))q(s_h,A_h(i))r(s_h,A_h(i))$$

$$+\sum_{i=1}^{|A_h|}\prod_{j=1}^{i-1}(1-\bar{q}^k(s_h,A_h(j)))\bar{q}^k(s_h,A_h(i))\left(\hat{p}^k(\cdot|s_h,A_h(i))^\top \bar{V}_{h+1}^k + b^{k,pV}(s_h,A_h(i))\right)$$

$$-\sum_{i=1}^{|A_h|}\prod_{j=1}^{i-1}(1-q(s_h,A_h(j)))q(s_h,A_h(i))p(\cdot|s_h,A_h(i))^\top \bar{V}_{h+1}^k\Bigg]$$

$$\leq \sum_{k=1}^{K}\sum_{h=1}^{H}\mathbb{E}\Bigg[2\sum_{i=1}^{|A_h|}\prod_{j=1}^{i-1}(1-q(s_h,A_h(j)))r(s_h,A_h(i))b^{k,q}(s_h,A_h(i))$$

$$+\sum_{i=1}^{|A_h|}\prod_{j=1}^{i-1}(1-q(s_h,A_h(j)))\left(q(s_h,A_h(i))+2b^{k,q}(s_h,A_h(i))\right)\cdot$$

$$\left(\hat{p}^k(\cdot|s_h,A_h(i))^\top \bar{V}_{h+1}^k + b^{k,pV}(s_h,A_h(i))\right)$$

$$-\sum_{i=1}^{|A_h|}\prod_{j=1}^{i-1}(1-q(s_h,A_h(j)))q(s_h,A_h(i))p(\cdot|s_h,A_h(i))^\top \bar{V}_{h+1}^k\Bigg]$$

$$\leq \sum_{k=1}^{K}\sum_{h=1}^{H}\mathbb{E}\Bigg[2\sum_{i=1}^{|A_h|}\prod_{j=1}^{i-1}(1-q(s_h,A_h(j)))r(s_h,A_h(i))b^{k,q}(s_h,A_h(i))$$

$$+\sum_{i=1}^{|A_h|}\prod_{j=1}^{i-1}(1-q(s_h,A_h(j)))q(s_h,A_h(i))\left(\hat{p}^k(\cdot|s_h,A_h(i))^\top \bar{V}_{h+1}^k + b^{k,pV}(s_h,A_h(i))\right.$$

$$\left.-p(\cdot|s_h,A_h(i))^\top \bar{V}_{h+1}^k\right)+4H\sum_{i=1}^{|A_h|}\prod_{j=1}^{i-1}(1-q(s_h,A_h(j)))b^{k,q}(s_h,A_h(i))\Bigg]$$

$$\overset{(a)}{\leq} \sum_{k=1}^{K}\sum_{h=1}^{H}\mathbb{E}\Bigg[6H\sum_{i=1}^{|A_h|-1}\prod_{j=1}^{i-1}(1-q(s_h,A_h(j)))b^{k,q}(s_h,A_h(i))$$

$$+\sum_{i=1}^{|A_h|}\prod_{j=1}^{i-1}(1-q(s_h,A_h(j)))q(s_h,A_h(i))b^{k,pV}(s_h,A_h(i))$$

$$+\sum_{i=1}^{|A_h|}\prod_{j=1}^{i-1}(1-q(s_h,A_h(j)))q(s_h,A_h(i))\left(\hat{p}^k(\cdot|s_h,A_h(i))-p(\cdot|s_h,A_h(i))\right)^\top V_{h+1}^*$$

$$+\sum_{i=1}^{|A_h|}\prod_{j=1}^{i-1}(1-q(s_h,A_h(j)))q(s_h,A_h(i))\left(\hat{p}^k(\cdot|s_h,A_h(i))-p(\cdot|s_h,A_h(i))\right)^\top \left(\bar{V}_{h+1}^k - V_{h+1}^*\right)\Bigg]$$

$$\leq \sum_{k=1}^{K}\sum_{h=1}^{H}\mathbb{E}\Bigg[6H\sum_{i=1}^{m}\sum_{(s,a)\in\mathcal{S}\times(A^{\mathrm{ground}}\setminus\{a_\perp\})}v_{k,h,i}^{observe,q}(s,a)b^{k,q}(s,a)$$

$$+\sum_{i=1}^{m+1}\sum_{(s,a)\in\mathcal{S}\times A^{\mathrm{ground}}}v_{k,h,i}^{observe,p}(s,a)b^{k,pV}(s,a)$$

$$+\sum_{i=1}^{m+1}\sum_{(s,a)\in\mathcal{S}\times A^{\mathrm{ground}}}v_{k,h,i}^{observe,p}(s,a)\left(\hat{p}^k(\cdot|s,a)-p(\cdot|s,a)\right)^\top V_{h+1}^*$$

$$
+ \sum_{i=1}^{m+1} \sum_{(s,a) \in \mathcal{S} \times A^{\text{ground}}} v_{k,h,i}^{observe,p}(s,a) \left( \hat{p}^k(\cdot|s,a) - p(\cdot|s,a) \right)^\top \left( \bar{V}_{h+1}^k - V_{h+1}^* \right) \Bigg]
$$

$$
\leq \sum_{k=1}^{K} \sum_{h=1}^{H} \sum_{(s,a) \in B_k^q} 6H v_{k,h}^{observe,q}(s,a) b^{k,q}(s,a)
$$

$$
+ \sum_{k=1}^{K} \sum_{h=1}^{H} \sum_{(s,a) \in B_k^p} \Bigg( v_{k,h}^{observe,p}(s,a) b^{k,pV}(s,a)
$$

$$
+ v_{k,h}^{observe,p}(s,a) \left( \hat{p}^k(\cdot|s,a) - p(\cdot|s,a) \right)^\top V_{h+1}^*
$$

$$
+ v_{k,h}^{observe,p}(s,a) \left( \hat{p}^k(\cdot|s,a) - p(\cdot|s,a) \right)^\top \left( \bar{V}_{h+1}^k - V_{h+1}^* \right) \Bigg)
$$

$$
+ 6H \sum_{k=1}^{K} \sum_{h=1}^{H} \sum_{(s,a) \notin B_k^q} v_{k,h}^{observe,q}(s,a) + 3H \sum_{k=1}^{K} \sum_{h=1}^{H} \sum_{(s,a) \notin B_k^p} v_{k,h}^{observe,p}(s,a)
$$

$$
\overset{(b)}{\leq} \sum_{k=1}^{K} \sum_{h=1}^{H} \sum_{(s,a) \in B_k^q} 6H v_{k,h}^{observe,q}(s,a) b^{k,q}(s,a)
$$

$$
+ \sum_{k=1}^{K} \sum_{h=1}^{H} \sum_{(s,a) \in B_k^p} \Bigg( v_{k,h}^{observe,p}(s,a) b^{k,pV}(s,a)
$$

$$
+ v_{k,h}^{observe,p}(s,a) \left( \hat{p}^k(\cdot|s,a) - p(\cdot|s,a) \right)^\top V_{h+1}^*
$$

$$
+ v_{k,h}^{observe,p}(s,a) \left( \hat{p}^k(\cdot|s,a) - p(\cdot|s,a) \right)^\top \left( \bar{V}_{h+1}^k - V_{h+1}^* \right) \Bigg) + 72H^2 SNL, \tag{14}
$$

where inequality (a) is due to that for any $k > 0$ and $s \in \mathcal{S}$, $b^{k,q}(s, a_\perp) := 0$, and inequality (b) uses Lemma 12.

We bound the four terms on the right-hand side of the above inequality as follows.

(i) Term 1:

$$
6H \sum_{k=1}^{K} \sum_{h=1}^{H} \sum_{(s,a) \in B_k^q} v_{k,h}^{observe,q}(s,a) b^{k,q}(s,a)
$$

$$
= 6H \sum_{k=1}^{K} \sum_{h=1}^{H} \sum_{(s,a) \in B_k^q} v_{k,h}^{observe,q}(s,a) \left( 2\sqrt{\frac{\hat{q}^k(s,a)(1 - \hat{q}^k(s,a))L}{n^{k,q}(s,a)}} + \frac{5L}{n^{k,q}(s,a)} \right)
$$

$$
\leq 6H \sum_{k=1}^{K} \sum_{h=1}^{H} \sum_{(s,a) \in B_k^q} v_{k,h}^{observe,q}(s,a) \left( 2\sqrt{\frac{q(s,a)(1 - q(s,a))L}{n^{k,q}(s,a)}} + \frac{9L}{n^{k,q}(s,a)} \right)
$$

$$
\leq 12H\sqrt{L} \sqrt{\sum_{k=1}^{K} \sum_{h=1}^{H} \sum_{(s,a) \in B_k^q} v_{k,h}^{observe,q}(s,a) q(s,a)} \cdot \sqrt{\sum_{k=1}^{K} \sum_{h=1}^{H} \sum_{(s,a) \in B_k^q} \frac{v_{k,h}^{observe,q}(s,a)}{n^{k,q}(s,a)}}
$$

$$
\overset{(a)}{\leq} 12H\sqrt{L} \cdot \sqrt{KH} \cdot \sqrt{8SN \log (KH)}
$$

$$
\leq 48HL\sqrt{KHSN},
$$

where inequality (a) is due to Lemma 9.

(ii) Term 2:

$$
\sum_{k=1}^{K} \sum_{h=1}^{H} \sum_{(s,a) \in B_k^p} v_{k,h}^{observe,p}(s,a) b^{k,pV}(s,a)
$$

$$
\begin{aligned}
=&\sum_{k=1}^{K}\sum_{h=1}^{H}\sum_{(s,a)\in B_k^p}v_{k,h}^{observe,p}(s,a)\left(2\sqrt{\frac{\mathrm{Var}_{s'\sim\hat{p}^k}\left(\bar{V}_{h+1}^k(s')\right)L}{n^{k,p}(s,a)}}\right.\\
&\left.+2\sqrt{\frac{\mathbb{E}_{s'\sim\hat{p}^k}\left[\left(\bar{V}_{h+1}^k(s')-\underline{V}_{h+1}^k(s')\right)^2\right]L}{n^{k,p}(s,a)}}+\frac{5HL}{n^{k,p}(s,a)}\right)\\
\leq& 2\sqrt{L}\sqrt{\sum_{k=1}^{K}\sum_{h=1}^{H}\sum_{(s,a)\in B_k^p}\frac{v_{k,h}^{observe,p}(s,a)}{n^{k,p}(s,a)}}\cdot\sqrt{\sum_{k=1}^{K}\sum_{h=1}^{H}\sum_{(s,a)\in B_k^p}v_{k,h}^{observe,p}(s,a)\mathrm{Var}_{s'\sim\hat{p}^k}\left(\bar{V}_{h+1}^k(s')\right)}\\
&+2\sqrt{L}\sqrt{\sum_{k=1}^{K}\sum_{h=1}^{H}\sum_{(s,a)\in B_k^p}\frac{v_{k,h}^{observe,p}(s,a)}{n^{k,p}(s,a)}}\cdot\\
&\left(\sqrt{\sum_{k=1}^{K}\sum_{h=1}^{H}\sum_{(s,a)\in B_k^p}v_{k,h}^{observe,p}(s,a)p(\cdot|s,a)^\top\left(\bar{V}_{h+1}^k(s')-\underline{V}_{h+1}^k(s')\right)^2}\right.\\
&\left.+\sqrt{\sum_{k=1}^{K}\sum_{h=1}^{H}\sum_{(s,a)\in B_k^p}v_{k,h}^{observe,p}(s,a)\left(\hat{p}^k(\cdot|s,a)-p(\cdot|s,a)\right)^\top\left(\bar{V}_{h+1}^k(s')-\underline{V}_{h+1}^k(s')\right)^2}\right)\\
&+5HL\sum_{k=1}^{K}\sum_{h=1}^{H}\sum_{(s,a)\in B_k^p}\frac{v_{k,h}^{observe,p}(s,a)}{n^{k,p}(s,a)}\\
\leq& 2\sqrt{L}\cdot\sqrt{8SN\log(KH)}\cdot H\sqrt{KH}\\
&+2\sqrt{L}\cdot\sqrt{8SN\log(KH)}\cdot\left(\sqrt{152192(m+1)H^5S^2NL^3}\right.\\
&\left.+\sqrt{H}\cdot\sqrt{1112H^2S^2NL^2\sqrt{(m+1)HL}}\right)+5HL\cdot8SN\log(KH)\\
\leq& 8HL\sqrt{KHSN}+3428H^2SNL^2\sqrt{(m+1)HSL}.
\end{aligned}
$$

(iii) Term 3:

$$
\begin{aligned}
&\sum_{k=1}^{K}\sum_{h=1}^{H}\sum_{(s,a)\in B_k^p}v_{k,h}^{observe,p}(s,a)\left(\hat{p}^k(\cdot|s,a)-p(\cdot|s,a)\right)^\top V_{h+1}^*\\
=&2\sqrt{L}\sum_{k=1}^{K}\sum_{h=1}^{H}\sum_{(s,a)\in B_k^p}v_{k,h}^{observe,p}(s,a)\sqrt{\frac{\mathrm{Var}_{s'\sim p(\cdot|s,a)}\left(V_{h+1}^*(s')\right)}{n^{k,p}(s,a)}}\\
&+HL\sum_{k=1}^{K}\sum_{h=1}^{H}\sum_{(s,a)\in B_k^p}\frac{v_{k,h}^{observe,p}(s,a)}{n^{k,p}(s,a)}\\
\leq& 2\sqrt{L}\sqrt{\sum_{k=1}^{K}\sum_{h=1}^{H}\sum_{(s,a)\in B_k^p}\frac{v_{k,h}^{observe,p}(s,a)}{n^{k,p}(s,a)}}\cdot\sqrt{\sum_{k=1}^{K}\sum_{h=1}^{H}\sum_{(s,a)\in B_k^p}v_{k,h}^{observe,p}(s,a)\mathrm{Var}_{s'\sim p(\cdot|s,a)}\left(V_{h+1}^*(s')\right)}\\
&+HL\sum_{k=1}^{K}\sum_{h=1}^{H}\sum_{(s,a)\in B_k^p}\frac{v_{k,h}^{observe,p}(s,a)}{n^{k,p}(s,a)}\\
\leq& 2\sqrt{L}\cdot\sqrt{8SN\log(KH)}\cdot H\sqrt{KH}+HL\cdot8SN\log(KH)\\
\leq& 8HL\sqrt{KHSN}+8HSNL^2.
\end{aligned}
$$

(iv) Term 4:

$$\sum_{k=1}^{K}\sum_{h=1}^{H}\sum_{(s,a)\in B_k^p} v_{k,h}^{observe,p}(s,a)\left(\hat{p}^k(\cdot|s,a)-p(\cdot|s,a)\right)^\top\left(\bar{V}_{h+1}^k-V_{h+1}^*\right)$$

$$\overset{(a)}{\leq}\sum_{k=1}^{K}\sum_{h=1}^{H}\sum_{(s,a)\in B_k^p} v_{k,h}^{observe,p}(s,a)\sum_{s'}\left(\sqrt{\frac{p(s'|s,a)(1-p(s'|s,a))L}{n^{k,p}(s,a)}}\left|\bar{V}_{h+1}^k(s')-V_{h+1}^*(s')\right|\right.$$

$$\left.+\frac{HL}{n^{k,p}(s,a)}\right)$$

$$\leq\sqrt{L}\sum_{k=1}^{K}\sum_{h=1}^{H}\sum_{(s,a)\in B_k^p} v_{k,h}^{observe,p}(s,a)\sum_{s'}\sqrt{\frac{p(s'|s,a)}{n^{k,p}(s,a)}}\left|\bar{V}_{h+1}^k(s')-V_{h+1}^*(s')\right|$$

$$+HL\sum_{k=1}^{K}\sum_{h=1}^{H}\sum_{(s,a)\in B_k^p}\frac{v_{k,h}^{observe,p}(s,a)}{n^{k,p}(s,a)}$$

$$\leq\sqrt{L}\sum_{k=1}^{K}\sum_{h=1}^{H}\sum_{(s,a)\in B_k^p} v_{k,h}^{observe,p}(s,a)\sqrt{S\sum_{s'}\frac{p(s'|s,a)}{n^{k,p}(s,a)}\left(\bar{V}_{h+1}^k(s')-V_{h+1}^*(s')\right)^2}$$

$$+HL\sum_{k=1}^{K}\sum_{h=1}^{H}\sum_{(s,a)\in B_k^p}\frac{v_{k,h}^{observe,p}(s,a)}{n^{k,p}(s,a)}$$

$$=\sqrt{SL}\sum_{k=1}^{K}\sum_{h=1}^{H}\sum_{(s,a)\in B_k^p} v_{k,h}^{observe,p}(s,a)\sqrt{\frac{p(\cdot|s,a)^\top\left(\bar{V}_{h+1}^k-V_{h+1}^*\right)^2}{n^{k,p}(s,a)}}$$

$$+HL\sum_{k=1}^{K}\sum_{h=1}^{H}\sum_{(s,a)\in B_k^p}\frac{v_{k,h}^{observe,p}(s,a)}{n^{k,p}(s,a)}$$

$$\leq\sqrt{SL}\sqrt{\sum_{k=1}^{K}\sum_{h=1}^{H}\sum_{(s,a)\in B_k^p}\frac{v_{k,h}^{observe,p}(s,a)}{n^{k,p}(s,a)}}\cdot$$

$$\sqrt{\sum_{k=1}^{K}\sum_{h=1}^{H}\sum_{(s,a)\in B_k^p} v_{k,h}^{observe,p}(s,a)\cdot p(\cdot|s,a)^\top\left(\bar{V}_{h+1}^k-V_{h+1}^*\right)^2}$$

$$+HL\sum_{k=1}^{K}\sum_{h=1}^{H}\sum_{(s,a)\in B_k^p}\frac{v_{k,h}^{observe,p}(s,a)}{n^{k,p}(s,a)}$$

$$\leq\sqrt{SL}\cdot\sqrt{8SN\log(KH)}\cdot\sqrt{152192(m+1)H^5S^2NL^3}+HL\cdot8SN\log(KH)$$

$$\leq1104H^2S^2NL^2\sqrt{(m+1)HL}+8HSNL^2$$

$$\leq1112H^2S^2NL^2\sqrt{(m+1)HL},$$

where inequality (a) follows from Eq. (7).

Plugging the above four terms into Eq. (14), we obtain

$$\mathcal{R}(K)=48HL\sqrt{KHSN}+8HL\sqrt{KHSN}+3428H^2SNL^2\sqrt{(m+1)HSL}$$

$$+8HL\sqrt{KHSN}+8HSNL^2+1112H^2S^2NL^2\sqrt{(m+1)HL}+72H^2SAL$$

$$=\tilde{O}\left(H\sqrt{KHSN}\right).$$

$$\square$$

# D ALGORITHM AND PROOFS FOR CASCADING RL WITH BEST POLICY IDENTIFICATION

In this section, we present algorithm `CascadingBPI` and proofs for cascading RL with the best policy identification objective.

## D.1 ALGORITHM `CascadingBPI`

---

**Algorithm 3: `CascadingBPI`**

---

**Input:** $\varepsilon$, $\delta$, $\delta' := \frac{\delta}{7}$. For any $\kappa \in (0,1)$ and $n > 0$, $L^*(\kappa, n) := \log(\frac{HSN}{\kappa}) + \log(8e(n+1))$
and $L(\kappa, n) := \log(\frac{HSN}{\kappa}) + S\log(8e(n+1))$. For any $k > 0$ and $s \in \mathcal{S}$,
$\bar{q}^k(s, a_\perp) = \underline{q}^k(s, a_\perp) := 1$ and $\bar{V}_{H+1}^k(s) = \underline{V}_{H+1}^k(s) := 0$. Initialize
$n^{1,q}(s,a) = n^{1,p}(s,a) := 0$ for any $(s,a) \in \mathcal{S} \times \mathcal{A}$.

1 **for** $k = 1, 2, \ldots, K$ **do**
2    **for** $h = H, H-1, \ldots, 1$ **do**
3      **for** $s \in \mathcal{S}$ **do**
4        **for** $a \in A^{\text{ground}} \setminus \{a_\perp\}$ **do**
5          $b^{k,q}(s,a) \leftarrow \min\{4\sqrt{\frac{\hat{q}^k(s,a)(1-\hat{q}^k(s,a))L^*(\delta',k)}{n^{k,q}(s,a)}} + \frac{15L^*(\delta',k)}{n^{k,q}(s,a)}, \ 1\}$;
6          $\bar{q}^k(s,a) \leftarrow \hat{q}^k(s,a) + b^{k,q}(s,a)$. $\underline{q}^k(s,a) \leftarrow \hat{q}^k(s,a) - b^{k,q}(s,a)$;
7        **for** $a \in A^{\text{ground}}$ **do**
8          $b^{k,pV}(s,a) \leftarrow \min\{4\sqrt{\frac{\text{Var}_{s' \sim \hat{p}^k}(\bar{V}_{h+1}^k(s'))L^*(\delta',k)}{n^{k,p}(s,a)}} + \frac{15H^2 L(\delta',k)}{n^{k,p}(s,a)} +$
         $\frac{2}{H}\hat{p}^k(\cdot|s,a)^\top(\bar{V}_{h+1}^k - \underline{V}_{h+1}^k), \ H\}$;
9          $\bar{w}^k(s,a) \leftarrow r(s,a) + \hat{p}^k(\cdot|s,a)^\top\bar{V}_{h+1}^k + b^{k,pV}(s,a)$;
10      $\bar{V}_h^k(s), \pi_h^k(s) \leftarrow \texttt{BestPerm}(A^{\text{ground}}, \bar{q}^k(s,\cdot), \bar{w}^k(s,\cdot))$;
11      $\bar{V}_h^k(s) \leftarrow \min\{\bar{V}_h^k(s), H\}$. $A' \leftarrow \pi_h^k(s)$;
12      $\underline{V}_h^k(s) \leftarrow \max\{\sum_{i=1}^{|A'|}\prod_{j=1}^{i-1}(1 - \bar{q}^k(s, A'(j)))\underline{q}^k(s, A'(i))(r(s, A'(i)) +$
     $\hat{p}^k(\cdot|s, A'(i))^\top\underline{V}_{h+1}^k - b^{k,pV}(s, A'(i))), \ 0\}$;
13      $G_h^k(s) \leftarrow \min\{\sum_{i=1}^{|A'|}\prod_{j=1}^{i-1}(1 - \bar{q}^k(s, A'(j)))(6Hb^{k,q}(s, A'(i)) +$
     $\underline{q}^k(s, A'(i))(2b^{k,pV}(s, A'(i)) + \hat{p}^k(\cdot|s, A'(i))^\top G_{h+1}^k)), \ H\}$;
14    **if** $G_1^k(s_1) \leq \varepsilon$ **then**
15      **return** $\pi^k$;                     // Estimation error is small enough
16    **for** $h = 1, 2, \ldots, H$ **do**
17      Observe the current state $s_h^k$;          // Take policy $\pi^k$ and observe the trajectory
18      Take action $A_h^k = \pi_h^k(s_h^k)$. $i \leftarrow 1$;
19      **while** $i \leq m$ **do**
20        Observe if $A_h^k(i)$ is clicked or not. Update $n^{k,q}(s_h^k, A_h^k(i))$ and $\hat{q}^k(s_h^k, A_h^k(i))$;
21        **if** $A_h^k(i)$ is clicked **then**
22          Receive reward $r(s_h^k, A_h^k(i))$, and transition to a next state
         $s_{h+1}^k \sim p(\cdot|s_h^k, A_h^k(i))$;
23          $I_{k,h} \leftarrow i$. Update $n^{k,p}(s_h^k, A_h^k(i))$ and $\hat{p}^k(\cdot|s_h^k, A_h^k(i))$;
24          **break while**;                     // Skip subsequent items
25        **else**
26          $i \leftarrow i + 1$;
27      **if** $i = m + 1$ **then**
28        Transition to a next state $s_{h+1}^k \sim p(\cdot|s_h^k, a_\perp)$;          // No item was clicked
29        Update $n^{k,p}(s_h^k, a_\perp)$ and $\hat{p}^k(\cdot|s_h^k, a_\perp)$;

---

Algorithm 3 gives the pseudo-code of `CascadingBPI`. Similar to `CascadingVI`, in each episode, we estimates the attraction and transition for each item independently, and calculates the optimistic attraction probability $\bar{q}^k(s,a)$ and weight $\bar{w}^k(s,a)$ using exploration bonuses. Here $\bar{w}^k(s,a)$ represents the optimistic cumulative reward that can be received if item $a$ is clicked in state $s$. Then, we call the oracle `BestPerm` to compute the maximum optimistic value $\bar{V}_h^k(s)$ and its greedy policy $\pi_h^k(s)$. Furthermore, we build an estimation error $G_h^k(s)$ which upper bounds the difference between $\bar{V}_h^k(s)$ and $V_h^{\pi^k}(s)$ with high confidence. If $G_h^k(s)$ shrinks within the accuracy parameter $\varepsilon$, we output the policy $\pi^k$. Otherwise, we play episode $k$ with policy $\pi^k$, and update the estimates of attraction and transition for the clicked item and the items prior to it.

Employing the efficient oracle `BestPerm`, `CascadingBPI` only maintains the estimated attraction and transition probabilities for each $a \in A^{\text{ground}}$, instead of calculating $\bar{Q}_h^k(s,A)$ for each $A \in \mathcal{A}$ as in a naive adaption of existing RL algorithms (Kaufmann et al., 2021; Ménard et al., 2021). Therefore, `CascadingBPI` achieves a superior computation cost that only depends on $N$, rather than $|\mathcal{A}|$.

### D.2 Sample Complexity for Algorithm `CascadingBPI`

In the following, we prove the sample complexity for algorithm `CascadingBPI`.

#### D.2.1 Concentration

For any $\kappa \in (0,1)$ and $n > 0$, let $L(\kappa, n) := \log(\frac{HSN}{\kappa}) + S\log(8e(n+1))$ and $L^*(\kappa, n) := \log(\frac{HSN}{\kappa}) + \log(8e(n+1))$.

Let event

$$
\mathcal{L} := \left\{ \left| \hat{q}^k(s,a) - q(s,a) \right| \le 4\sqrt{\frac{\hat{q}^k(s,a)(1 - \hat{q}^k(s,a))L^*(\delta',k)}{n^{k,q}(s,a)}} + \frac{15L^*(\delta',k)}{n^{k,q}(s,a)}, \right.
$$

$$
\left| \sqrt{\hat{q}^k(s,a)(1 - \hat{q}^k(s,a))} - \sqrt{q(s,a)(1 - q(s,a))} \right| \le 4\sqrt{\frac{L^*(\delta',k)}{n^{k,q}(s,a)}},
$$

$$
\left. \forall k > 0, \forall h \in [H], \forall (s,a) \in \mathcal{S} \times A^{\text{ground}} \setminus \{a_\perp\} \right\}.
$$

**Lemma 18** (Concentration of Attractive Probability). *It holds that*
$$
\Pr[\mathcal{L}] \ge 1 - 4\delta'.
$$

*Proof of Lemma 18.* Using a similar analysis as that for Lemma 6 and a union bound over $k = 1, \ldots, \infty$, we have that event $\mathcal{L}$ holds with probability $1 - 4\delta'$. □

Let event

$$
\mathcal{M} := \left\{ \text{KL}\left( \hat{p}^k(\cdot|s,a) \| p(\cdot|s,a) \right) \le \frac{L(\delta',k)}{n^{k,p}(s,a)}, \right.
$$

$$
\left| \left( \hat{p}^k(\cdot|s,a) - p(\cdot|s,a) \right)^\top V_{h+1}^* \right| \le 2\sqrt{\frac{\text{Var}_{s' \sim p}\left( V_{h+1}^*(s') \right) L^*(\delta',k)}{n^{k,p}(s,a)}} + \frac{3HL^*(\delta',k)}{n^{k,p}(s,a)},
$$

$$
\left. \forall k > 0, \forall h \in [H], \forall (s,a) \in \mathcal{S} \times A^{\text{ground}} \right\}.
$$

**Lemma 19** (Concentration of Transition Probability). *It holds that*
$$
\Pr[\mathcal{M}] \ge 1 - 3\delta'.
$$
*Furthermore, if event $\mathcal{L} \cap \mathcal{M}$ holds, we have that for any $k > 0$, $h \in [H]$ and $(s,a) \in \mathcal{S} \times A^{\text{ground}}$,*

$$
\left| \left( \hat{p}^k(\cdot|s,a) - p(\cdot|s,a) \right)^\top V_{h+1}^* \right| \le 4\sqrt{\frac{\text{Var}_{s' \sim \hat{p}^k}\left( \bar{V}_{h+1}^k(s') \right) L^*(\delta',k)}{n^{k,p}(s,a)}} + \frac{15H^2 L(\delta',k)}{n^{k,p}(s,a)}
$$

$$+ \frac{2}{H} \hat{p}^k(\cdot|s,a)^\top \left( \bar{V}_{h+1}^k - \underline{V}_{h+1}^k \right).$$

*Proof of Lemma 19.* Following Lemma 3 in (Ménard et al., 2021), we have $\Pr[\mathcal{M}] \geq 1 - 3\delta'$. Using Eq. (22) and (23) in Lemma 22, we have

$$\sqrt{\mathrm{Var}_{s' \sim p} \left( V_{h+1}^*(s') \right) \cdot \frac{L^*(\delta',k)}{n^{k,p}(s,a)}}$$

$$\leq \sqrt{2\mathrm{Var}_{s' \sim \hat{p}^k} \left( V_{h+1}^*(s') \right) \cdot \frac{L^*(\delta',k)}{n^{k,p}(s,a)} + \frac{4H^2 L(\delta',k)}{n^{k,p}(s,a)} \cdot \frac{L^*(\delta',k)}{n^{k,p}(s,a)}}$$

$$\leq \sqrt{4\mathrm{Var}_{s' \sim \hat{p}^k} \left( \bar{V}_{h+1}^k(s') \right) \cdot \frac{L^*(\delta',k)}{n^{k,p}(s,a)} + 4H\hat{p}^k(\cdot|s,a)^\top \left( \bar{V}_{h+1}^k - V_{h+1}^* \right) \cdot \frac{L^*(\delta',k)}{n^{k,p}(s,a)}}$$

$$\quad + \frac{2HL(\delta',k)}{n^{k,p}(s,a)}$$

$$\overset{(a)}{\leq} \sqrt{4\mathrm{Var}_{s' \sim \hat{p}^k} \left( \bar{V}_{h+1}^k(s') \right) \cdot \frac{L^*(\delta',k)}{n^{k,p}(s,a)}} + \sqrt{\frac{1}{H}\hat{p}^k(\cdot|s,a)^\top \left( \bar{V}_{h+1}^k - \underline{V}_{h+1}^k \right) \cdot \frac{4H^2 L^*(\delta',k)}{n^{k,p}(s,a)}}$$

$$\quad + \frac{2HL(\delta',k)}{n^{k,p}(s,a)}$$

$$\leq 2\sqrt{\mathrm{Var}_{s' \sim \hat{p}^k} \left( \bar{V}_{h+1}^k(s') \right) \cdot \frac{L^*(\delta',k)}{n^{k,p}(s,a)}} + \frac{1}{H}\hat{p}^k(\cdot|s,a)^\top \left( \bar{V}_{h+1}^k - \underline{V}_{h+1}^k \right) + \frac{6H^2 L(\delta',k)}{n^{k,p}(s,a)},$$

where inequality (a) uses the induction hypothesis of Lemma 15.

Thus, we have

$$\left| \left( \hat{p}^k(\cdot|s,a) - p(\cdot|s,a) \right)^\top V_{h+1}^* \right| \leq 4\sqrt{\mathrm{Var}_{s' \sim \hat{p}^k} \left( \bar{V}_{h+1}^k(s') \right) \cdot \frac{L^*(\delta',k)}{n^{k,p}(s,a)}}$$

$$+ \frac{2}{H}\hat{p}^k(\cdot|s,a)^\top \left( \bar{V}_{h+1}^k - \underline{V}_{h+1}^k \right) + \frac{15H^2 L(\delta',k)}{n^{k,p}(s,a)}.$$

$\square$

### D.2.2 OPTIMISM AND ESTIMATION ERROR

For any $k > 0$, $h \in [H]$ and $s \in \mathcal{S}$, we define

$$\begin{cases} b^{k,q}(s,a) := \min \left\{ 4\sqrt{\frac{\hat{q}^k(s,a)(1 - \hat{q}^k(s,a))L^*(\delta',k)}{n^{k,q}(s,a)}} + \frac{15L^*(\delta',k)}{n^{k,q}(s,a)}, \; 1 \right\}, \\ \qquad \forall a \in A^{\mathrm{ground}} \setminus \{a_\perp\}, \\ b^{k,q}(s,a_\perp) := 0, \\ b^{k,pV}(s,a) := \min \left\{ 4\sqrt{\frac{\mathrm{Var}_{s' \sim \hat{p}^k} \left( \bar{V}_{h+1}^k(s') \right) L^*(\delta',k)}{n^{k,p}(s,a)}} + \frac{15H^2 L(\delta',k)}{n^{k,p}(s,a)} \\ \qquad + \frac{2}{H}\hat{p}^k(\cdot|s,a)^\top \left( \bar{V}_{h+1}^k - \underline{V}_{h+1}^k \right), \; H \right\}, \; \forall a \in A^{\mathrm{ground}}. \end{cases}$$

For any $k > 0$, $h \in [H]$, $s \in \mathcal{S}$ and $A \in \mathcal{A}$, we define

$$
\begin{cases}
\bar{Q}_h^k(s, A) = \min \Bigg\{ \sum_{i=1}^{|A|} \prod_{j=1}^{i-1}(1 - \bar{q}^k(s, A(j)))\bar{q}^k(s, A(i)) \cdot \\
\qquad\qquad \Big( r(s, A(i)) + \hat{p}^k(\cdot|s, A(i))^\top \bar{V}_{h+1}^k + b^{k,pV}(s, A(i)) \Big), \ H \Bigg\} \\
\pi_h^k(s) = \operatorname*{argmax}_{A \in \mathcal{A}} \bar{Q}_h^k(s, A), \\
\bar{V}_h^k(s) = \bar{Q}_h^k(s, \pi_h^k(s)), \\
\bar{V}_{H+1}^k(s) = 0,
\end{cases}
$$

$$
\begin{cases}
\underline{Q}_h^k(s, A) = \max \Bigg\{ \sum_{i=1}^{|A|} \prod_{j=1}^{i-1}(1 - \bar{q}^k(s, A(j)))\underline{q}^k(s, A(i)) \cdot \\
\qquad\qquad \Big( r(s, A(i)) + \hat{p}^k(\cdot|s, A(i))^\top \underline{V}_{h+1}^k - b^{k,pV}(s, A(i)) \Big), \ 0 \Bigg\}, \\
\underline{V}_h^k(s) = \underline{Q}_h^k(s, \pi_h^k(s)), \\
\underline{V}_{H+1}^k(s) = 0,
\end{cases}
$$

$$
\begin{cases}
G_h^k(s, A) = \sum_{i=1}^{|A|} \prod_{j=1}^{i-1}(1 - \bar{q}^k(s, A(j))) \Bigg( 6H \min \Bigg\{ 4\sqrt{\frac{\hat{q}^k(s,a)(1 - \hat{q}^k(s,a))L^*(\delta', k)}{n^{k,q}(s,a)}} \\
\qquad + \frac{15L^*(\delta', k)}{n^{k,q}(s,a)}, \ 1 \Bigg\} + \underline{q}^k(s, A(i)) \cdot \Bigg( \min \Bigg\{ 8\sqrt{\frac{\operatorname{Var}_{s' \sim \hat{p}^k}\left(\bar{V}_{h+1}^k(s')\right) L^*(\delta', k)}{n^{k,p}(s,a)}} \\
\qquad + \frac{30H^2 L(\delta', k)}{n^{k,p}(s,a)} + \frac{4}{H}\hat{p}^k(\cdot|s,a)^\top G_{h+1}^k, \ 2H \Bigg\} + \hat{p}^k(\cdot|s,a)^\top G_{h+1}^k \Bigg) \Bigg), \\
G_h^k(s) = G_h^k(s, \pi_h^k(s)), \\
G_{H+1}^k(s) = 0.
\end{cases}
$$

**Lemma 20** (Optimism). *Assume that event $\mathcal{L} \cap \mathcal{M}$ holds. Then, for any $k > 0$, $h \in [H]$ and $s \in \mathcal{S}$,*
$$
\bar{V}_h^k(s) \geq V_h^*(s).
$$

*Proof of Lemma 20.* By a similar analysis as that for Lemma 15 with different definitions of $b^{k,q}(s, a), b^{k,pV}(s, a)$ and Lemmas 18, 19, we obtain this lemma.

$\square$

**Lemma 21** (Estimation Error). *Assume that event $\mathcal{K} \cap \mathcal{L} \cap \mathcal{M}$ holds. Then, with probability at least $1 - \delta$, for any $k > 0$, $h \in [H]$ and $s \in \mathcal{S}$,*
$$
\bar{V}_h^k(s) - \min\{\underline{V}_h^k(s), V_h^{\pi^k}(s)\} \leq G_h^k(s).
$$

*Proof of Lemma 21.* For any $k > 0$ and $s \in \mathcal{S}$, it trivially holds that $\bar{V}_{H+1}^k(s) - \min\{\underline{V}_{H+1}^k(s), V_{H+1}^{\pi^k}(s)\} \leq G_{H+1}^k(s)$.

For any $k > 0$, $h \in [H]$ and $(s, A) \in \mathcal{S} \times \mathcal{A}$, if $G_h^k(s, A) = H$, then it trivially holds that $\bar{Q}_{h+1}^k(s, A) - \min\{\underline{Q}_{h+1}^k(s, A), Q_{h+1}^{\pi^k}(s, A)\} \leq G_{h+1}^k(s, A)$.

Otherwise, supposing $\bar{V}_{h+1}^k(s) - \min\{\underline{V}_{h+1}^k(s), V_{h+1}^{\pi^k}(s)\} \leq G_{h+1}^k(s)$, we first prove $\bar{Q}_h^k(s, A) - \underline{Q}_h^k(s, A) \leq G_h^k(s, A)$.

$$\bar{Q}_h^k(s, A) - \underline{Q}_h^k(s, A)$$

$$\leq \sum_{i=1}^{|A|} \prod_{j=1}^{i-1} (1 - \bar{q}^k(s, A(j)))(\underline{q}^k(s, A(i)) + 2b^{k,q}(s, A(i)))\cdot$$

$$\left(r(s, A(i)) + \hat{p}^k(\cdot|s, A(i))^\top \bar{V}_{h+1}^k + b^{k,pV}(s, A(i))\right)$$

$$- \sum_{i=1}^{|A|} \prod_{j=1}^{i-1} (1 - \bar{q}^k(s, A(j)))\underline{q}^k(s, A(i)) \left(r(s, A(i)) + \hat{p}^k(\cdot|s, A(i))^\top \underline{V}_{h+1}^k - b^{k,pV}(s, A(i))\right)$$

$$= \sum_{i=1}^{|A|} \prod_{j=1}^{i-1} (1 - \bar{q}^k(s, A(j)))\left(6Hb^{k,q}(s, A(i))\right.$$

$$\left. + \underline{q}^k(s, A(i)) \left(2b^{k,pV}(s, A(i)) + \hat{p}^k(\cdot|s, A(i))^\top \left(\bar{V}_{h+1}^k - \underline{V}_{h+1}^k\right)\right) \right)$$

$$\leq \sum_{i=1}^{|A|} \prod_{j=1}^{i-1} (1 - \bar{q}^k(s, A(j)))\left(6Hb^{k,q}(s, A(i)) + \underline{q}^k(s, A(i)) \left(2b^{k,pV}(s, A(i)) + \hat{p}^k(\cdot|s, A(i))^\top G_{h+1}^k\right)\right)$$

$$= G_h^k(s, A).$$

Now, we prove $\bar{Q}_h^k(s, A) - Q_h^{\pi^k}(s, A) \leq G_h^k(s, A)$.

$$\bar{Q}_h^k(s, A) - Q_h^{\pi^k}(s, A)$$

$$\leq \sum_{i=1}^{|A|} \prod_{j=1}^{i-1} (1 - \bar{q}^k(s, A(j)))(\underline{q}^k(s, A(i)) + 2b^{k,q}(s, A(i)))\cdot$$

$$\left(r(s, A(i)) + \hat{p}^k(\cdot|s, A(i))^\top \bar{V}_{h+1}^k + b^{k,pV}(s, A(i))\right)$$

$$- \sum_{i=1}^{|A|} \prod_{j=1}^{i-1} (1 - \bar{q}^k(s, A(j)))\underline{q}^k(s, A(i)) \left(r(s, A(i)) + p(\cdot|s, A(i))^\top V_{h+1}^{\pi^k}\right)$$

$$\leq \sum_{i=1}^{|A|} \prod_{j=1}^{i-1} (1 - \bar{q}^k(s, A(j)))\left(6Hb^{k,q}(s, A(i)) + \underline{q}^k(s, A(i))\left(b^{k,pV}(s, A(i))\right.\right.$$

$$\left.\left. + \hat{p}^k(\cdot|s, A(i))^\top \bar{V}_{h+1}^k - p(\cdot|s, A(i))^\top V_{h+1}^{\pi^k}\right)\right)$$

$$\leq \sum_{i=1}^{|A|} \prod_{j=1}^{i-1} (1 - \bar{q}^k(s, A(j)))\left(6Hb^{k,q}(s, A(i)) + \underline{q}^k(s, A(i))\left(b^{k,pV}(s, A(i))\right.\right.$$

$$+ \hat{p}^k(\cdot|s, A(i))^\top \left(\bar{V}_{h+1}^k - V_{h+1}^{\pi^k}\right) + \left(\hat{p}^k(\cdot|s, A(i)) - p(\cdot|s, A(i))\right)^\top V_{h+1}^*$$

$$\left.\left. + \left(p(\cdot|s, A(i)) - \hat{p}^k(\cdot|s, A(i))\right)^\top \left(V_{h+1}^* - V_{h+1}^{\pi^k}\right)\right)\right) \tag{16}$$

In addition, for any $(s, a) \in \mathcal{S} \times A^{\text{ground}}$, we have

$$\left(p(\cdot|s, a) - \hat{p}^k(\cdot|s, a)\right)^\top \left(V_{h+1}^* - V_{h+1}^{\pi^k}\right)$$

$$\leq 2\sqrt{\frac{\text{Var}_{s' \sim p}\left(V_{h+1}^*(s') - V_{h+1}^{\pi^k}(s')\right) L(\delta', k)}{n^{k,p}(s, a)}} + \frac{3HL(\delta', k)}{n^{k,p}(s, a)}$$

$$\leq 2\sqrt{\frac{L(\delta', k)}{n^{k,p}(s, a)} \cdot \left(2\text{Var}_{s' \sim \hat{p}^k}\left(V_{h+1}^*(s') - V_{h+1}^{\pi^k}(s')\right) + \frac{4H^2 L(\delta', k)}{n^{k,p}(s, a)}\right)} + \frac{3HL(\delta', k)}{n^{k,p}(s, a)}$$

$$\leq 2\sqrt{\frac{L(\delta',k)}{n^{k,p}(s,a)}\cdot\left(2H\hat{p}^k(\cdot|s,a)^\top\left(V^*_{h+1}-V^{\pi^k}_{h+1}\right)+\frac{4H^2L(\delta',k)}{n^{k,p}(s,a)}\right)}+\frac{3HL(\delta',k)}{n^{k,p}(s,a)}$$

$$\leq 2\sqrt{\frac{2H^2L(\delta',k)}{n^{k,p}(s,a)}\cdot\frac{1}{H}\hat{p}^k(\cdot|s,a)^\top\left(V^*_{h+1}-V^{\pi^k}_{h+1}\right)}+\frac{4HL(\delta',k)}{n^{k,p}(s,a)}+\frac{3HL(\delta',k)}{n^{k,p}(s,a)}$$

$$\leq\frac{2}{H}\hat{p}^k(\cdot|s,a)^\top\left(V^*_{h+1}-V^{\pi^k}_{h+1}\right)+\frac{11H^2L(\delta',k)}{n^{k,p}(s,a)}\tag{17}$$

Plugging Eq. (17) into Eq. (16), we have

$$\bar{Q}^k_h(s,A)-Q^{\pi^k}_h(s,A)$$

$$\leq\sum_{i=1}^{|A|}\prod_{j=1}^{i-1}(1-\bar{q}^k(s,A(j)))\left(6H\left(4\sqrt{\frac{\hat{q}^k(s,a)(1-\hat{q}^k(s,a))L^*(\delta',k)}{n^{k,q}(s,a)}}+\frac{15L^*(\delta',k)}{n^{k,q}(s,a)}\right)\right.$$

$$+\underline{q}^k(s,A(i))\left(4\sqrt{\frac{\mathrm{Var}_{s'\sim\hat{p}^k}\left(\bar{V}^k_{h+1}(s')\right)L^*(\delta',k)}{n^{k,p}(s,a)}}\right.$$

$$+\frac{15H^2L(\delta',k)}{n^{k,p}(s,a)}+\frac{2}{H}\hat{p}^k(\cdot|s,a)^\top\left(\bar{V}^k_{h+1}-\underline{V}^k_{h+1}\right)$$

$$+\hat{p}^k(\cdot|s,A(i))^\top\left(\bar{V}^k_{h+1}-V^{\pi^k}_{h+1}\right)+2\sqrt{\frac{\mathrm{Var}_{s'\sim\hat{p}^k}\left(\bar{V}^k_{h+1}(s')\right)L^*(\delta',k)}{n^{k,p}(s,a)}}+\frac{3HL(\delta',k)}{n^{k,p}(s,a)}$$

$$\left.\left.+\frac{2}{H}\hat{p}^k(\cdot|s,a)^\top\left(V^*_{h+1}-V^{\pi^k}_{h+1}\right)+\frac{11H^2L(\delta',k)}{n^{k,p}(s,a)}\right)\right)$$

$$\leq\sum_{i=1}^{|A|}\prod_{j=1}^{i-1}(1-\bar{q}^k(s,A(j)))\left(6H\left(4\sqrt{\frac{\hat{q}^k(s,a)(1-\hat{q}^k(s,a))L^*(\delta',k)}{n^{k,q}(s,a)}}+\frac{15L^*(\delta',k)}{n^{k,q}(s,a)}\right)\right.$$

$$+\underline{q}^k(s,A(i))\left(6\sqrt{\frac{\mathrm{Var}_{s'\sim\hat{p}^k}\left(\bar{V}^k_{h+1}(s')\right)L^*(\delta',k)}{n^{k,p}(s,a)}}\right.$$

$$\left.\left.+\frac{29H^2L(\delta',k)}{n^{k,p}(s,a)}+\left(1+\frac{4}{H}\right)\hat{p}^k(\cdot|s,a)^\top G^k_{h+1}\right)\right)$$

By the clipping bound in Eq. (16), we have

$$\bar{Q}^k_h(s,A)-Q^{\pi^k}_h(s,A)$$

$$\leq\sum_{i=1}^{|A|}\prod_{j=1}^{i-1}(1-\bar{q}^k(s,A(j)))\left(6H\min\left\{4\sqrt{\frac{\hat{q}^k(s,a)(1-\hat{q}^k(s,a))L^*(\delta',k)}{n^{k,q}(s,a)}}+\frac{15L^*(\delta',k)}{n^{k,q}(s,a)},1\right\}\right.$$

$$+\underline{q}^k(s,A(i))\cdot\left(\min\left\{6\sqrt{\frac{\mathrm{Var}_{s'\sim\hat{p}^k}\left(\bar{V}^k_{h+1}(s')\right)L^*(\delta',k)}{n^{k,p}(s,a)}}+\frac{29H^2L(\delta',k)}{n^{k,p}(s,a)}\right.\right.$$

$$\left.\left.\left.+\frac{4}{H}\hat{p}^k(\cdot|s,a)^\top G^k_{h+1},\,2H\right\}+\hat{p}^k(\cdot|s,a)^\top G^k_{h+1}\right)\right)$$

$$\leq G^k_{h+1}(s,A).$$

Then, we have

$$\bar{V}^k_h(s)-\min\{\underline{V}^k_h(s),V^{\pi^k}_h(s)\}=\bar{Q}^k_h(s,\pi^k_h(s))-\min\{\underline{Q}^k_h(s,\pi^k_h(s)),Q^{\pi^k}_h(s,\pi^k_h(s))\}$$

$$\leq G^k_h(s,\pi^k_h(s))$$

$$=G^k_h(s),$$

which completes the proof. □

### D.2.3 PROOF OF THEOREM 2

*Proof of Theorem 2.* Recall that $\delta' := \frac{\delta}{7}$. From Lemmas 10, 18 and 19, we have $\Pr[\mathcal{K} \cap \mathcal{L} \cap \mathcal{M}] \geq 1 - 7\delta' = 1 - \delta$. Below we assume that event $\mathcal{K} \cap \mathcal{L} \cap \mathcal{M}$ holds, and then prove the correctness and sample complexity.

First, we prove the correctness. Using Lemma 20 and 21, we have that the output policy $\pi^{K+1}$ satisfies that

$$V_1^*(s_1) - V_1^{\pi^{K+1}}(s_1) \leq \bar{V}_1^{K+1}(s_1) - V_1^{\pi^{K+1}}(s_1) \leq G_1^{K+1}(s_1) \leq \varepsilon,$$

which indicates that policy $\pi^{K+1}$ is $\varepsilon$-optimal.

Next, we prove sample complexity.

For any $k > 0$, $h \in [H]$, $s \in \mathcal{S}$ and $A \in \mathcal{A}$,

$$
\begin{aligned}
G_h^k(s, A) &\leq \sum_{i=1}^{|A|} \prod_{j=1}^{i-1} (1 - \bar{q}^k(s, A(j))) \Bigg( 6H \Bigg( 4\sqrt{\frac{\hat{q}^k(s,a)(1 - \hat{q}^k(s,a))L^*(\delta', k)}{n^{k,q}(s,a)}} + \frac{15L^*(\delta', k)}{n^{k,q}(s,a)} \Bigg) \\
&\quad + \underline{q}^k(s, A(i)) \Bigg( 8\sqrt{\frac{\mathrm{Var}_{s' \sim \hat{p}^k}\left(\bar{V}_{h+1}^k(s')\right) L^*(\delta', k)}{n^{k,p}(s, A(i))}} + \frac{30H^2 L(\delta', k)}{n^{k,p}(s, A(i))} \\
&\quad + \left(1 + \frac{4}{H}\right) \hat{p}^k(\cdot | s, A(i))^\top G_{h+1}^k \Bigg) \Bigg) \\
&\leq \sum_{i=1}^{|A|} \prod_{j=1}^{i-1} (1 - q(s, A(j))) \Bigg( 24H \sqrt{\frac{\hat{q}^k(s,a)(1 - \hat{q}^k(s,a))L^*(\delta', k)}{n^{k,q}(s,a)}} + \frac{90H L^*(\delta', k)}{n^{k,q}(s,a)} \\
&\quad + q(s, A(i)) \Bigg( 8\sqrt{\frac{\mathrm{Var}_{s' \sim \hat{p}^k}\left(\bar{V}_{h+1}^k(s')\right) L^*(\delta', k)}{n^{k,p}(s, A(i))}} + \frac{30H^2 L(\delta', k)}{n^{k,p}(s, A(i))} \\
&\quad + \left(1 + \frac{4}{H}\right) \hat{p}^k(\cdot | s, A(i))^\top G_{h+1}^k \Bigg) \Bigg) \\
&\overset{(a)}{\leq} \sum_{i=1}^{|A|} \prod_{j=1}^{i-1} (1 - q(s, A(j))) \Bigg( 24H \sqrt{\frac{q(s,a)(1 - q(s,a))L^*(\delta', k)}{n^{k,q}(s,a)}} + \frac{186H L^*(\delta', k)}{n^{k,q}(s,a)} \\
&\quad + q(s, A(i)) \Bigg( 8\sqrt{\frac{\mathrm{Var}_{s' \sim \hat{p}^k}\left(\bar{V}_{h+1}^k(s')\right) L^*(\delta', k)}{n^{k,p}(s, A(i))}} + \frac{30H^2 L(\delta', k)}{n^{k,p}(s, A(i))} \\
&\quad + \left(1 + \frac{4}{H}\right) p(\cdot | s, A(i))^\top G_{h+1}^k + \left(1 + \frac{4}{H}\right) \left(\hat{p}^k(\cdot | s, A(i)) - p(\cdot | s, A(i))\right)^\top G_{h+1}^k \Bigg) \Bigg),
\end{aligned}
$$

$$(18)$$

where inequality (a) comes from Eq. (15).

Using Eq. (21) in Lemma 22, we have

$$
\begin{aligned}
\left(\hat{p}^k(\cdot | s, A(i)) - p(\cdot | s, A(i))\right)^\top G_{h+1}^k &\leq \sqrt{\frac{2\mathrm{Var}_{s' \sim p}\left(G_{h+1}^k(s')\right) L(\delta, k)}{n^{k,p}(s, A(i))}} + \frac{2HL(\delta, k)}{3n^{k,p}(s, A(i))} \\
&\leq \sqrt{\frac{1}{H} p(\cdot | s, A(i))^\top G_{h+1}^k \cdot \frac{2H^2 L(\delta, k)}{n^{k,p}(s, A(i))}} + \frac{2HL(\delta, k)}{3n^{k,p}(s, A(i))} \\
&\leq \frac{1}{H} p(\cdot | s, A(i))^\top G_{h+1}^k + \frac{3H^2 L(\delta, k)}{n^{k,p}(s, A(i))}
\end{aligned}
$$

$$(19)$$

According to Eqs. (22) and (23) in Lemma 22, we have

$$
\sqrt{\mathrm{Var}_{s'\sim\hat{p}^k}\left(\bar{V}_{h+1}^k(s')\right)} \leq \sqrt{2\mathrm{Var}_{s'\sim p}\left(\bar{V}_{h+1}^k(s')\right) + \frac{4H^2 L(\delta,k)}{n^{k,p}(s,a)}}
$$

$$
\leq \sqrt{4\mathrm{Var}_{s'\sim p}\left(V_{h+1}^{\pi^k}(s')\right) + 4Hp(\cdot|s,a)^\top\left(\bar{V}_{h+1}^k - V_{h+1}^{\pi^k}\right) + \frac{4H^2 L(\delta,k)}{n^{k,p}(s,a)}}
$$

$$
\leq \sqrt{4\mathrm{Var}_{s'\sim p}\left(V_{h+1}^{\pi^k}(s')\right) + 4Hp(\cdot|s,a)^\top\left(\bar{V}_{h+1}^k - \underline{V}_{h+1}^k\right) + \frac{4H^2 L(\delta,k)}{n^{k,p}(s,a)}}
$$

$$
\leq \sqrt{4\mathrm{Var}_{s'\sim p}\left(V_{h+1}^{\pi^k}(s')\right)} + \sqrt{4Hp(\cdot|s,a)^\top G_{h+1}^k} + \sqrt{\frac{4H^2 L(\delta,k)}{n^{k,p}(s,a)}}
$$

Then,

$$
\sqrt{\frac{\mathrm{Var}_{s'\sim\hat{p}^k}\left(\bar{V}_{h+1}^k(s')\right)L^*(\delta',k)}{n^{k,p}(s,a)}} \leq \sqrt{\frac{4\mathrm{Var}_{s'\sim p}\left(V_{h+1}^{\pi^k}(s')\right)L^*(\delta',k)}{n^{k,p}(s,a)}}
$$

$$
+ \sqrt{\frac{4H^2 L^*(\delta',k)}{n^{k,p}(s,a)}\cdot\frac{1}{H}p(\cdot|s,a)^\top G_{h+1}^k} + \frac{2HL(\delta',k)}{n^{k,p}(s,a)}
$$

$$
\leq 2\sqrt{\frac{\mathrm{Var}_{s'\sim p}\left(V_{h+1}^{\pi^k}(s')\right)L^*(\delta',k)}{n^{k,p}(s,a)}} + \frac{6H^2 L(\delta',k)}{n^{k,p}(s,a)}
$$

$$
+ \frac{1}{H}p(\cdot|s,a)^\top G_{h+1}^k \tag{20}
$$

Plugging Eqs. (19) and (20) into Eq. (18), we obtain

$$
G_h^k(s,A) \leq \sum_{i=1}^{|A|}\prod_{j=1}^{i-1}(1-q(s,A(j)))\Bigg(24H\sqrt{\frac{q(s,a)(1-q(s,a))L^*(\delta',k)}{n^{k,q}(s,a)}} + \frac{186HL^*(\delta',k)}{n^{k,q}(s,a)}
$$

$$
+ q(s,A(i))\Bigg(16\sqrt{\frac{\mathrm{Var}_{s'\sim p}\left(V_{h+1}^{\pi^k}(s')\right)L^*(\delta',k)}{n^{k,p}(s,a)}} + \frac{42H^2 L(\delta',k)}{n^{k,p}(s,a)} + \frac{8}{H}p(\cdot|s,a)^\top G_{h+1}^k
$$

$$
+ \frac{30H^2 L(\delta',k)}{n^{k,p}(s,A(i))} + \left(1+\frac{4}{H}\right)p(\cdot|s,A(i))^\top G_{h+1}^k
$$

$$
+ \left(1+\frac{4}{H}\right)\cdot\left(\frac{1}{H}p(\cdot|s,A(i))^\top G_{h+1}^k + \frac{3H^2 L(\delta,k)}{n^{k,p}(s,A(i))}\right)\Bigg)\Bigg)
$$

$$
\leq \sum_{i=1}^{|A|}\prod_{j=1}^{i-1}(1-q(s,A(j)))\Bigg(24H\sqrt{\frac{q(s,a)(1-q(s,a))L^*(\delta',k)}{n^{k,q}(s,a)}} + \frac{186HL^*(\delta',k)}{n^{k,q}(s,a)}
$$

$$
+ q(s,A(i))\Bigg(16\sqrt{\frac{\mathrm{Var}_{s'\sim p}\left(V_{h+1}^{\pi^k}(s')\right)L^*(\delta',k)}{n^{k,p}(s,a)}} + \frac{87H^2 L(\delta',k)}{n^{k,p}(s,A(i))}
$$

$$
+ \left(1+\frac{17}{H}\right)p(\cdot|s,A(i))^\top G_{h+1}^k\Bigg)\Bigg)
$$

Unfolding the above inequality over $h = 1, 2, \ldots, H$, we have

$$
G_1^k(s_1) \leq \sum_{h=1}^{H}\sum_{(s,a)\in\mathcal{S}\times A^{\mathrm{ground}}\backslash\{a_\perp\}}v_{k,h}^{observe,q}(s,a)\Bigg(24H\sqrt{\frac{q(s,a)(1-q(s,a))L^*(\delta',k)}{n^{k,q}(s,a)}} + \frac{186HL^*(\delta',k)}{n^{k,q}(s,a)}\Bigg)
$$

$$
+ \sum_{h=1}^{H}\sum_{(s,a)\in\mathcal{S}\times A^{\mathrm{ground}}}v_{k,h}^{observe,p}(s,a)\Bigg(16e^{17}\sqrt{\frac{\mathrm{Var}_{s'\sim p}\left(V_{h+1}^{\pi^k}(s')\right)L^*(\delta',k)}{n^{k,p}(s,a)}}
$$

$$+ \frac{87e^{17}H^2L(\delta',k)}{n^{k,p}(s,a)} \Bigg)$$

$$\leq 24H \sum_{h=1}^{H} \sum_{(s,a)\in\mathcal{S}\times A^{\text{ground}}\backslash\{a_\perp\}} v_{k,h}^{observe,q}(s,a) \sqrt{\frac{q(s,a)(1-q(s,a))L^*(\delta',k)}{n^{k,q}(s,a)}}$$

$$+ 186H \sum_{h=1}^{H} \sum_{(s,a)\in\mathcal{S}\times A^{\text{ground}}\backslash\{a_\perp\}} v_{k,h}^{observe,q}(s,a) \frac{L^*(\delta',k)}{n^{k,q}(s,a)}$$

$$+ 16e^{17} \sqrt{\sum_{h=1}^{H} \sum_{(s,a)\in\mathcal{S}\times A^{\text{ground}}} v_{k,h}^{observe,p}(s,a)\text{Var}_{s'\sim p}\left(V_{h+1}^{\pi^k}(s')\right)} \cdot$$

$$\sqrt{\sum_{h=1}^{H} \sum_{(s,a)\in\mathcal{S}\times A^{\text{ground}}} v_{k,h}^{observe,p}(s,a) \frac{L^*(\delta',k)}{n^{k,p}(s,a)}}$$

$$+ 87e^{17} \sum_{h=1}^{H} \sum_{(s,a)\in\mathcal{S}\times A^{\text{ground}}} v_{k,h}^{observe,p}(s,a) \frac{H^2 L(\delta',k)}{n^{k,p}(s,a)}$$

$$\leq 24H \sum_{h=1}^{H} \sum_{(s,a)\in\mathcal{S}\times A^{\text{ground}}\backslash\{a_\perp\}} v_{k,h}^{observe,q}(s,a) \sqrt{\frac{q(s,a)(1-q(s,a))L^*(\delta',k)}{n^{k,q}(s,a)}}$$

$$+ 186H \sum_{h=1}^{H} \sum_{(s,a)\in\mathcal{S}\times A^{\text{ground}}\backslash\{a_\perp\}} v_{k,h}^{observe,q}(s,a) \frac{L^*(\delta',k)}{n^{k,q}(s,a)}$$

$$+ 16e^{17}H\sqrt{H} \sqrt{\sum_{h=1}^{H} \sum_{(s,a)\in\mathcal{S}\times A^{\text{ground}}} v_{k,h}^{observe,p}(s,a) \frac{L^*(\delta',k)}{n^{k,p}(s,a)}}$$

$$+ 87e^{17}H^2 \sum_{h=1}^{H} \sum_{(s,a)\in\mathcal{S}\times A^{\text{ground}}} v_{k,h}^{observe,p}(s,a) \frac{L(\delta',k)}{n^{k,p}(s,a)}.$$

Let $K$ denote the number of episodes that algorithm `CascadingBPI` plays. According to the stopping rule of algorithm `CascadingBPI`, we have that for any $k \leq K$,

$$\varepsilon < G_1^k(s_1).$$

Summing the above inequality over $k = 1, \ldots, K$, dividing $(s,a)$ by sets $B_k^q$ and $B_k^p$, and using the clipping construction of $G_h^k(s,A)$, we obtain

$$K\varepsilon < 24H \sum_{k=1}^{K} \sum_{h=1}^{H} \sum_{(s,a)\in B_k^q} v_{k,h}^{observe,q}(s,a) \sqrt{\frac{q(s,a)(1-q(s,a))L^*(\delta',k)}{n^{k,q}(s,a)}}$$

$$+ 186H \sum_{k=1}^{K} \sum_{h=1}^{H} \sum_{(s,a)\in B_k^q} v_{k,h}^{observe,q}(s,a) \frac{L^*(\delta',k)}{n^{k,q}(s,a)}$$

$$+ 16e^{17}H\sqrt{H} \sum_{k=1}^{K} \sqrt{\sum_{h=1}^{H} \sum_{(s,a)\in B_k^p} v_{k,h}^{observe,p}(s,a) \frac{L^*(\delta',k)}{n^{k,p}(s,a)}}$$

$$+ 87e^{17}H^2 \sum_{k=1}^{K} \sum_{h=1}^{H} \sum_{(s,a)\in B_k^p} v_{k,h}^{observe,p}(s,a) \frac{L(\delta',k)}{n^{k,p}(s,a)}$$

$$+ \sum_{k=1}^{K} \sum_{h=1}^{H} \sum_{(s,a)\notin B_k^q} v_{k,h}^{observe,q}(s,a) 6H + \sum_{k=1}^{K} \sum_{h=1}^{H} \sum_{(s,a)\notin B_k^p} v_{k,h}^{observe,p}(s,a) 2H$$

$$\leq 24H \sqrt{\sum_{k=1}^{K} \sum_{h=1}^{H} \sum_{(s,a)\in B_k^q} v_{k,h}^{observe,q}(s,a) q(s,a)} \sqrt{\sum_{k=1}^{K} \sum_{h=1}^{H} \sum_{(s,a)\in B_k^q} \frac{v_{k,h}^{observe,q}(s,a) L^*(\delta',k)}{n^{k,q}(s,a)}}$$

$$+ 186H \sum_{k=1}^{K} \sum_{h=1}^{H} \sum_{(s,a)\in B_k^q} v_{k,h}^{observe,q}(s,a) \frac{L^*(\delta',k)}{n^{k,q}(s,a)}$$

$$+ 16e^{17}H\sqrt{KH} \sqrt{\sum_{k=1}^{K} \sum_{h=1}^{H} \sum_{(s,a)\in B_k^p} v_{k,h}^{observe,p}(s,a) \frac{L^*(\delta',k)}{n^{k,p}(s,a)}}$$

$$+ 87e^{17}H^2 \sum_{k=1}^{K} \sum_{h=1}^{H} \sum_{(s,a)\in B_k^p} v_{k,h}^{observe,p}(s,a) \frac{L(\delta',k)}{n^{k,p}(s,a)} + 64H^2 SA \log\left(\frac{HSN}{\delta'}\right)$$

$$\overset{(a)}{\leq} 96H\sqrt{KHSA \log(KH) L^*(\delta',K)} + 1488HSA \log(KH) L^*(\delta',K)$$
$$+ 64e^{17}H\sqrt{KHSA \log(KH) L^*(\delta',K)} + 696e^{17}H^2 SA \log(KH) L(\delta',K)$$
$$+ 64H^2 SA \log\left(\frac{HSN}{\delta'}\right)$$

$$\leq 160e^{17}H\sqrt{HSA}\sqrt{K\left(\log\left(\frac{HSN}{\delta'}\right)\log(KH) + \log^2(8eH(K+1))\right)}$$
$$+ 2248e^{17}H^2 SA \left(\log\left(\frac{HSN}{\delta'}\right)\log\left(\frac{KHSN}{\delta'}\right) + S\log^2\left(\frac{8eHSN(K+1)}{\delta'}\right)\right),$$

where inequality (a) uses Lemma 9.

Thus, we have

$$K < \frac{160e^{17}H\sqrt{HSA}}{\varepsilon}\sqrt{K\left(\log\left(\frac{HSN}{\delta'}\right)\log(KH) + \log^2(8eH(K+1))\right)}$$
$$+ \frac{2248e^{17}H^2 SA}{\varepsilon}\left(\log\left(\frac{HSN}{\delta'}\right)\log\left(\frac{KHSN}{\delta'}\right) + S\log^2\left(\frac{8eHSN(K+1)}{\delta'}\right)\right)$$

Using Lemma 23 with $\tau = K+1$, $\alpha = \frac{8eHSN}{\delta'}$, $A = \log\left(\frac{HSN}{\delta'}\right)$, $B = 1$, $C = \frac{160e^{17}H\sqrt{HSA}}{\varepsilon}$, $D = \frac{2248e^{17}H^2 SA}{\varepsilon}$ and $E = S$, we have

$$K + 1 \leq O\left(\frac{H^3 SA}{\varepsilon^2}\log\left(\frac{HSN}{\delta}\right)C_1^2 + \left(\frac{H^2 SA}{\varepsilon} + \frac{H^2\sqrt{H}SA}{\varepsilon\sqrt{\varepsilon}}\right)\left(\log\left(\frac{HSN}{\delta}\right) + S\right)C_1^2\right),$$

where

$$C_1 = O\left(\log\left(\frac{HSN}{\delta\varepsilon}\left(\log\left(\frac{HSN}{\delta}\right) + S\right)\right)\right).$$

Therefore, we obtain Theorem 2. $\qquad\square$

## E  TECHNICAL LEMMAS

In this section, we present two useful technical lemmas.

**Lemma 22** (Lemmas 10, 11, 12 in (Ménard et al., 2021))**.** *Let $p_1$ and $p_2$ be two distributions on $\mathcal{S}$ such that* $\mathrm{KL}(p_1, p_2) \leq \alpha$. *Let $f$ and $g$ be two functions defined on $\mathcal{S}$ such that for any $s \in \mathcal{S}$,*

$0 \leq f(s), g(s) \leq b$. *Then,*

$$\left| p_1^\top f - p_2^\top f \right| \leq \sqrt{2\text{Var}_{p_2}(f)\alpha} + \frac{2}{3}b\alpha, \tag{21}$$

$$\text{Var}_{p_2}(f) \leq 2\text{Var}_{p_1}(f) + 4b^2\alpha, \quad \text{Var}_{p_1}(f) \leq 2\text{Var}_{p_2}(f) + 4b^2\alpha, \tag{22}$$

$$\text{Var}_{p_1}(f) \leq 2\text{Var}_{p_1}(g) + 2b \cdot p_1^\top |f - g|. \tag{23}$$

**Lemma 23.** *Let $A$, $B$, $C$, $D$, $E$ and $\alpha$ be positive scalars such that $1 \leq B \leq E$ and $\alpha \geq e$. If $\tau \geq 0$ satisfies*

$$\tau \leq C\sqrt{\tau \left(A\log(\alpha\tau) + B\log^2(\alpha\tau)\right)} + D\left(A\log(\alpha\tau) + E\log^2(\alpha\tau)\right),$$

*then*

$$\tau \leq C^2(A+B)C_1^2 + \left(D + 2\sqrt{D}C\right)(A+E)C_1^2 + 1,$$

*where*

$$C_1 = \frac{8}{5}\log\left(11\alpha^2(A+E)(C+D)\right).$$

