# OpenReview forum: "Cascading Reinforcement Learning"
_ICLR.cc/2024/Conference — ICLR 2024 spotlight_

### Official Review · Reviewer_6goX · 2023-10-23

**Soundness:** 3 good
**Presentation:** 2 fair
**Contribution:** 3 good
**Rating:** 6
**Confidence:** 3

**Summary:**

The paper tackles the Cascading Bandits problem over a Markov Decision Process, where time is divided into episode of fixed length $H$, and the attractiveness of items (which determined the expected state transitions and rewards) is dependent on a state at the current step in the episode. At each step in the episode, the system gets to observe the current state and proposed a ranked list of items and the environment provides a click on one of the items according to the cascading click model. For this problem, the paper provides a method for computing the optimal offline strategy when all distributions are known and an introduces the CascadingVI algorithm along with a regret bound of $O(H\sqrt{HSNK})$, where $H$ is the episode length, $S$ is the number of states, $K$ the number of episodes and $N$ is the number of individual items available to rank. The performance of the proposed algorithm is compared numerically with a baseline algorithm produced by adapting an existing algorithm to the combinatorial space, referred to here as AdaptRM.

**Strengths:**

- generalizes the cascading bandit problem to MDPs
- provides numerical experiments, even though, in my opinion the baseline is too weak
- provides theoretical guarantees and proofs of correctness
- generally well written, despite the convoluted algorithms I could navigate my way around the paper.

**Weaknesses:**

- The significance of the contribution is not substantial enough to justify acceptance into the venue in my opinion. One of the main contributions is providing an algorithm that does not scale in complexity (both sample or computational) with the space of all possible rankings. I believe we have well-known recipes for how such algorithms can be formulated since the papers on Cascading Bandits from 2015 (Kveton et. al, Combes et al.). Providing a solution scaling with the number of items is expected in my opinion and not a surprising contribution (we know that estimating individual CTRs is enough to construct optimal rankings, we don't need to keep statistics on each individual ranking). This also leads me to believe the baseline used in the numerical experiments is weak.
- The contribution related to the generalization to MDPs is not opening a significant amount of new doors. In my opinion, this generalization does not provide further theoretical insights and has limited additional practical relevance.
- The experimental section use a setting that is too simplistic (very few items and states) and a weak baseline, in my opinion, thus not being representative enough of what we can expect from this algorithm in practice.

**Questions:**

On line 8 in Algorithm 2, what is the complexity of computing $E_{s'\sim p^k}[ .... ]$?

Regarding BestPerm, to me it feels like the biggest hurdle is the computational complexity of computing the value functions $w$, which BestPerm assumes is an input. How does your proposed algorithm navigate this difficulty?

I would also like to see more intuition regarding the exploration bonus $b^{k, pV}(s, a)$, in particular: how is the optimism of $\overline{V}^k_h(s)$ connected to the uncertainty in the estimates of state transition probabilities $p$? The algorithm is already fairly convoluted and it is hard to see how we can construct optimistic estimates of the value functions in light of estimation noise in the state transition probabilities. I believe it is very valuable to clearly articulate such insights in the main body of the paper.

---

> ### Author Response · Authors · 2023-11-17
> **Response to Reviewer 6goX (Part 1/3)**
>
> Thank you very much for your time and effort in reviewing our paper! We have incorporated your comments in the revised version, and highlighted our revision in *red* font.
>
>
> **1. Contribution of our Generalization to Cascading MDPs**
>
> We emphasize that our generalization to cascading MDPs is very different from prior cascading bandit works [Kveton et al. 2015; Combes et al. 2015], since we face a unique challenge on **how to efficiently compute the optimal permutation**.
> To tackle this challenge, we make use of special properties of the objective function for cascading MDPs, and develop a novel and efficient oracle BestPerm using a dynamic programming approach for combinatorial optimization.
>
> Specifically, prior cascading bandit works [Kveton et al. 2015; Combes et al. 2015] consider finding the optimal subset of items as follows:
> $$
> \max_{A \in \mathcal{A}} \quad f(A) = \sum_{i=1}^{|A|} \prod_{j=1}^{i-1} \Big( 1-q(A(j)) \Big) q(A(i)) = 1- \prod_{i=1}^{|A|} \Big( 1-q(A(i)) \Big) . \quad Eq. (i)
> $$
> This objective function only depends on the attraction probabilities of items $q(A(i))$. Thus, prior works **only need to select $m$ items with the maximum attraction probabilities**, which do not involve computational difficulty.
>
> In contrast, our cascading MDP problem considers  finding the optimal ordered subset of items as follows (for step $h=H,H-1,\dots,1$):
> $$
> \max_{A \in \mathcal{A}} \quad Q^{*}_h(s,A)
> $$
>
> $$
> \ \qquad = \sum_{i=1}^{|A|} \prod_{j=1}^{i-1} \Big( 1-q(s,A(j)) \Big) q(s,A(i)) \Big( r(s,A(i))+p(\cdot|s,A(i))^\top V^{*}_{h+1} \Big) .
> $$
>
> Our objective function depends on both the attraction probabilities $q(s,A(i))$ and the expected future rewards $r(s,A(i))+p(\cdot|s,A(i))^\top V^{*}_{h+1}$.
>
> This requires us to **solve a combinatorial optimization problem** for each $s \in \mathcal{S}$ as follows:
> $$
> \max_{A \in \mathcal{A}} \quad f(A,q,w) =  \sum_{i=1}^{|A|} \prod_{j=1}^{i-1} \Big( 1-q(A(j)) \Big) q(A(i)) w(A(i)) , \quad Eq. (ii)
> $$
> where $q(a):=q(s,a)$ and $w(a):=r(s,a)+p(\cdot|s,a)^\top V^{*}_{h+1}$
>
> (in the online RL process, we can have the estimates of $r(s,a)$, $p(s'|s,a)$ and $V^*_{h+1}(s')$).
> This optimization involves two attributes of items $q(a)$ and $w(a)$, instead of just depending on $q(a)$, which is far more challenging to efficiently compute than Eq. (i).
>
> To overcome this computational difficulty, we first analyze the special properties of $f(A,q,w)$, including the fact that sorting items in descending order of $w(a)$ gives the optimal permutation (Section 4.1). Then, based on these properties, we design an efficient oracle BestPerm to solve Eq. (ii), building upon **a novel dynamic programming approach** for finding the best subset from sorted items (Section 4.2):
> $$
> F[i][k] = \max \\{ F[i+1][k],\  q(a_i)  w(a_i) +(1-u(a_i)) F[i+1][k-1] \\} ,\quad 1\leq i \leq J-1,\ 1 \leq k \leq \min\\{m,J-i+1\\} .
> $$
> Here $F[i][k]:=\max_{(a'_1,\dots,a'_k) \subseteq (a_i,\dots,a_J)} f((a'_1,\dots,a'_k),q,w)$ denotes the optimal objective value that can be achieved by selecting $k$ items from items $a_i,\dots,a_J$ which are sorted in descending order of $w(a)$. Furthermore, we also provide rigorous analyses for the properties of $f(A,q,w)$ and the correctness of oracle BestPerm (Lemmas 1 and 2).
>
> To sum up, our generalization to cascading MDPs has the following novelties: (1) This generalization introduces a significant computational challenge on how to efficiently find the optimal permutation. (2) To resolve this challenge, we analyze the special properties of the objective function for cascading MDPs, and design an efficient oracle BestPerm based on a novel dynamic programming.
> Rigorous proofs are provided for the properties of the objective function and the correctness of oracle BestPerm.
> (3) Building upon oracle BestPerm and the carefully-designed exploration bonuses for estimates, we develop provably computationally-efficient and sample-efficient algorithms for cascading RL.

---

> ### Author Response · Authors · 2023-11-17
> **Response to Reviewer 6goX (Part 2/3)**
>
> **2. Experiment**
>
> Following the reviewer's suggestion, we conduct experiments on a larger-scale **real-world** dataset MovieLens [Harper \& Konstan 2015] (also used in prior works [Zong et al. 2016; Vial et al. 2022]) with more baselines. We present preliminary experimental results here (due to time limit), and will certainly include more results for larger numbers of states and items in our revision.
>
> The MovieLens dataset contains 25 million ratings on a 5-star scale for 62000 movies by 162000 users.
> We regard each user as a state, and each movie as an item. For each user-movie pair, we scale the rating to [0,1] and regard it as the attraction probability.
> The reward of each user-movie pair is set to 1. For each user-movie pair which has a rating no lower than 4.5 stars, we set the transition probability to this state (user) itself as 0.9, and that to other states (users) as $\frac{0.9}{S-1}$. For each user-movie pair which has a rating lower than 4.5 stars, we set the transition probability to all states (users) as $\frac{1}{S}$.
> We use a subset of data from MovieLens, and set $\delta=0.005$, $K=100000$, $H=3$, $m=3$, $S=20$, $N \in \\{10,15,20,25,30,40\\}$ (the number of items) and $|\mathcal{A}|=\sum_{\tilde{m}=1}^{m} \binom{N}{\tilde{m}} \tilde{m}! \in \\{820,2955,7240,14425,25260,60880\\}$ (the number of item lists).
>
> We compare our algorithm CascadingVI with **three baselines**, i.e., CascadingVI-Oracle, CascadingVI-Bonus and AdaptVI. Specifically, CascadingVI-Oracle is an ablated version of CascadingVI which replaces the efficient oracle BestPerm by a naive exhaustive search. CascadingVI-Bonus is an ablated variant of CascadingVI which replaces the variance-aware exploration bonus $b^{k,q}$ by a variance-unaware bonus.
> AdaptVI adapts the classic RL algorithm [Zanette
> \& Brunskill 2019] to the combinatorial action space, which maintains the estimates for all $(s,A)$ rather than $(s,a)$.
> The following table shows the cumulative regrets and running times of these algorithms. (CascadingVI-Oracle, which does not use our efficient computational oracle, is slow and still running in some instances with large $N$. We will update its results once it finishes.)
>
> | Regret; Running time  | CascadingVI (ours)  | CascadingVI-Oracle  | CascadingVI-Bonus  |  AdaptVI |
> | ------------ | ------------ | ------------ | ------------ | ------------ |
> | $N=10, \mid \mathcal{A} \mid =820$  | 2781.48; 1194.29s  | 4097.56; 21809.87s  | 8503.45; 1099.32s  | 29874.84; 28066.90s  |
> | $N=15, \mid \mathcal{A} \mid =2955$  | 4630.11; 2291.00s  | 6485.36; 71190.59s  | 13436.38; 2144.96s  | 30694.40; 53214.95s  |
> |  $N=20, \mid \mathcal{A} \mid =7240$ | 6446.51; 2770.94s  | 9950.22; 145794.64s  | 17472.21; 2731.21s  | 34761.91; 81135.42s |
> |  $N=25, \mid \mathcal{A} \mid =14425$ | 8283.60; 5216.00s   | 13170.39; 196125.72s  |  21868.54; 5066.27s | 42805.06; 95235.55s  |
> |  $N=30, \mid \mathcal{A} \mid =25260$ | 11412.85; 7334.13s  |  Still running | 27738.31; 7256.94s  | 52505.85; 188589.60s  |
> |  $N=40, \mid \mathcal{A} \mid =60880$ | 16825.65; 18628.83s  |  Still running | 37606.95; 17829.40s  |  56045.01; 319837.83s |
>
> As shown in the above table, our algorithm CascadingVI achieves the lowest regret and a fast running time. CascadingVI-Oracle has a comparative regret performance to CascadingVI, but suffers a much higher running time, which demonstrates the power of our oracle BestPerm in computation.
> CascadingVI-Bonus attains a similar running time as CascadingVI, but has a worse regret (CascadingVI-Bonus looks slightly faster because computing a variance-unaware exploration bonus is simpler). This corroborates the effectiveness of our variance-aware exploration bonus in enhancing sample efficiency.
> AdaptVI suffers a very high regret and running time, since it learns the information of all permutations independently and its statistical and computational complexities depend on $|\mathcal{A}|$.
>
> We have also presented the figures of these experimental results in our revised version.

---

> ### Author Response · Authors · 2023-11-17
> **Response to Reviewer 6goX (Part 3/3)**
>
> **3. Replies to Questions**
>
> **3.1 Computational Complexity of $\mathbb{E}_{s'\sim\tilde{p}^k}[\dots]$**
>
> The complexity of computing $\mathbb{E} (s'\sim\hat{p}^k) [( \bar{V}^k_{h+1}(s')-\underline{V}^k_{h+1}(s') )^2 ]$ in Algorithm 2 is $O(\mathcal{S})$. (In this response, we use $(s'\sim\hat{p}^k)$ to denote the subscript of $\mathbb{E}$ due to the formula display issue of OpenReview.)
>
> **3.2 Computation of the Value Function $w$**
>
> Algorithm CascadingVI performs optimistic value iteration by backward iteration for step $h=H,H-1,\dots,1$.
> At each step $h$, we already have the estimates $\hat{p}^k(s'|s,a)$ and $\bar{V}^k_{h+1}(s')$, and then we can compute the optimistic estimate for $w(s,a)$ as $\bar{w}^k(s,a) \leftarrow r(s,a)+\hat{p}^k(\cdot|s,a)^\top \bar{V}^k_{h+1}+ b^{k,pV}(s,a)$ (Line 10 of Algorithm 2). The complexity of computing $\bar{w}^k(s,a)$ is $O(\mathcal{S})$. After that, for each state $s \in \mathcal{S}$, we call oracle BestPerm with $\bar{w}^k(s,a)$ to efficiently calculate the optimal value and the optimal permutation, i.e., $\bar{V}^k_h(s),\ \pi^k_h(s) \leftarrow BestPerm(A^{ground}, \bar{q}^k(s,\cdot), \bar{w}^k(s,\cdot))$ (Line 11 of Algorithm 2).
>
> Actually, the biggest computational difficulty of BestPerm is how to efficiently solve the combinatorial optimization Eq. (ii) in our Reply 1. BestPerm takes advantage of the special properties of the objective function for cascading MDPs, and adopts a novel dynamic programming approach to solve it (see our Reply 1).
>
> **3.3 Intuition of the Exploration Bonus $b^{k,pV}(s,a)$**
>
> Our construction of $b^{k,pV}(s,a)$ follows that in prior RL work [Zanette \& Brunskill 2019], which guarantees the optimistic estimation for $p(\cdot|s,a)^{\top}  V^*_{h+1}$. Below we describe the intuition behind $b^{k,pV}(s,a)$ (Lemmas 7-8 in Appendix C.2).
>
> According to Bernstern's inequality, we have that with high probability,
> $$
> | (\hat{p}^k(\cdot|s,a) - p(\cdot|s,a))^{\top}  V^*_{h+1} | \leq 2 \sqrt{ \frac{Var_{s'\sim p}(V^*_{h+1}(s')) \log( \frac{KHSA}{\delta'} ) }{n^{k,p}(s,a)} } + \frac{H \log( \frac{KHSA}{\delta'} ) }{n^{k,p}(s,a)} . \quad Eq. (iii)
> $$
> Hence, the right-hand side of Eq. (iii) is the Bernstern-type exploration bonus of $p(\cdot|s,a)^{\top}  V^*_{h+1}$. However, the  algorithm does not know the exact $Var_{s'\sim p}(V^*_{h+1}(s'))$, and thus we want to use the estimate of $Var_{s'\sim p}(V^*_{h+1}(s'))$ instead of itself.
>
> Following the analysis in [Zanette \& Brunskill 2019], we have that if $\bar{V}^k_{h+1}(s') \geq V^*_{h+1}(s') \geq \underline{V}^k_{h+1}(s')$ for any $s' \in \mathcal{S}$, then with high probability,
> $$
> \sqrt{Var_{s'\sim p }(V^*_{h+1}(s'))}  \leq \sqrt{Var_{s'\sim \hat{p}^k }(\bar{V}^k_{h+1}(s'))} + \sqrt{\mathbb{E} (s'\sim\hat{p}^k) [( \bar{V}^k_{h+1}(s')-\underline{V}^k_{h+1}(s') )^2 ]} + 2H \sqrt{\frac{ \log( \frac{KHSA}{\delta'}) }{n^{k,p}(s,a)}} .
> $$
>
> Plugging the above inequality into Eq. (iii), we have that with high probability,
>
> $$
> | (\hat{p}^k(\cdot|s,a) - p(\cdot|s,a))^{\top}  V^*_{h+1} | \leq 2 \sqrt{ \frac{Var_{s'\sim \hat{p}^k} (\bar{V}^k_{h+1}(s')) \log( \frac{KHSA}{\delta'} ) }{n^{k,p}(s,a)} } + 2 \sqrt{ \frac{\mathbb{E} (s'\sim\hat{p}^k) [( \bar{V}^k_{h+1}(s')-\underline{V}^k_{h+1}(s') )^2 ] \log( \frac{KHSA}{\delta'} ) }{n^{k,p}(s,a)} } + \frac{5H \log( \frac{KHSA}{\delta'} ) }{n^{k,p}(s,a)} := b^{k,pV}(s,a) .
> $$
>
> Thus, if $\bar{V}^k_{h+1}(s') \geq V^*_{h+1}(s') \geq \underline{V}^k_{h+1}(s')$ for any $s' \in \mathcal{S}$, then $\hat{p}^k(\cdot|s,a)^\top \bar{V}^k_{h+1}+ b^{k,pV}(s,a)$ is an optimistic estimate of $p(\cdot|s,a)^{\top}  V^*_{h+1}$.
>
> In the following, we discuss how to guarantee the optimism of $\bar{V}^k_{h+1}$, and the proof for pessimism is similar (Lemmas 14-15 in Appendix C.2).
>
> First, we prove that $f(A,q,w)$ satisfies the monotonicity property, i.e.,  if $\bar{q} \geq q$ element-wise and the items in $A$ are ranked in descending order of $w$, then $f(A,\bar{q},w) \geq f(A,q,w)$.
>
> Then, we prove the optimism of $\bar{V}^k_{h+1}$ by induction.
> For $h=H$, it trivially holds that $\bar{V}^k_{H+1} \geq V^*_{H+1}$ element-wise, and thus $\hat{p}^k(\cdot|s,a)^\top \bar{V}^k_{H+1}+ b^{k,pV}(s,a)$ is an optimistic estimate of $p(\cdot|s,a)^{\top}  V^*_{H+1}$.
> Using the monotonicity property of $f(A,q,w)$, the fact that the items in the optimal permutation are ranked in descending order of weights, and the optimism of $\bar{w}^k(s,a) = r(s,a)+\hat{p}^k(\cdot|s,a)^\top \bar{V}^k_{H+1}+ b^{k,pV}(s,a)$, we can prove that the output value $\bar{V}^k_H$ of oracle BestPerm is an optimistic estimate of $V^*_{H}$.
> For $h=H-1,H-2,\dots,1$, by induction and similarly using the optimism of $\bar{V}^k_{h+1}$ and $\hat{p}^k(\cdot|s,a)^\top \bar{V}^k_{h+1}+ b^{k,pV}(s,a)$, we can prove that $\bar{V}^k_h$ is an optimistic estimate of $V^*_{h}$ for all $h \in [H]$.
>
> We have added more explanation in Section 5.1 in our revision.

---

> > ### Comment · Reviewer_6goX · 2023-11-20
> > **Other clarifications**
> >
> > Thank you for the clarifications. The intuition behind the exploration bonus is more clear to me now.
> >
> > Thank you for introducing the real world experiments. These baselines are more meaningful in my opinion.

---

> ### Comment · Reviewer_6goX · 2023-11-20
> **Still not convinced on significance / novelty**
>
> 1. Contribution of our Generalization to Cascading MDPs
>
> Thank you for the detailed response. Please allow to further articulate my point of view on why I believe this extension's significance is limited relative to what I believe is already known from the vanilla cascading bandits (and please correct me where I am missing something).
>
> Dynamic programming is already an established method for computing optimal policies in an MDP. The proposed novelty here is the reduction of computation time for this algorithm by exploiting the cascading structure, i.e. the fact that "basic" actions are rankings (and hence the number of actual items in our ranking pool is much much smaller than that of actions). But in my opinion, it is already known that we don't need to evaluate every individual ranking, since it is sufficient to know the expected value of each individual item in this state (probability of click and expected future value) to construct the optimal ranking in a given state and avoid complexities in $O(|\mathcal{A}|)$. Therefore I am not convinced this result (expanding the setting to the RL setting and providing an oracle algorithm avoiding $O(|\mathcal{A}|)$ computational complexity) represents a sufficiently significant addition to the literature.
>
> I have one other standing curiosity for which I wonder if you can provide any insights: Would a naive algorithm that simply takes the top $m$ items as sorted by $q(s, i) \times w(s, i)$ (the expected return of item $i$ weighted by its chance to be clicked when inspected) and further sorts these by $w(s, i)$ (items with the highest expected return at the top, regardless of their attractiveness) produces optimal rankings?
>
> Any insights you can offer are appreciated (I am not expecting a proof, just wondering if you have any counter-example at hand or whether you've seen this fail in your experiments).

---

> ### Author Response · Authors · 2023-11-21
> **Additional Response to Reviewer 6goX**
>
> Thank you for giving us an opportunity to interact with you!
>
> **1. Significance of Our Contribution**
>
> Indeed it is already known that one only needs to evaluate each item, instead of each ranking, in classic cascading (combinatorial) bandits [Kveton et al. 2015]. However, the objective function in cascading bandits
> $$
> \max_{A \in \mathcal{A}} \quad f(A,q) = \sum_{i=1}^{|A|} \prod_{j=1}^{i-1} \Big( 1-q(A(j)) \Big) q(A(i)) = 1- \prod_{i=1}^{|A|} \Big( 1-q(A(i)) \Big) . \quad Eq. (i)
> $$
> only involves **one attribute** of items, i.e., the attraction probability $q(a)$. This objective function completely depends on $q(a)$, which is obvious to compute. They only need to select $m$ items with the maximum attraction probabilities $q(a)$.
>
> However, when generalizing to RL, the objective function at each step $h$ is
> $$
> \max_{A \in \mathcal{A}} f(A,q,w) :=  \sum_{i=1}^{|A|} \prod_{j=1}^{i-1} \Big( 1-q(A(j)) \Big) q(A(i)) w(A(i)) , \quad Eq. (ii)
> $$
> which involves **two attributes** of items, i.e., the attraction probability $q(a)$ and the expected current and future reward $w(a)$ (here $w(a)=r(s,a)+p(\cdot|s,a)^\top V^*_{h+1}$ given the state $s$ and step $h$).
> **Eq. (ii) itself is a complex combinatorial optimization problem** even when one only evaluates the attributes of individual items and ignores the recursion over step $h$, and there is **no obvious monotone rule** with respect to $q(a)$, $w(a)$ or $q(a)*w(a)$.
>
> To solve Eq. (ii), we design an efficient computational oracle BestPerm, using a novel dynamic programming rule based on the cascading feature of $q(a)$ and $w(a)$:
> $$
> F[i][k] = \max \\{ F[i+1][k],\  q(a_i)  w(a_i) +(1-q(a_i)) F[i+1][k-1] \\} ,\quad 1\leq i \leq J-1,\ 1 \leq k \leq \min\\{m,J-i+1\\} , \quad Eq. (iv)
> $$
> where
>
> $
> F[i][k]:=\max_{(a'_1,\dots,a'_k) \subseteq (a_i,\dots,a_J)} f((a'_1,\dots,a'_k),q,w)
> $
>
> denotes the optimal objective value achieved by selecting $k$ items from items $a_i,\dots,a_J$ which are sorted in descending order of $w(a)$.
> The intuition behind our dynamic programming rule is that: for a sorted item sequence $a_i,a_{i+1},\dots,a_J$, if we do not select $a_{i}$, then the optimal objective value is equal to that achieved by selecting $k$ items from $a_{i+1},\dots,a_J$, i.e., $F[i+1][k]$. Otherwise, if we select $a_{i}$, then the optimal objective value is equal to $(1-q(a_i))$ multiplying the optimal value achieved by selecting $k-1$ items from $a_{i+1},\dots,a_J$, i.e., $(1-q(a_i)) F[i+1][k-1]$, due to the cascading feature of the objective function.
>
> We note that the "dynamic programming" we mentioned in our contribution statements refers to the dynamic programming Eq. (iv) that we design to solve the combinatorial optimization Eq. (ii), which iterates over the indices of items $1\leq i \leq J-1$ and the number of selected items $1 \leq k \leq \min\\{m,J-i+1\\}$. **We were not referring to the well-known MDP dynamic programming that iterates over step $h=H,H-1,\dots,1$.**
>
> In other words, our dynamic programming rule (Eq. (iv)) and oracle BestPerm provide an efficient computational solution to the combinatorial optimization Eq. (ii), which one needs to solve at each step $h$ in cascading RL even when only using the attributes of individual items. This is a significant contribution and novel to the combinatorial bandit/RL literature.
>
> **2. Counter-example for the Naive Algorithm**
>
>  A naive algorithm that first sorts items by $q(a)*w(a)$ and further sorts them by $w(a)$ cannot find the optimal ranking.
>
>  Consider a counter-example as follows:
>  There are 10 items $a_1,a_2,\dots,a_{10}$, and we want to find the optimal ranking with the maximum cardinality $m=3$. The attraction probabilities $q(a)$, weights $w(a)$ and their products $q(a)*w(a)$ of items are shown in the following table.
>
> |   |  $a_1$ |  $a_2$ | $a_3$  | $a_4$  | $a_5$  | $a_6$  | $a_7$  |  $a_8$ | $a_9$  | $a_{10}$  |
> | ------------ | ------------ | ------------ | ------------ | ------------ | ------------ | ------------ | ------------ | ------------ | ------------ | ------------ |
> |  $q(a)$ | 0.1  |  0.2 |  0.3 |  0.4 | 0.5  | 0.6  | 0.7  | 0.8  | 0.9  | 1  |
> |  $w(a)$ |  1 |  0.9 |  0.8 | 0.7  | 0.6  |  0.5 |  0.4 | 0.3  |  0.2 | 0.1  |
> | $q(a)*w(a)$  |  0.1 | 0.18| 0.24| 0.28| 0.3 | 0.3 | 0.28| 0.24| 0.18| 0.1 |
>
> The naive algorithm the reviewer mentioned will output $(a_4,a_5,a_6)$, while our computational oracle BestPerm will output $(a_3,a_4,a_5)$ according to our dynamic programming rule Eq. (iv).
>
> The objective values $f(A,q,w) :=  \sum_{i=1}^{|A|} \prod_{j=1}^{i-1} ( 1-q(A(j)) ) q(A(i)) w(A(i))$ of $(a_4,a_5,a_6)$ and $(a_3,a_4,a_5)$ are
>
> $$
> f((a_4,a_5,a_6),q,w)=0.28+(1-0.4)*0.3
> $$
>
> $$
> \qquad\qquad\qquad\qquad\qquad +(1-0.4)*(1-0.5)*0.3=0.55 ,
> $$
>
> $$
> f((a_3,a_4,a_5),q,w)=0.24+(1-0.3)*0.28
> $$
>
> $$
> \qquad\qquad\qquad\qquad\qquad +(1-0.3)*(1-0.4)*0.3=0.562 > f((a_4,a_5,a_6),q,w) .
> $$
>
> This demonstrates that the output of the naive algorithm $(a_4,a_5,a_6)$ is not the optimal solution.

---

> > ### Comment · Reviewer_6goX · 2023-11-21
> > **Thank you for the clarifications**
> >
> > Thank you for the clarifications and especially for the insight on the difficulty of computing the best permutation. My concerns about the novelty are now somewhat eased. I recommend that in future revisions of the paper you put more emphasis on the difficulties solved by BestPerm and presenting this contrast between such a naive policy (picking the top $m$ items by expected value $q \times w$) and the actual optimal ranking. The first few reads, I did not pick up on the nuances of this difficulty, my intuition being that it is fairly low hanging fruit.

---

> > > ### Author Response · Authors · 2023-11-21
> > > **Thank you very much for your time in interacting with us!**
> > >
> > > Thank you very much for your time in interacting with us and insightful suggestions! We will certainly include more discussion on the difficulties solved by BestPerm and the differences between a naive policy and the actual optimal ranking in our future revision.

---

### Official Review · Reviewer_Rma4 · 2023-10-29

**Soundness:** 3 good
**Presentation:** 3 good
**Contribution:** 3 good
**Rating:** 8
**Confidence:** 3

**Summary:**

This paper proposes an extension of the cascading bandits framework to a more general reinforcement learning framework. The authors thereby attempt to model user sessions or historical user behavior and their impact on the click behavior and payoff.  While the action space (i.e., combination of items) is combinatorial, the feedback is item-wise so that attraction and transition probabilities can be estimated efficiently. This allows the authors to design algorithms with non-trivial regret and sample complexity guarantees. In particular, the authors rely on monotonicity properties to design an efficient oracle. Finally, the authors support their theoretical findings through experiments which support the efficiency of the proposed algorithms.

**Strengths:**

- Firstly, I think that the studied problem is interesting and the presentation of the paper clear.
- The extension of the cascading bandit model to an RL formulation is highly non-trivial and the paper contains novel contributions including the RL formulation itself and an interesting algorithm design which relies on properties of the value function.
- The authors do a good job explaining their algorithms and provide intuition for their results.
- The necessity of adopting standard RL algorithms to the proposed cascading model is highlighted in the paper several times and the authors thereby provide proper justification for the proposed algorithms.

**Weaknesses:**

- I am not fully convinced of the practicality of this model. Firstly, historical user behavior can be modeled as part of the context in contextual bandit frameworks. Moreover, "artificially" creating states appears fairly cumbersome compared to the contextual bandit structure and it is also not entirely clear how such states would be defined in practice. However, I could be convinced otherwise.

**Questions:**

- Could you further explain why contextual approaches are insufficient and what the merit of the RL formulation is in contrast to contextual bandits or recommendation approaches based on context trees? In the RL framework it is important what states the user transitions to, as some states may have the potential to yield higher rewards than others. Do you think that this is realistic in practice?

- It would be great if you could go into a bit more detail in your related work section and more clearly highlight the differences to prior models. For example, in contrast to the classical cascading bandits, the order of items also matters in your case as you usually would like the item with largest reward in state s to be clicked.

Minor things:
- In the 4th contribution, I found the statement about "$\varepsilon$ sufficiently large" slightly confusing. It only becomes clearer later in Theorem 2 when the full bound is stated. Maybe there is a way to state this less conusingly in the introduction.
- Typo in the 3rd paragraph of Section 7: "Regarding the best policy identification objective" instead of "Regarding the regret minimization objective".

---

> ### Author Response · Authors · 2023-11-17
> **Response to Reviewer Rma4**
>
> Thank you very much for your time in reviewing our paper and your positive comments! We have revised our paper according to your suggestions. Our revision is highlighted in *orange* font.
>
> **1. Merit of the RL Formulation Compared to Contextual Bandits**
>
> We agree with the reviewer that contextual bandits can also be used to formulate historical user behaviors as part of contexts.
> We think that RL is more suitable for long-term reward maximization, since it considers state transition and potential future rewards in decision making.
>
> For example, consider a video recommendation scenario. The recommendation system encounters a user who is interested in funny videos and TV series.
> Given this context, contextual bandits mainly recommend videos which can maximize the reward in the current context. Hence, contextual bandits may recommend funny videos which usually give higher instantaneous utilities.
> In contrast, RL recommends videos which maximize the expected cumulative (long-term) reward, which also considers the potential rewards generated by future states. Thus, in this case, RL may recommend TV series videos. Although TV series videos may have lower instantaneous utilities, once the user is attracted to some of them, the user can enter a high-reward successor state, i.e., he/she is obsessed with it and keeps watching subsequent videos of this TV series. Therefore, using the RL formulation can obtain higher rewards in the long term.
>
> **2. Related Works**
>
> We detail the differences of our cascading RL model from prior works below.
>
> [Kveton et al. 2015; Cheung et al. 2019; Zhong et al. 2021; Vial et al. 2022] study the cascading bandit model. In their model, there is no state (context), and the reward generated by each item (if clicked) is one. Thus, the order of selected items does not matter, and they only need to select $m$ items with the maximum attraction probabilities.
> [Zong et al. 2016; Li \& Zhang 2018] consider the  contextual cascading bandit problem. In their problem, the agent first observes the context (i.e., the feature vector of each item) at each timestep, and the attraction probability of each item is the inner-product of its feature vector and a global parameter. The order of selected items still does not matter, and they need to select $m$ items with the maximum attraction probabilities in the current context.
> [Li et al. 2016] investigate contextual cascading bandits with position discount factors and a general reward function. In their model, the order of selected items matters, and they assume to have access to an oracle that can output the optimal ordered subset of items for the general reward function.
>
> Different from the above works, our cascading RL formulation further considers state transition, and the attraction probabilities and rewards of items depend on states. Thus, we need to put the items with higher rewards in the current state in the front. In addition, we require to both maximize the attraction probabilities of selected items, and optimize the potential rewards from future states. Furthermore, instead of assuming access to an oracle, we design an efficient oracle to find the optimal ordered subset of items.
>
> **3. Replies to Minor Things**
>
> Thank you for pointing out the unclear sentence and typo!
>
> We have revised the statement in the fourth contribution to "CascadingBPI is
> optimal up to a factor of $\tilde{O}(H)$ when $\varepsilon < \frac{H}{S^2}$" to avoid confusion, and fixed the typo.
>
>
> ---
>
> References:
> [1] Branislav Kveton, Csaba Szepesvari, Zheng Wen, and Azin Ashkan. Cascading bandits: Learning
> to rank in the cascade model. ICML 2015.
> [2] Wang Chi Cheung, Vincent Tan, and Zixin Zhong. A thompson sampling algorithm for cascading
> bandits. AISTATS
> 2019.
> [3] Zixin Zhong, Wang Chi Chueng, and Vincent YF Tan. Thompson sampling algorithms for cascading
> bandits. JMLR 2021.
> [4] Daniel Vial, Sujay Sanghavi, Sanjay Shakkottai, and R Srikant. Minimax regret for cascading bandits.
> NeurIPS 2022.
> [5] Shi Zong, Hao Ni, Kenny Sung, Nan Rosemary Ke, Zheng Wen, and Branislav Kveton. Cascading
> bandits for large-scale recommendation problems. UAI 2016.
> [6] Shuai Li and Shengyu Zhang. Online clustering of contextual cascading bandits. AAAI 2018.
> [7] Shuai Li, Baoxiang Wang, Shengyu Zhang, and Wei Chen. Contextual combinatorial cascading
> bandits. ICML 2016.

---

> > ### Comment · Reviewer_Rma4 · 2023-11-19
> >
> > Thank you for responding to my comments.
> >
> > Regarding your first answer. While it does make sense to me that you can model a "customer journey" like this and that this can be useful, defining sensible states seems quite hard and could simply boil down to something like a context tree (which would be simpler than the MDP framework). However, I still appreciate the advantages of "forward" planning and trying to transition to rewarding states, despite the challenges in designing such states in practice. I do not wish to change my original assessment and I am still in favour of accepting the paper.

---

> > > ### Author Response · Authors · 2023-11-19
> > > **Thank you very much for your appreciation!**
> > >
> > > Thank you very much for your appreciation for our work and raising an insightful future direction! We will further explore how to design sensible states for better practicability in our future work.

---

### Official Review · Reviewer_MkFu · 2023-11-03

**Soundness:** 3 good
**Presentation:** 3 good
**Contribution:** 2 fair
**Rating:** 6
**Confidence:** 4

**Summary:**

The paper presents a new framework for reinforcement learning (RL) called cascading RL, which builds upon the concept of cascading bandits but incorporates state information into the decision-making process. This framework is aimed at applications such as personalized recommendation systems and online advertising, where cascading items are displayed to users one by one.

Key challenges addressed include computational difficulty due to the combinatorial nature of actions and ensuring sample efficiency without relying on the exponential number of potential actions. To overcome these, the authors developed an efficient oracle called BestPerm, which uses dynamic programming to optimize item selection. They also introduced two RL algorithms: CascadingVI for regret minimization and CascadingBPI for sample complexity, both of which utilize BestPerm to achieve polynomial regret and sample complexity.

CascadingVI achieves near-optimal regret matching a known lower bound. CascadingBPI offers efficient computation and sample complexity, reaching near-optimal performance.

In summary, the paper contributes a new cascading RL framework that efficiently handles the computational and sample complexity challenges of stateful item selection, supported by theoretical guarantees and demonstrated effectiveness through experiments.

**Strengths:**

- As far as I know, this would be the first theory RL work with cascading actions. The formulation is sound.
- The paper proposes an efficient optimization oracle for cascading RL, providing its correctness (it would be better to show it in the main text rather than the appendix).
- Although it is not groundbreaking, the paper provides both regret analysis and sample complexity analysis.

**Weaknesses:**

- It seems that the techniques used for regret analysis and sample complexity are largely borrowed from the previous literature. I wonder if there are sufficient technical challenges once you know how to solve Problem (2).
- The readability of the algorithms can be improved. e.g., Update rule of $\bar{q}^k$ and $\underbar{q}^k$ in Line 6 of Algorithm 2, Line break in Line 8 of Algorithm 2, etc.

**Questions:**

Where does the gap $\sqrt{H}$ appear? Can you elaborate on this? Any possible direction to carve off this given that the tabular RL methods with a single action can achieve minimax optimality?

---

> ### Author Response · Authors · 2023-11-17
> **Response to Reviewer MkFu (Part 1/2)**
>
> Thank you very much for your time and effort in reviewing our paper! We have revised our paper according to your comments. Our revision is highlighted in *orange* font.
>
> **1. Presentation on the Correctness of Oracle BestPerm**
>
> Thank you for your helpful suggestion! We have moved the correctness guarantee of oracle BestPerm to the main text (Lemma 2) in our revision.
>
> **2. Technical Challenges**
>
> Below we elaborate the technical challenges and novelties in addition to the oracle for solving Problem Eq. (2).
>
>
> - Even if we know how to solve Problem Eq. (2), in the online RL process, we only have the estimates of attraction probabilities, transition distributions and value functions. Then, how to **guarantee the optimism of the output value by oracle BestPerm** with optimistic estimated inputs is a crucial challenge. To handle this challenge, we prove the *monotonicity* property of the objective function for cascading RL by induction, leveraging the fact that *the items in the optimal permutation are ranked in descending order of $w$*  (Lemma 14). This monotonicity property guarantees the optimism of our estimation under the use of oracle BestPerm with optimistic estimates (Lemma 15).
>
> - Since the objective function for cascading RL involves two attributes of items, i.e., attraction probabilities $q(s,a)$ and the expected future rewards $p(\cdot|s,a)^\top V^*_{h+1}$, how to construct exploration bonuses for them is an important step in algorithm design.
> We build the exploration bonuses for $q(s,a)$ and $p(\cdot|s,a)^\top V^*_{h+1}$ separately for  $(s,a)$, rather than adding an overall exploration bonus for each $(s,A)$. This exploration bonus design enables us to achieve regret and sample complexity bounds that depend only on the number of items $N$, instead of the number of item lists $|\mathcal{A}|$.
>
> - Another critical challenge is how to **achieve the optimal regret bound when our problem degenerates to cascading bandits** [Vial et al. 2022] given that cascading RL has a more complicated state-transition structure. To tackle this challenge, we use a variance-aware exploration bonus $b^{k,q}(s,a)=\sqrt{\frac{\hat{q}^k(s,a) (1-\hat{q}^k(s,a)) L}{n^{k,q}(s,a)}} + \frac{5 L}{n^{k,q}(s,a)}$, which can adaptively shrink when the attraction probability $q(s,a)$ is small.
> In regret analysis, when performing summation over $b^{k,q}(s,a)$, this variance-awareness saves a factor of $\sqrt{m}$, and enables our regret bound to match the optimal result in cascading bandits [Vial et al. 2022].
>
>
> **3. Readability**
>
> Thank you for your valuable suggestions! We have revised Algorithm 2 according to your comments to improve readability in our revision.

---

> ### Author Response · Authors · 2023-11-17
> **Response to Reviewer MkFu (Part 2/2)**
>
> **4. The Gap $\sqrt{H}$**
>
> The gap $\sqrt{H}$ appears because we construct the exploration bonuses for $q(s,a)$ and $p(\cdot|s,a)^\top V^*_{h+1}$ separately, which leads to an additional $\sqrt{H}$ factor when summing up $b^{k,q}(s,a)$ and $b^{k,pV}(s,a)$ in regret analysis.
>
> Below we elaborate this in detail.  In regret analysis, using value difference lemma, the regret decomposition contains the following term
> $$
> \sum_{k=1}^{K} \sum_{h=1}^{H} \mathbb{E} \Bigg[
> \sum_{i=1}^{|A_h|} \prod_{j=1}^{i-1} (1-q(s_h,A_h(j))) \bigg( \bar{q}^k(s_h,A_h(i)) \left( \hat{p}^k(\cdot|s_h,A_h(i))^\top \bar{V}^k_{h+1} + b^{k,pV}(s_h,A_h(i)) \right)- q(s_h,A_h(i)) p(\cdot|s_h,A_h(i))^\top \bar{V}^k_{h+1} \bigg) \Bigg]
> $$
> $$
> = \sum_{k=1}^{K} \sum_{h=1}^{H} \mathbb{E} \Bigg[ \sum_{i=1}^{|A_h|} \prod_{j=1}^{i-1} (1-q(s_h,A_h(j))) \bigg( \Big( q(s_h,A_h(i))+2b^{k,q}(s_h,A_h(i)) \Big)  \Big( \hat{p}^k(\cdot|s_h,A_h(i))^\top \bar{V}^k_{h+1} + b^{k,pV}(s_h,A_h(i)) \Big) - q(s_h,A_h(i)) p(\cdot|s_h,A_h(i))^\top \bar{V}^k_{h+1} \bigg) \Bigg] \quad Eq. (i)
> $$
> $$
> \leq \sum_{k=1}^{K} \sum_{h=1}^{H} \mathbb{E} \Bigg[ \sum_{i=1}^{|A_h|} \prod_{j=1}^{i-1} (1-q(s_h,A_h(j))) \bigg( q(s_h,A_h(i)) \Big( \hat{p}^k(\cdot|s_h,A_h(i))^\top \bar{V}^k_{h+1} + b^{k,pV}(s_h,A_h(i)) - p(\cdot|s_h,A_h(i))^\top \bar{V}^k_{h+1} \Big) + 4Hb^{k,q}(s_h,A_h(i)) \bigg) \quad Eq. (ii) \Bigg]
> $$
> Since we construct exploration bonuses for $q(s,a)$ and $p(\cdot|s,a)^\top V^*_{h+1}$ separately, in Eq. (i) above, the regret contains a product $(q+b^{k,q})(\hat{p}^\top \bar{V}+b^{k,pV}-p^\top V)$, while the regret for classic RL only contains the term $\hat{p}^\top \bar{V}+b^{k,pV}-p^\top V$. To handle this product, we bound it by $q(\hat{p}^\top \bar{V}+b^{k,pV}-p^\top V)+b^{k,q}(\hat{p}^\top \bar{V}+b^{k,pV}-p^\top V)=q(\hat{p}^\top \bar{V}+b^{k,pV}-p^\top V)+O(H b^{k,q})$. Here the $O(H b^{k,q})$ term directly leads to a $\tilde{O}(H\sqrt{HK})$ regret.
>
> A straightforward idea to improve the $\sqrt{H}$ gap is to combine the attraction probability $q$ and transition distribution $p$ as an integrated transition distribution $\tilde{p}(s'|s,A)$ for each item list $A$, and construct an overall exploration bonus for $\tilde{p}(s'|s,A)$. However, this strategy forces us to maintain the exploration bonus for each $A \in \mathcal{A}$, which will incur an additional dependency on $\mathcal{A}$ in the regret bound and is computationally inefficient.
>
> We think that a more fine-grained analysis for bounding $b^{k,q}(\hat{p}^\top \bar{V}+b^{k,pV}-p^\top V)$ or a more advancing bonus construction approach may be helpful to close the $\sqrt{H}$ gap. We plan to further investigate it in our future work.
>
> ---
>
> Reference:
> Daniel Vial, Sujay Sanghavi, Sanjay Shakkottai, and R Srikant. Minimax regret for cascading bandits.
> NeurIPS 2022.

---

> > ### Comment · Reviewer_MkFu · 2023-11-22
> > **Thanks**
> >
> > Thanks for the responses. I will keep my positive score.

---

> > > ### Author Response · Authors · 2023-11-22
> > > **Thank you very much again for your time in reviewing our paper!**
> > >
> > > Thank you very much again for your positive comments and time in reviewing our paper!

---

### Official Review · Reviewer_z4b8 · 2023-11-06

**Soundness:** 4 excellent
**Presentation:** 3 good
**Contribution:** 3 good
**Rating:** 8
**Confidence:** 4

**Summary:**

This paper studies cascading reinforcement learning, which is a natural extension of cascading bandits to the episodic reinforcement learning setting. In particular, this paper has

- proposed the cascading reinforcement learning framework;

- developed a computationally efficient algorithm to solve the "cascading MDPs" (i.e. the cascading reinforcement learning problems with known models). The key ideas are summarized in the BestPerm algorithm (Algorithm 1).

- developed a learning algorithm for the cumulative regret minimization setting, which is referred to as CascadingVI (Algorithm 2). A regret bound is also established (Theorem 1). This paper has also discussed the tightness of this regret bound.

- also developed a learning algorithm for the best policy identification setting, which is referred to as CascadingBPI. A sample complexity bound is established (Theorem 2). This paper has also discussed the tightness of this regret bound.

- demonstrated preliminary experiment results (Section 7).

**Strengths:**

In general, I think this paper is a strong theoretical paper, for the following reasons:

- This paper studies a natural extension of a well-studied problem. Moreover, as summarized above, the contributions of this paper are clear.

- The main results of this paper, summarized in Section 4 (efficient oracle), Section 5 (regret minimization), and Section 6 (best policy identification), are interesting and non-trivial. In particular, the regret bound in Theorem 1 and the sample complexity bound in Theorem 2 are non-trivial. Moreover, this paper has also discussed the tightness of these two bounds by comparing with existing lower bounds. Both bounds are near-optimal.

- The paper is well-written in general, and is easy to read.

**Weaknesses:**

- I am wondering if the developed algorithms are useful for practical recommendation problems. The reason is that, this paper considers a tabular setting, thus the developed regret bound and the sample complexity bound depend on the number of states $S$. However, my understanding is that for most practical recommendation problems, $S$ will be exponentially large. Might the authors identify a setting where $S$ is not too large and hence the proposed algorithms are practical?

- Currently the experiment results are very limited. In particular, this paper has only demonstrated experiment results in small-scale synthetic problems. I think experiment results on large-scale practical problems will further strengthen this paper.

**Questions:**

Please address the weaknesses above.

---

> ### Author Response · Authors · 2023-11-17
> **Response to Reviewer z4b8 (Part 1/2)**
>
> Thank you very much for reviewing our paper and your positive comments! We have incorporated your suggestions in the revised version, and highlighted our revision in *teal* font.
>
> **1. The Scenario Where $S$ Is Not Too Large**
>
> Consider a video recommendation scenario where the videos are categorized into multiple types, e.g., news, education, entertainment and movies.
> Here each item is a video, and each action (item list) is a list of videos. We can define each state as the types of the last one or two videos that the user just watched, which represents the recent preference and focus of the user. Then, the recommendation system suggests a list of videos according to the recent viewing record of the user. After the user chooses a new video to watch, the environment transitions to a next state which represents the user's latest interest.
> In such scenarios, the number of states $S$ is polynomial in the number of types of videos, which is not too large. Therefore, our cascading RL formulation and algorithms are useful and practical for these applications.

---

> > ### Author Response · Authors · 2023-11-19
> > **Response to Reviewer z4b8 (Part 2/2)**
> >
> > **2. Experiment**
> >
> > Following the reviewer's suggestion, we conduct experiments on a larger-scale **real-world** dataset MovieLens [Harper \& Konstan 2015] (also used in prior works [Zong et al. 2016; Vial et al. 2022]) with more baselines. We present preliminary experimental results here (due to time limit), and will certainly include more results for larger numbers of states and items in our revision.
> >
> > The MovieLens dataset contains 25 million ratings on a 5-star scale for 62000 movies by 162000 users.
> > We regard each user as a state, and each movie as an item. For each user-movie pair, we scale the rating to [0,1] and regard it as the attraction probability.
> > The reward of each user-movie pair is set to 1. For each user-movie pair which has a rating no lower than 4.5 stars, we set the transition probability to this state (user) itself as 0.9, and that to other states (users) as $\frac{0.9}{S-1}$. For each user-movie pair which has a rating lower than 4.5 stars, we set the transition probability to all states (users) as $\frac{1}{S}$.
> > We use a subset of data from MovieLens, and set $\delta=0.005$, $K=100000$, $H=3$, $m=3$, $S=20$, $N \in \\{10,15,20,25,30,40\\}$ (the number of items) and $|\mathcal{A}|=\sum_{\tilde{m}=1}^{m} \binom{N}{\tilde{m}} \tilde{m}! \in \\{820,2955,7240,14425,25260,60880\\}$ (the number of item lists).
> >
> > We compare our algorithm CascadingVI with three baselines, i.e., CascadingVI-Oracle, CascadingVI-Bonus and AdaptVI. Specifically, CascadingVI-Oracle is an ablated version of CascadingVI which replaces the efficient oracle BestPerm by a naive exhaustive search. CascadingVI-Bonus is an ablated variant of CascadingVI which replaces the variance-aware exploration bonus $b^{k,q}$ by a variance-unaware bonus.
> > AdaptVI adapts the classic RL algorithm [Zanette
> > \& Brunskill 2019] to the combinatorial action space, which maintains the estimates for all $(s,A)$ rather than $(s,a)$.
> > The following table shows the cumulative regrets and running times of these algorithms. (CascadingVI-Oracle, which does not use our efficient computational oracle, is slow and still running in some instances with large $N$. We will update its results once it finishes.)
> >
> > | Regret; Running time  | CascadingVI (ours)  | CascadingVI-Oracle  | CascadingVI-Bonus  |  AdaptVI |
> > | ------------ | ------------ | ------------ | ------------ | ------------ |
> > | $N=10, \mid \mathcal{A} \mid =820$  | 2781.48; 1194.29s  | 4097.56; 21809.87s  | 8503.45; 1099.32s  | 29874.84; 28066.90s  |
> > | $N=15, \mid \mathcal{A} \mid =2955$  | 4630.11; 2291.00s  | 6485.36; 71190.59s  | 13436.38; 2144.96s  | 30694.40; 53214.95s  |
> > |  $N=20, \mid \mathcal{A} \mid =7240$ | 6446.51; 2770.94s  | 9950.22; 145794.64s  | 17472.21; 2731.21s  | 34761.91; 81135.42s |
> > |  $N=25, \mid \mathcal{A} \mid =14425$ | 8283.60; 5216.00s   | 13170.39; 196125.72s  |  21868.54; 5066.27s | 42805.06; 95235.55s  |
> > |  $N=30, \mid \mathcal{A} \mid =25260$ | 11412.85; 7334.13s  |  Still running | 27738.31; 7256.94s  | 52505.85; 188589.60s  |
> > |  $N=40, \mid \mathcal{A} \mid =60880$ | 16825.65; 18628.83s  |  Still running | 37606.95; 17829.40s  |  56045.01; 319837.83s |
> >
> > As shown in the above table, our algorithm CascadingVI achieves the lowest regret and a fast running time. CascadingVI-Oracle has a comparative regret performance to CascadingVI, but suffers a much higher running time, which demonstrates the power of our oracle BestPerm in computation.
> > CascadingVI-Bonus attains a similar running time as CascadingVI, but has a worse regret (CascadingVI-Bonus looks slightly faster because computing a variance-unaware exploration bonus is simpler). This corroborates the effectiveness of our variance-aware exploration bonus in enhancing sample efficiency.
> > AdaptVI suffers a very high regret and running time, since it learns the information of all permutations independently and its statistical and computational complexities depend on $|\mathcal{A}|$.
> >
> > We have also presented the figures of these experimental results in our revised version.

---

> > > ### Comment · Reviewer_z4b8 · 2023-11-22
> > > **Thanks for the experiment results!**
> > >
> > > Thanks for the experiment results! They will definitely further strengthen the paper!

---

> > > > ### Author Response · Authors · 2023-11-22
> > > > **Thank you very much again for your time in reviewing our paper!**
> > > >
> > > > Thank you very much again for your time in reviewing our paper and reading our rebuttal!

---

> > ### Comment · Reviewer_z4b8 · 2023-11-22
> > **Thanks!**
> >
> > Thanks for the response! It has addressed one of my concerns.

---

### Meta-Review · Area_Chair_HZqs · 2023-12-05

**Metareview:**

This paper generalizes online learning to rank in the cascade model to reinforcement learning (RL). The main difference from prior works in the bandit setting is that the agent also optimizes for future states. The paper was a clear accept after the author-reviewer discussion. Some of the initial concerns, that the model does not apply to more realistic problems and that the technical contribution is limited, were addressed in the rebuttal.

**Justification For Why Not Higher Score:**

Although this work can have a reasonably large follow-up, the portion of the RL community interested in ranking, or vice versa, is small.

**Justification For Why Not Lower Score:**

The cascade model is one of the two most popular models for learning to rank. Therefore, this work can have a reasonably large follow-up, where more modern and realistic models for learning to rank are extended to RL. This could have a major impact on sequential search and recommendations.

---

### Decision · Program_Chairs · 2024-01-16

Accept (spotlight)